# VHL suppresses autophagy and tumor growth through PHD1-dependent Beclin1 hydroxylation

Zheng Wang[1,2,14], Meisi Yan[3,14], Leiguang Ye[4,14], Qimin Zhou ⬤[5,14], Yuran Duan[1,2], Hongfei Jiang[6], Lei Wang ⬤[1,2], Yuan Ouyang ⬤[7,8,9,10], Huahe Zhang[11,12], Yuli Shen[1,2], Guimei Ji[1,2], Xiaohan Chen[13], Qi Tian[1,2], Liwei Xiao[1,2], Qingang Wu ⬤[1,2], Ying Meng[1,2], Guijun Liu[1,2], Leina Ma[6], Bo Lei ⬤[11,12 ✉], Zhimin Lu ⬤[1,2 ✉] & Daqian Xu ⬤[1,2 ✉]

## Abstract

The Von Hippel–Lindau (VHL) protein, which is frequently mutated in clear-cell renal cell carcinoma (ccRCC), is a master regulator of hypoxia-inducible factor (HIF) that is involved in oxidative stresses. However, whether VHL possesses HIF-independent tumor-suppressing activity remains largely unclear. Here, we demonstrate that VHL suppresses nutrient stress-induced autophagy, and its deficiency in sporadic ccRCC specimens is linked to substantially elevated levels of autophagy and correlates with poorer patient prognosis. Mechanistically, VHL directly binds to the autophagy regulator Beclin1, after its PHD1-mediated hydroxylation on Pro54. This binding inhibits the association of Beclin1-VPS34 complexes with ATG14L, thereby inhibiting autophagy initiation in response to nutrient deficiency. Expression of non-hydroxylatable Beclin1 P54A abrogates VHL-mediated autophagy inhibition and significantly reduces the tumor-suppressing effect of VHL. In addition, Beclin1 P54-OH levels are inversely correlated with autophagy levels in wild-type VHL-expressing human ccRCC specimens, and with poor patient prognosis. Furthermore, combined treatment of VHL-deficient mouse tumors with autophagy inhibitors and HIF2α inhibitors suppresses tumor growth. These findings reveal an unexpected mechanism by which VHL suppresses tumor growth, and suggest a potential treatment for ccRCC through combined inhibition of both autophagy and HIF2α.

**Keywords** VHL; Beclin1; Hydroxylation; Autophagy; ccRCC
**Subject Categories** Autophagy & Cell Death; Cancer

## Introduction

Somatic mutations of the Von Hippel–Lindau tumor suppressor gene (*VHL*) are commonly detected in sporadic renal cell carcinoma (RCC), with clear-cell renal cell carcinoma (ccRCC) being the most common subtype (Foster et al, 1994; Gossage et al, 2015). The VHL protein (VHL), as the substrate recognition module of a cullin scaffold protein (CUL2)-RING ubiquitin ligase complex, binds to hypoxia-inducible factor (HIF)1/2α. This binding is dependent on the hydroxylation of two conserved proline residues in HIF1/2α by prolyl hydroxylase enzymes (PHD1, also known as EGLN2), PHD2 (EGLN1), and PHD3 (EGLN3), which require oxygen as a co-substrate (Hon et al, 2002; Ivan and Kaelin, 2017). Thus, PHD-mediated HIF1/2α hydroxylation leads to VHL-mediated HIF1/2α ubiquitylation and degradation under normoxic conditions, but not hypoxic conditions, in which the activity of PHDs is inhibited by low $O_2$ levels, resulting in enhanced HIF1/2α expression to induce downstream gene expression (Lisztwan et al, 1999; Ohh et al, 2000; Tanimoto et al, 2000). The *VHL*-defective lesions arising in the kidneys of VHL patients demonstrate an increase in HIF and HIF target genes, and mouse $VHL^{-/-}$ ccRCC xenograft experiments revealed a tumor-promoting role of HIF2α and a tumor-constraining role of HIF1α (Kaelin, 2022). However, some $VHL^{-/-}$ ccRCC lines are not affected by manipulations of HIF2α activity (Cho et al, 2016; Stransky et al, 2022), and some VHL-mutant ccRCCs were resistant to HIF2α inhibitor treatment (Choueiri and Kaelin, 2020), suggesting that deficiency of VHL promotes tumor development by yet unidentified and HIF2α-independent mechanisms.

Autophagy is frequently dysregulated in cancer cells and plays a crucial role in tumorigenesis. Cancer cells, which can have high levels of basal autophagy, upregulate autophagy to survive microenvironmental stress and to increase growth and

[1]Zhejiang Provincial Key Laboratory of Pancreatic Disease, The First Affiliated Hospital, and Institute of Translational Medicine, Zhejiang University School of Medicine, Zhejiang University, 310029 Hangzhou, China. [2]Cancer Center, Zhejiang University, 310029 Hangzhou, Zhejiang, China. [3]Department of Pathology, Harbin Medical University, Harbin, China. [4]Department of Medical Oncology, Harbin Medical University Cancer Hospital, Harbin, China. [5]Department of Plastic and Reconstructive Surgery, Shanghai Ninth People's Hospital, Shanghai Jiao Tong University School of Medicine, 200011 Shanghai, China. [6]Department of Oncology, Cancer Institute, The Affiliated Hospital of Qingdao University, Qingdao University, Qingdao Cancer Institute, 266061 Qingdao, Shandong, China. [7]Laboratory of Oral Microbiota and Systemic Diseases, Shanghai Ninth People's Hospital, College of Stomatology, Shanghai Jiao Tong University School of Medicine, Shanghai, China. [8]National Center for Stomatology, Shanghai, China. [9]National Clinical Research Center for Oral Diseases, Shanghai, China. [10]Shanghai Key Laboratory of Stomatology, Shanghai, China. [11]Department of Breast Surgery, Harbin Medical University Cancer Hospital, Harbin, China. [12]NHC Key Laboratory of Cell Transplantation, The First Affiliated Hospital of Harbin Medical University, 150001 Harbin, Heilongjiang Province, China. [13]Department of Oncology, Harbin Medical University Cancer Hospital, Harbin Medical University, 150001 Harbin, Heilongjiang Province, China. [14]These authors contributed equally: Zheng Wang, Meisi Yan, Leiguang Ye, Qimin Zhou. ✉E-mail: leibo@hrbmu.edu.cn; zhiminlu@zju.edu.cn; xudaqian@zju.edu.cn

aggressiveness by utilization of autophagic breakdown products, such as amino acids, nucleotides, carbohydrates, and fatty acids, for biosynthesis and energy generation (Kimmelman and White, 2017; Levine and Kroemer, 2008; White, 2012, 2015). During the initiation of autophagy, autophagosome nucleation requires a complex in which mammalian Beclin1 (APG6 in yeast) recruits the class III phosphatidylinositol 3-kinase VPS34 to generate phosphatidylinositol 3-phosphate (PI(3)P) for binding of the proteins with PI(3)P-binding domains, thereby modulating intracellular trafficking and autophagosome formation (Funderburk et al, 2010; Kim et al, 2013; Saito et al, 2016). ATG14L, which is in complex with VPS34, interacts with Beclin1 and targets VPS34 as well as its membrane anchor VPS15 to the pre-autophagosomal structure and promotes autophagosome–endolysosome fusion (Diao et al, 2015; Kametaka et al, 1998; Obara et al, 2006). In addition to the regulation of autophagy through ATG14L-containing Beclin1/VPS34, Beclin1/VPS34 forms a distinct complex with UV radiation resistance-associated gene (UVRAG) to regulate endocytic trafficking (Itakura and Mizushima, 2009; Kim et al, 2013). AMP-activated protein kinase (AMPK), the mechanistic target of rapamycin (mTOR)-inhibited UNC-51-like *kinase*-1 (ULK1), phosphoglycerate kinase-1 (PGK1), and Bcl-2-dependent regulation of the Beclin1/VPS34/ATG14 complex play instrumental roles in autophagic activity (Funderburk et al, 2010; Garcia and Shaw, 2017; Herzig and Shaw, 2018; Kim et al, 2013; Pattingre et al, 2005; Qian et al, 2017a; Russell et al, 2013). However, whether Beclin1/VPS34/ATG14L complex activity and autophagy are directly and differentially regulated by a tumor suppressor in a tumor-type-specific manner remains unknown.

In this report, we demonstrate that VHL suppresses nutrient stress-induced autophagy in ccRCC cells by directly binding to Beclin1 hydroxylated at P54 by PHD1 and inhibiting the association of ATG14 with Beclin1/VPS34. Beclin1 P54A knock-in expression abrogates VHL-mediated autophagy inhibition and promotes tumor growth.

## Results

### VHL inhibits nutrient deficiency-induced autophagy in ccRCC cells independent of its E3 ligase activity and its regulation of HIF2α expression

To identify HIF2α-independent tumor-suppressing function of VHL and determine whether autophagy is differentially regulated between ccRCC cells and normal kidney cells, we performed immunohistochemistry (IHC) staining of 30 ccRCC specimens with their adjacent normal tissues. We showed that the autophagy levels, which were reflected by the upregulated expression of LC3B (for the formation of autophagosomes) were much higher in tumor tissues than that in normal tissues (Appendix Fig. S1A). Notably, ccRCC specimens with VHL deficiency exhibited elevated levels of autophagy compared to wild-type (WT) VHL-expressing tumor tissues (Fig. 1A,B) and were associated with poorer prognosis of the patients (Appendix Fig. S1B). These results suggested that ccRCC has increased autophagy levels and that VHL loss promotes autophagy and is correlated with poor prognosis.

A previous report showed that p62 (a receptor encoded by *SQSTM1* for cargo destined to be degraded by autophagy) was upregulated due to the gain of *SQSTM1* copy number in ccRCC cell lines (Li et al, 2013). Consistently, IHC staining and quantitative PCR analysis of ccRCC tissues showed that both p62 expression levels and mRNA levels of *SQSTM1* were higher in tumors containing WT VHL than in their adjacent normal tissues (Appendix Fig. S1A,1C). Of note, p62 expression, but not mRNA levels of *SQSTM1*, was decreased in human ccRCC specimens containing deficient VHL expression compared to those expressing WT VHL. These results suggested that p62 protein expression is dynamically controlled by multi-layers' regulations including transcriptional and posttranslational regulations, and that elevated autophagy in VHL-deficient ccRCC cells promotes p62 degradation.

To validate the results from clinical samples, we cultured VHL-deficient 786-O or RCC4 ccRCC cells in glucose-deprived medium. As expected, glucose deprivation resulted in enhanced autophagy initiation reflected by increased PI(3)P production (Fig. 1C), enhanced autophagosome formation reflected by the augmented GFP-LC3B (Fig. 1D) and enhanced autophagosomes/autolysosomes formation reflected by endogenous LC3B puncta (Fig. 1E) appearance, and elevated conversion of LC3B-I to LC3B-II with decreased p62 expression (Fig. 1F; Appendix Fig. S1D). Consistent with a previous publication (Lonergan et al, 1998), ectopical expression of WT Flag-VHL, but not Flag-VHL C162F E3 ligase activity-dead mutant, reduced HIF2α expression (Fig. 1F; Appendix Fig. S1D), which is the primarily expressed HIF isoform in ccRCC (Hoefflin et al, 2020). However, both proteins equivalently inhibited glucose deprivation-induced autophagy in VHL-deficient ccRCC cells (Fig. 1C–F; Appendix Fig. S1D) and reduced lysosomal inhibitor chloroquine (CQ)-increased LC3B expression (Fig. 1F; Appendix Fig. S1D), suggesting that VHL inhibits initiation of autophagy rather than lysosomal activity-dependent degradation of autophagosome components. In contrast, VHL depletion in VHL-proficient SN12C and TK-10 renal carcinoma cells (Appendix Fig. S1E) dramatically enhanced glucose deprivation-induced autophagy as reflected by enhanced PI(3)P production (Appendix Fig. S1F) and elevated conversion of LC3B-I to LC3B-II with decreased p62 expression (Fig. 1G; Appendix Fig. S1G). Consistently, the enhanced PI(3)P production (Appendix Fig. S1F) and autophagy (Fig. 1G; Appendix Fig. S1G) in VHL-depleted SN12C and TK-10 cells was abrogated by reconstituted expression of WT VHL and Flag-VHL C162F accompanying with inhibited and non-inhibited HIF2α, respectively (Appendix Fig. S1F,G). Collectively, these results indicated that VHL inhibits autophagy initiation independent of its E3 ligase activity.

Of note, neither Flag-VHL expression nor VHL depletion (Fig. EV1A) affected glucose deprivation-induced AMPK activation and mTOR inhibition, as reflected by phosphorylation levels of the downstream protein substrates acetyl-CoA carboxylase (ACC)1 and S6, respectively, suggesting that VHL regulates autophagy without perturbing AMPK and mTOR activities. Similarly, neither HIF2α, HIF1β (Fig. EV1B), nor Cul2 (Fig. EV1C) depletion affected glucose deprivation-induced autophagy. In addition, VHL depletion did not alter the expression levels (Fig. EV1D) and half-life (Fig. EV1E) of Beclin1, and neither VHL or Cul2 depletion affected ubiquitylation (Fig. EV1F,G) of Beclin1. Consistently, Cul2 depletion did not affect VHL overexpression-suppressed PI(3)P production (Fig. EV1H) or autophagy (Fig. EV1I) in VHL-deficient ccRCC cells. These results suggested that VHL regulates autophagy in its E3 ligase activity- and HIF2α/HIF1β-independent manner.

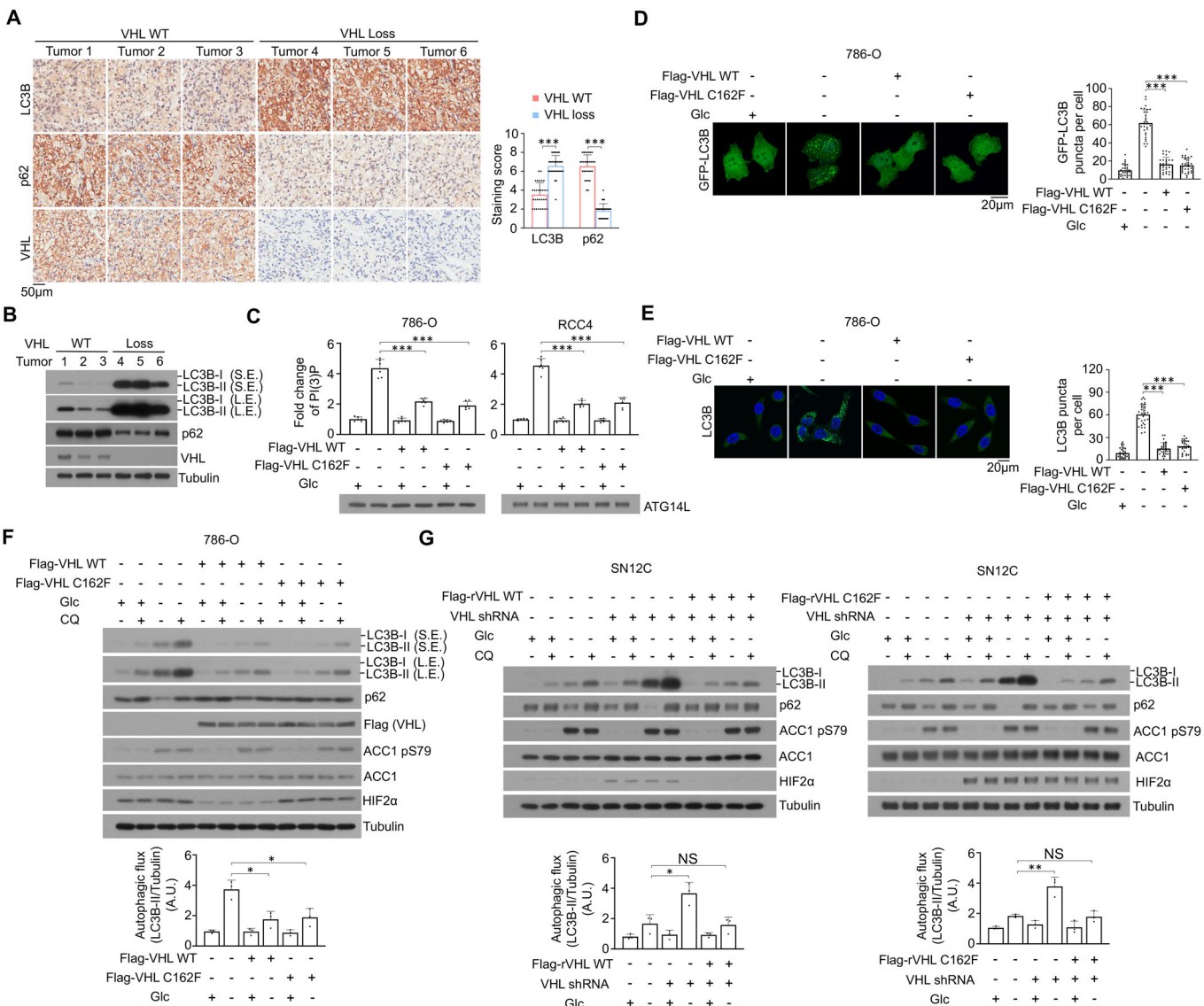

**Figure 1. VHL inhibits nutrient deficiency-induced autophagy in ccRCC cells independent of its E3 ligase activity and its regulation of HIF2α expression.**

(A) Representative immunohistochemistry (IHC) staining of ccRCC patient tissues using the indicated antibodies (left). The indicated staining scores for LC3B and p62 expression levels of different ccRCC patient tissues were compared (right). ***$P < 0.001$ by Mann–Whitney $U$ test. (B) Representative immunoblotting analyses of different human ccRCC samples were performed with the indicated antibodies. (C) Flag-VHL WT or C162F transfected 786-O and RCC4 cells were treated with or without glucose deprivation for 2 h. VPS34 complexes were immunoprecipitated by ATG14L antibody followed by PI(3)P detection by a quantitative ELISA. The PI(3)P level was normalized to the amount of ATG14L used in the assay. ***$P < 0.001$ by two-tailed Student's $t$ test. (D, E) 786-O cells were stably transduced with the indicated plasmids and treated with or without glucose deprivation for 2 h. Representative images of GFP-LC3B (D) and endogenous LC3B (E) puncta are shown (left). The numbers of LC3 puncta from 30 cells were quantitated (right). ***$P < 0.001$ two-tailed Student's $t$ test. (F) 786-O cells transfected with the indicated plasmids were treated with or without glucose deprivation in the presence or absence of 20 μM chloroquine (CQ) for 2 h. Whole-cell lysates were harvested for immunoblotting analyses as indicated. S.E. short exposure, L.E. long exposure. Autophagic flux is shown. (G) SN12C cells with or without VHL depletion and reconstituted expression of the indicated shRNA-resistant Flag-VHL proteins were treated with or without glucose deprivation in the presence or absence of 20 μM CQ for 2 h. Whole-cell lysates were harvested for immunoblotting analyses as indicated. Autophagic flux was shown. Data information: data represent the mean ± SD. Means were compared using the indicated statistical method. Immunoblotting and immunofluorescence analyses were performed three times with similar results. Source data are available online for this figure.

Alike to glucose deprivation, deprivation of amino acid, serum, or glutamine all induced autophagy; this induction was inhibited by the expression of Flag-VHL in 786-O cells (Fig. EV1J) or enhanced by the depletion of VHL in SN12C cells (Fig. EV1K). Taken together, these results indicated that VHL inhibits nutrient deficiency-induced autophagy in ccRCC cells independent of its E3 ligase activity and its regulation of HIF2α expression.

## VHL binds to Beclin1 and inhibits the association of ATG14 with the Beclin1/VPS34/VPS15 complex

To determine the mechanism underlying VHL-inhibited autophagy initiation, we performed a co-immunoprecipitation with an anti-VHL antibody and showed that VHL was associated with Beclin1, VPS34, and VPS15, but not ATG14L (Fig. 2A). As the activity of

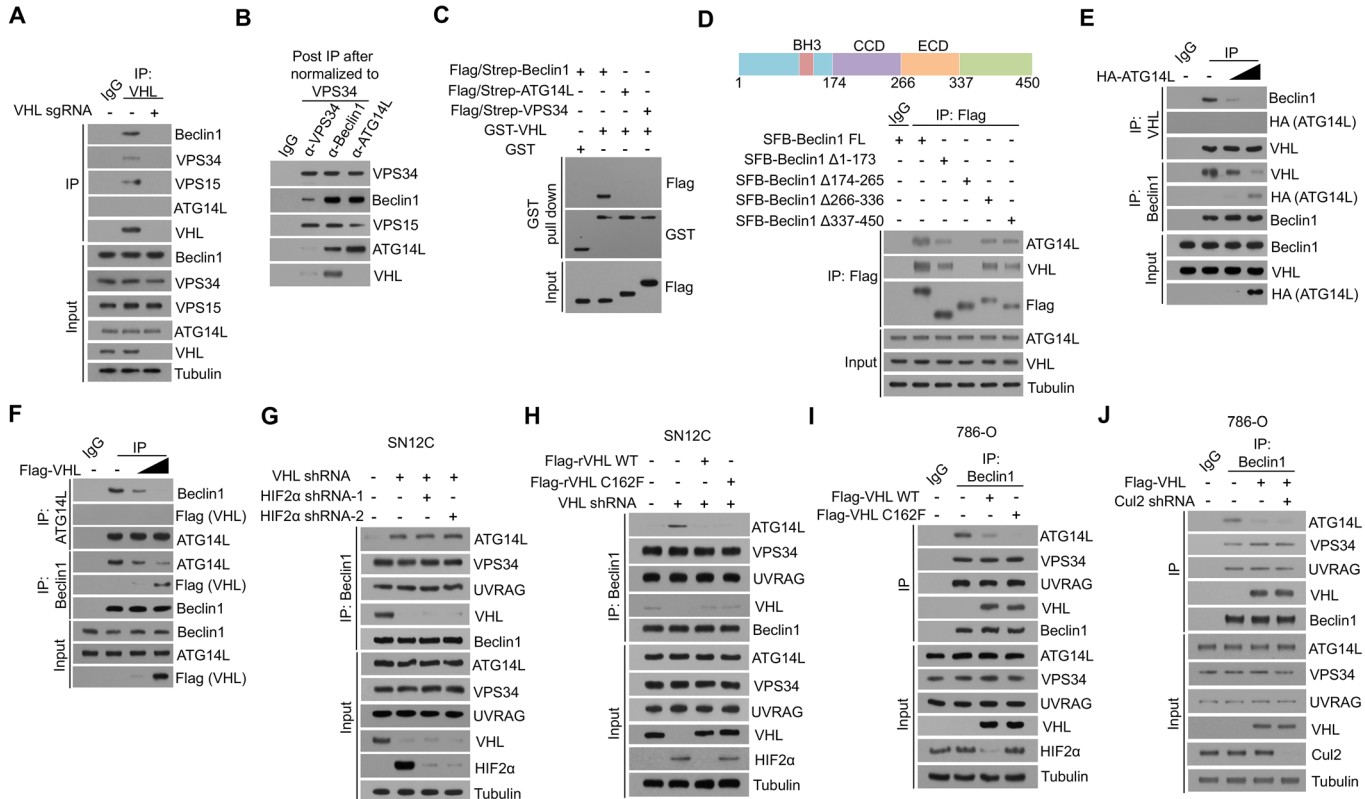

**Figure 2. VHL binds to Beclin1 and inhibits the association of ATG14 with the Beclin1/VPS34/VPS15 complex.**

(A–J) Immunoprecipitation (IP) or immunoblotting analyses were performed as indicated three times with similar results. (**A**) The cell lysates of SN12C cells were used for immunoprecipitation and immunoblotting with the antibodies as indicated. (**B**) Different VPS34 complexes of SN12C cells were immunopurified using the indicated antibodies. The relative abundance of VPS34-binding partners was determined by immunoblotting. Each immunoprecipitation was normalized to the amount of VPS34. (**C**) GST or GST-fused VHL was incubated with the indicated Flag/Strep-tagged proteins purified from 293T cells for a pull-down assay. (**D–F**) 293T cells were transiently transfected with the plasmids as indicated. (**G**) SN12C cells stably transfected with the indicated shRNA were harvested for immunoprecipitation or immunoblotting analyses as indicated. (**H**) SN12C cells with or without VHL depletion and reconstituted expression of the indicated shRNA-resistant Flag-rVHL proteins were harvested for immunoprecipitation or immunoblotting analyses as indicated. (**I, J**) 786-O cells transfected with the indicated plasmids were harvested for immunoprecipitation or immunoblotting analyses as indicated. Data information: all experiments were repeated three times independently with similar results. Source data are available online for this figure.

distinct VPS34 complexes is differentially regulated upon energy stress (Kim et al, 2013), we next determined which VPS34 pool(s) VHL is associated with. We isolated various VPS34-containing pools by immunoprecipitation with antibodies against VPS34, Beclin1, or ATG14L (Fig. 2B). After normalization to the VPS34 protein levels, we revealed that most VHL proteins were enriched in the Beclin1-associated VPS34 pool, while VPS34 precipitants contained moderate levels of VHL protein. However, no detectable level of VHL was found in the ATG14L-immunoprecipitants (Fig. 2B). A GST pull-down assay showed that GST-VHL bound to purified Flag/strep-Beclin1, but not purified ATG14L or VPS34 (Fig. 2C). These results indicated that VHL binds directly to Beclin1 without ATG14L in the same protein complex.

ATG14L is known to bind to Beclin1 CCD domain (Kang et al, 2011). We introduced different truncated variants of Beclin1 and found that CCD domain deleted Beclin1 lost its binding to both ATG14L and VHL (Fig. 2D). These results suggested that VHL and ATG14L competitively bind to the same region of Beclin1. Consistently, the binding of VHL or ATG14L to Beclin1 was dose-dependently inhibited by ATG14L (Fig. 2E) and VHL

(Fig. 2F), respectively, further suggesting that VHL and ATG14L bind to the Beclin1/VPS34/VPS15 complex in a mutually exclusive manner. Additional evidence showed that VHL depletion in VHL-proficient SN12C cells specifically increased the binding of Beclin1 to ATG14L, but not UVRAG (Fig. 2G). This increase, which was unaffected by HIF2α depletion (Fig. 2G), was diminished by reconstituted expression of either WT VHL or VHL C162F mutant (Fig. 2H). In addition, overexpression of WT VHL or VHL C162F mutant in VHL-deficient 786-O cells reduced the binding affinity of Beclin1 to ATG14L, but not UVRAG (Fig. 2I). In line with this finding, depletion of Cul2 did not affect VHL overexpression-suppressed binding of ATG14L to Beclin1 (Fig. 2J), suggesting that this suppression is independent of VHL E3 ligase activity. Consistent with the essential role of ATG14L in autophagy (Diao et al, 2015; Kametaka et al, 1998; Obara et al, 2006), the activity of VHL-associated VSP34 complex was abolished compared to that of ATG14L-associated VSP34 complex (Fig. EV1L), further supporting the inhibitory effect of VHL on VPS34-dependent PI(3)P production and subsequent autophagy initiation. These results indicated that VHL binds to Beclin1 and subsequently inhibits the

association of ATG14 with the Beclin1/VPS34/VPS15 complex, VPS34 activity, and autophagy without altering endocytic trafficking-related UVRAG-linked Beclin1/VPS34 complex.

## PHD1 is required for the binding of VHL to Beclin1 and VHL-mediated autophagy inhibition

VHL binds to PHD-hydroxylated HIF proteins (Hon et al, 2002; Ivan and Kaelin, 2017). Co-immunoprecipitation assay showed that VHL Y98H, Y111H, and W117R mutants, which are defective in binding to proline-hydroxylated proteins, were unable to bind to Beclin1 independent of HIF2α expression (Appendix Fig. S2A). This result was further validated in an in vitro GST pull-down assay, in which purified GST-fused VHL variants failed to bind to purified 293T cells-expressed Flag/strep-Beclin1

(Appendix Fig. S2B). In addition, ectopic expression of these mutants in VHL-deficient 786-O and RCC4 cells (Fig. 3A,B) or reconstituted expression of the shRNA-resistant (r) mutants in endogenous VHL-depleted SN12C and TK-10 cells (Appendix Fig. S2C,D) showed that these mutants, unlike their WT counterparts, failed to inhibit the association between Beclin1 and ATG14L (Fig. 3A; Appendix Fig. S2C), glucose deprivation-increased PI(3)P production (Fig. 3B; Appendix Fig. S2D), numbers of the GFP-LC3 puncta (Fig. 3C), endogenous LC3 puncta (Fig. 3D), LC3B-II expression, and p62 downregulation (Fig. 3E; Appendix Fig. S2E). These results indicated that the hydroxylated proline-binding ability of VHL is required for its interaction with Beclin1 and suppression of autophagy.

In line with these findings, the prolyl hydroxylase (PHD) inhibitor dimethyloxalylglycine (DMOG) increased glucose deprivation-induced

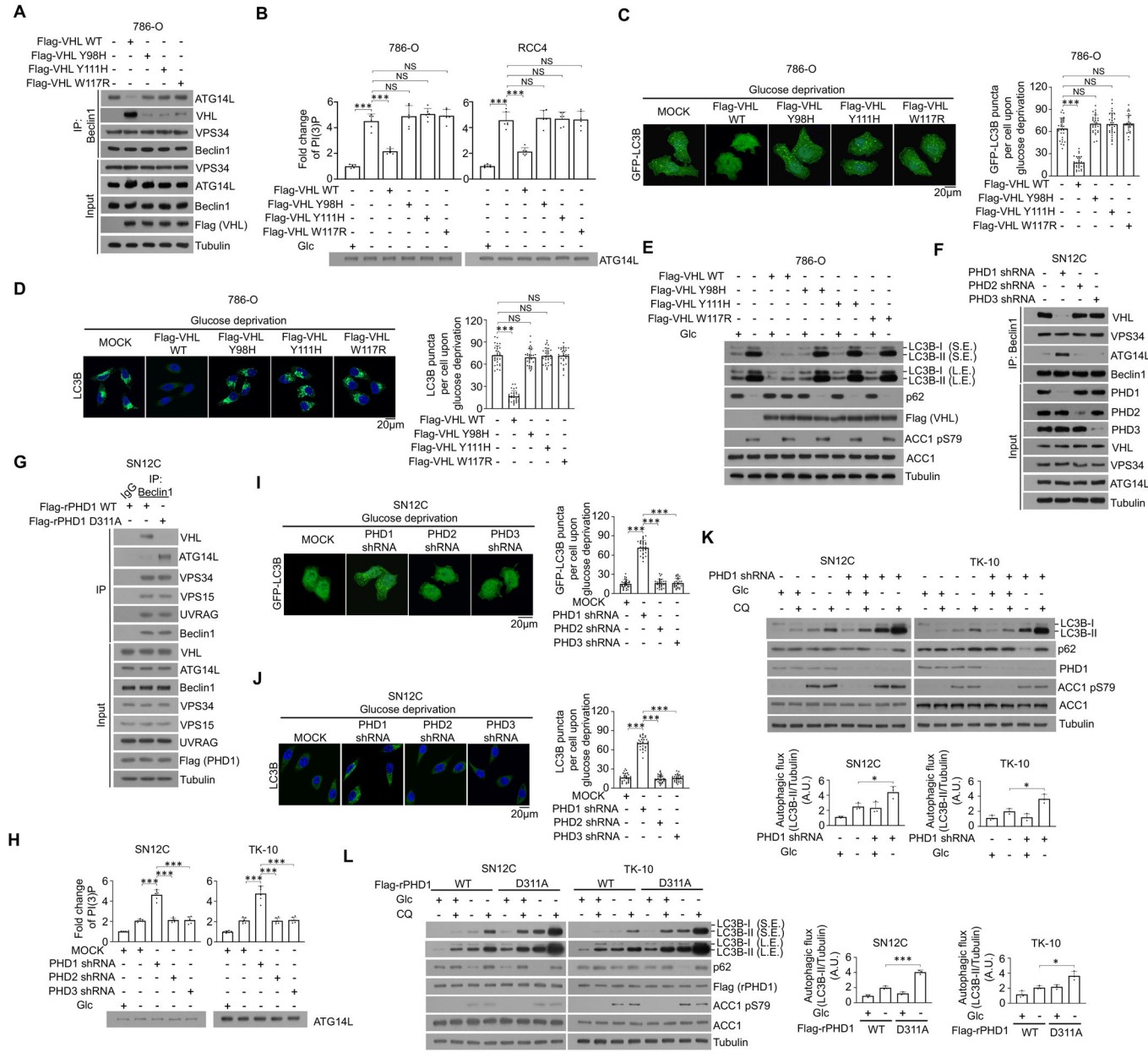

◄ **Figure 3. PHD1 is required for the binding of VHL to Beclin1 and VHL-mediated autophagy inhibition.**

(A) 786-O cells were transfected with the indicated plasmids and harvested for immunoprecipitation and immunoblotting analyses as indicated. (B) 786-O and RCC4 cells transfected with different Flag-VHL WT or mutant constructs were treated with or without glucose deprivation for 2 h. VPS34 complexes were immunoprecipitated by ATG14L antibody followed by PI(3)P detection by the quantitative ELISA. The PI(3)P level was normalized to the amount of ATG14L used in the assay. (C, D) 786-O cells transfected with different Flag-VHL WT or mutant constructs were treated with glucose deprivation for 2 h. Representative images of GFP-LC3B (C) and endogenous LC3B (D) puncta are shown (left). The numbers of LC3 puncta from 30 cells were quantitated (right). (E) 786-O cells transfected with different Flag-VHL WT or mutant constructs were treated with or without glucose deprivation for 2 h. Whole-cell lysates were harvested for immunoblotting analyses as indicated. (F) SN12C cells stably transfected with the indicated shRNA were harvested for immunoprecipitation or immunoblotting analyses as indicated. (G) SN12C cells with PHD1 depletion and reconstituted expression of the indicated shRNA-resistant Flag-rPHD1 proteins were harvested for immunoprecipitation or immunoblotting analyses as indicated. (H) SN12C and TK-10 cells stably transfected with the indicated shRNA were treated with or without glucose deprivation for 2 h. VPS34 complexes were immunoprecipitated by ATG14L antibody followed by PI(3)P detection by the quantitative ELISA. The PI(3)P level was normalized to the amount of ATG14L used in the assay. (I, J) SN12C cells stably transfected with the indicated shRNA were treated with glucose deprivation for 2 h. Representative images of GFP-LC3B (I) and endogenous LC3B (J) puncta are shown (left). The numbers of LC3 puncta from 30 cells were quantitated (right). (K) SN12C and TK-10 cells stably transfected with PHD1 shRNA were treated with or without glucose deprivation in the presence or absence of 20 μM chloroquine (CQ) for 2 h. Whole-cell lysates were harvested for immunoblotting analyses as indicated. Autophagic flux were shown. (L) SN12C and TK-10 cells with PHD1 depletion and reconstituted expression of the indicated shRNA-resistant Flag-rPHD1 proteins were treated with or without glucose deprivation in the presence or absence of 20 μM chloroquine (CQ) for 2 h. Whole-cell lysates were harvested for immunoblotting analyses as indicated. Autophagic flux are shown. Data information: data represent the mean ± SD. The statistical significance was determined using Student's t test. ***P < 0.001; NS no significance. All experiments were repeated three times with similar results. Source data are available online for this figure.

autophagy only in VHL-proficient SN12C cells (Fig. EV2A), but not VHL-deficient 786-O cells (Fig. EV2B), in a HIF2α-independent manner. Furthermore, VHL overexpression-mediated suppression of autophagy in 786-O cells was rescued by DMOG (Fig. EV2C). These results suggested that VHL inhibits nutrient stress-induced autophagy in a PHD-dependent manner.

Of note, depletion of PHD1, PHD2, and PHD3 in endogenous VHL-expressing SN12C cells (Figs. 3F and EV2D) or VHL-reexpressed 786-O cells (Fig. EV2E) showed that only PHD1 depletion inhibited the interaction between VHL and Beclin1 with correspondingly increased binding of ATG14L to Beclin1 in a HIF2α-independent manner. Consistently, expression of catalytically inactive PHD1 D311A mutant disrupted that the association of Beclin1 with VHL in ccRCC cells (Fig. 3G). In addition, depletion of PHD1, rather than PHD2 or PHD3 in VHL-proficient SN12C or TK-10 cells enhanced PI(3)P production (Fig. 3H), numbers of the GFP-LC3 puncta (Fig. 3I), endogenous LC3 puncta (Fig. 3J), LC3B-II expression, and p62 downregulation (Figs. 3K and EV2F,G) upon glucose deprivation in a HIF2α-independent manner. Consistently, PHD1 D311A expression recapitulated the effect of PHD1 depletion on the Beclin1-ATG14L interaction (Fig. EV2H) and glucose deprivation-increased PI(3)P production (Fig. EV2I), numbers of GFP-LC3 (Fig. EV2J) and endogenous LC3 (Fig. EV2K) puncta, LC3B-II expression, and p62 downregulation (Fig. 3L) in VHL-proficient SN12C or TK-10 cells. In contrast, in VHL-deficient 786-O cells, PHD1 depletion (Fig. EV2L) or reconstituted expression of catalytically inactive PHD1 D311A (Fig. EV2M) did not affect glucose deprivation-induced autophagy, but abrogated ectopic VHL expression-induced suppression of autophagy in these cells under glucose deprivation (Fig. EV2N,O). These results indicated that PHD1 is required for the binding of VHL to Beclin1 and VHL-mediated autophagy inhibition under nutrient stress conditions.

## PHD1-mediated Beclin1 P54 hydroxylation is required for the binding of VHL to Beclin1 and VHL-mediated autophagy inhibition

To determine whether Beclin1 is a substrate of PHD1, we performed a co-immunoprecipitation assay and found that PHD1, but not

PHD2 or PHD3, bound Beclin1 (Fig. EV3A). Immunofluorescence staining (Fig. EV3B) and cell fractionation analyses (Fig. EV3C) showed that PHD1 was localized in both cytosol and the nucleus of ccRCC cells. Co-immunoprecipitation analyses showed that Beclin1 interacted with PHD1 in the cytosol of the cells (Fig. EV3D). Immunoprecipitation of Beclin1 and subsequent liquid chromatography-tandem mass spectrometry/mass spectrometry (LC-MS/MS) analyses showed that Beclin1 was hydroxylated at proline (P)54 (Fig. EV3E), an evolutionarily conserved residue across different species (Fig. EV3F). In vitro hydroxylation assay showed that incubation of Beclin1 P54A mutant (Fig. 4A) or a peptide containing Beclin1 P54A mutation (Fig. 4B) abolished Beclin1 hydroxylation induced by purified bacteria-expressed WT His-PHD1, but not inactive His-PHD1 D311A, as detected by a specificity-validated anti-Beclin1 P54-OH antibody (Fig. EV3G–I). In addition, PHD1 depletion (Fig. EV3J) or reconstituted expression of inactive PHD1 D311A, but not its WT counterpart (Figs. 4C and EV3K), reduced Beclin1 P54 hydroxylation in different ccRCC cells.

Reconstituted expression of Beclin1 P54A, which lost the binding affinity to VHL, reduced ectopic VHL-mediated suppression of the interaction between ATG14L and Beclin1 in VHL-deficient 786-O and RCC4 cells (Fig. 4D). A similar change was also observed by knock-in expression of Beclin1 P54A in VHL-proficient SN12C and TK-10 renal carcinoma cells using CRISPR gene-editing technology (Figs. 4E and EV3L–O). In addition, Beclin1 P54A knock-in expression or reconstitution further enhanced glucose deprivation-induced PI(3)P production (Fig. 4F), GFP-LC3 (Fig. 4G), and endogenous LC3 (Fig. 4H) puncta formation, LC3B-II expression, and p62 downregulation (Fig. 4I) in VHL-proficient SN12C or TK-10 cells, but not in VHL-deficient 786-O or RCC4 cells (Appendix Fig. S3A–D). These results indicated that PHD1-mediated Beclin1 P54 hydroxylation is required for the binding of VHL to Beclin1 and VHL-mediated autophagy inhibition in response to nutrient stress. In addition, Beclin1 P54A knock-in expression alleviated the autophagy-suppressing effects mediated by ectopically expressed VHL in 786-O or RCC4 cells (Fig. 4J–L; Appendix Fig. S3E), indicating that VHL inhibits glucose deprivation-induced autophagy in a Beclin1 P54 hydroxylation-dependent manner. Of note, Beclin1 P54A expression lost its regulation of the VPS34 activity

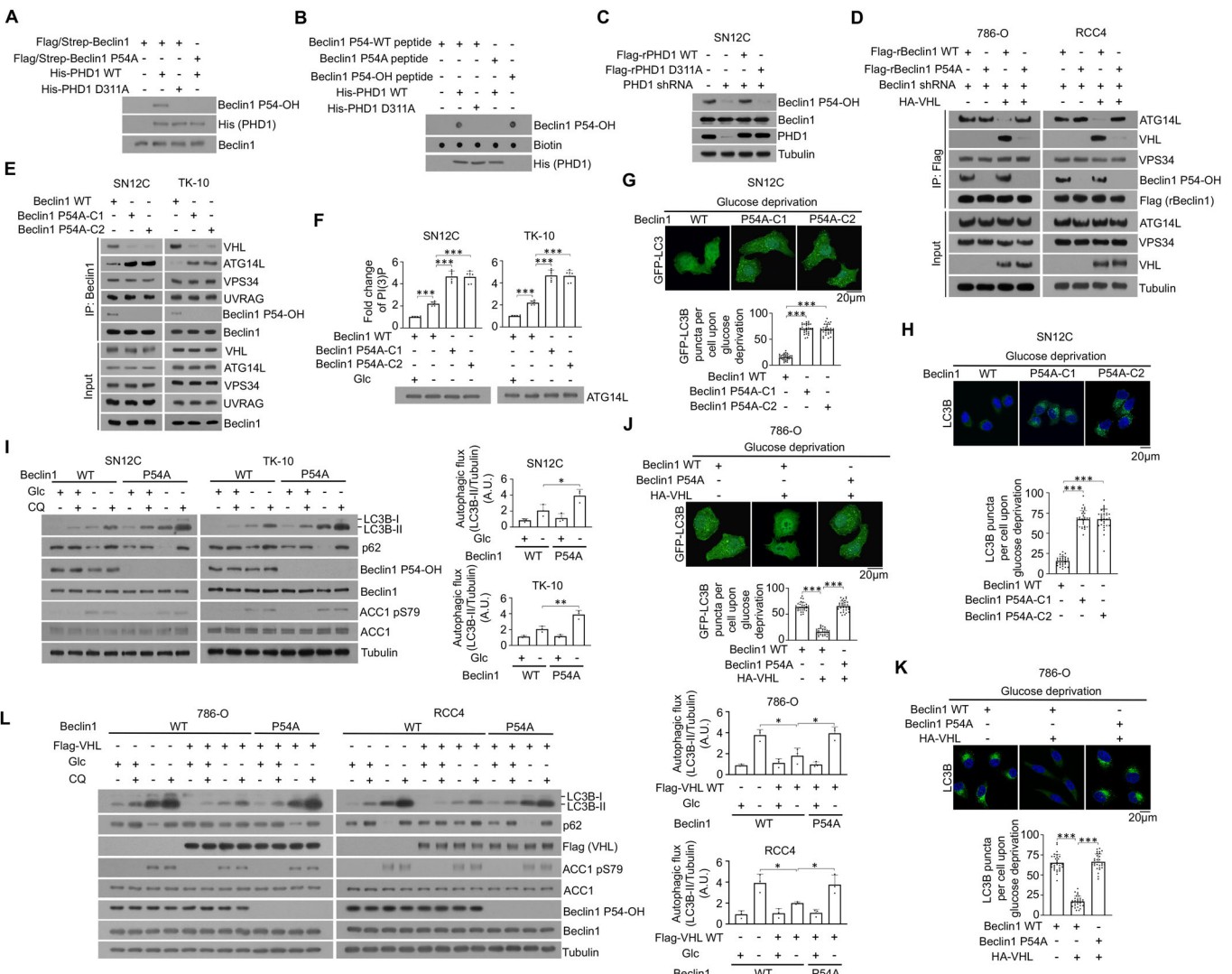

**Figure 4. PHD1-mediated Beclin1 P54 hydroxylation is required for the binding of VHL to Beclin1 and VHL-mediated autophagy inhibition.**

(A) In vitro hydroxylation assays were performed by mixing recombinant His-PHD1 WT or D311A with Flag/Strep Beclin1 proteins purified from 293T cells. Immunoblotting assays were performed as indicated. (B) In vitro hydroxylation assays were performed by mixing recombinant His-PHD1 WT or D311A with various Beclin1 peptides for dot immunoblotting analysis as indicated. (C) PHD1-depleted SN12C cells reconstituted with shRNA-resistant PHD1 WT or D311A mutant were harvested for immunoblotting analysis as indicated. (D) Beclin1-depleted 786-O and RCC4 cells reconstituted with shRNA-resistant Beclin1 WT or P54A mutant were transfected with the indicated plasmids and harvested for immunoprecipitation and immunoblotting analysis as indicated. (E) Parental SN12C and TK-10 cells as well as the indicated clones with knock-in expression of Beclin1 P54A were harvested for immunoprecipitation and immunoblotting as indicated. (F) Parental SN12C and TK-10 cells as well as the indicated clones with knock-in expression of Beclin1 P54A were treated with or without glucose deprivation for 2 h. VPS34 complexes were immunoprecipitated by ATG14L antibody followed by PI(3)P detection by the quantitative ELISA. The PI(3)P level was normalized to the amount of ATG14L used in the assay. (G, H) Parental SN12C cells and the indicated clones with knock-in expression of Beclin1 P54A were stably transfected with or without GFP-LC3B and treated with glucose deprivation for 2 h. Representative images of GFP-LC3 (G) or endogenous LC3B (H) puncta are shown (upper). The numbers of GFP-LC3 or endogenous LC3B puncta from 30 cells were quantitated (lower). (I) Parental SN12C and TK-10 cells as well as the indicated clones with knock-in expression of Beclin1 P54A were treated with or without glucose deprivation in the presence or absence of 20 μM chloroquine (CQ) for 2 h. Whole-cell lysates were harvested for immunoblotting analyses as indicated. Autophagic flux were shown. (J, K) Parental 786-O cells and the indicated clones with knock-in expression of Beclin1 P54A were transfected with HA-VHL and treated with glucose deprivation for 2 h. Representative images of GFP-LC3 (J) or endogenous LC3B (K) puncta are shown (upper). The numbers of LC3B puncta from 30 cells were quantitated (lower). (L) Parental 786-O and RCC4 cells as well as the indicated clones with knock-in expression of Beclin1 P54A were transfected with HA-VHL and treated with or without glucose deprivation for 2 h. Whole-cell lysates were harvested for immunoblotting analyses as indicated. Autophagic flux were shown. Data information: data represent the mean ± SD. The statistical significance was determined using Student's t test. ***P < 0.001; NS no significance. C1 clone 1, C2 clone 2. All experiments were repeated at least twice with similar results. Source data are available online for this figure.

(Appendix Fig. S3F) and failed to induce autophagy (Appendix Fig. S3G) in VHL-depleted SN12C cells, indicating that Beclin1 P54 hydroxylation inhibit glucose deprivation-induced autophagy in a VHL-dependent manner.

To hydroxylate protein substrates, PHD requires oxygen as a co-substrate (Hon et al, 2002; Ivan and Kaelin, 2017). As expected, hypoxic stimulation (Fig. EV4A,B), but not deprivation of glucose (Fig. EV4C) or amino acid (Fig. EV4D) under normoxic conditions,

diminished Beclin1 P54 hydroxylation, which was significantly reduced by the reducing agent Trolox or N-acetyl-L-cysteine (NAC) (Fig. EV4A), and abolished the binding of VHL to WT Beclin1 (Fig. EV4B). In addition, WT Beclin1 and Beclin1 P54A expression exhibited no differential effect on VPS34 activity (Fig. EV4E) and autophagy induction (Fig. EV4F) in VHL-proficient SN12C cells under hypoxic conditions. Consistently, hypoxic stimulation abolished VHL overexpression-suppressed or Beclin1 P54A-enhanced VPS34 activity (Fig. EV4G) and autophagy (Fig. EV4H) in the 786-O cells. These results further supported the finding that VHL suppresses autophagy in a manner depended on PHD1-mediated Beclin1 P54 hydroxylation and plays a distinct role in nutrient stress- and hypoxia-induced autophagy.

Similar to the effects on glucose deprivation-induced autophagy, reconstituted expression of Beclin1 P54A could increase amino acid starvation-induced autophagy compared to its WT counterpart in VHL-proficient ccRCC cells (Fig. EV4I). Besides, expression of Flag-VHL in the VHL-deficient ccRCC cells suppressed amino acid starvation-induced autophagy in the tumor cells expressing WT Beclin1, but not Beclin1 P54A (Fig. EV4J). These results suggested that Beclin1 P54 hydroxylation plays a critical role in the regulation of amino acid- and glucose starvation-induced autophagy.

We next examined the effect of PHD1-mediated Beclin1 P54 hydroxylation on autophagy in different types of tumor cells, including HCT116 human colon cancer cells, MDA-MB-231 human breast cancer cells, or HeLa human cervical cancer cells. We showed that depletion of PHD1 (Appendix Fig. S4A) or reconstituted expression of Beclin1 P54A (Appendix Fig. S4B,C) disrupted the binding of VHL to Beclin1 with correspondingly increased interaction between ATG14L and Beclin1 in these tumor cells. In addition, compared to the expression of WT Beclin1, Beclin1 P54A expression enhanced glucose deprivation-induced autophagy (Appendix Fig. S4D). Consistently, overexpression of Flag-VHL suppressed autophagy in these cells expressing WT Beclin1, but not Beclin1 P54A upon glucose deprivation (Appendix Fig. S4E). Thus, PHD1-dependent Beclin1 hydroxylation and VHL-suppressed autophagy is not tumor-type-specific.

## PHD1-mediated Beclin1 P54 hydroxylation is required for VHL-inhibited tumor growth

Autophagy is critical for tumor cell survival and tumor growth (White, 2012). Beclin1 P54A endogenous gene knock-in expression suppressed glucose deprivation-induced apoptosis in VHL-expressing SN12C and TK-10 cells (Fig. 5A; Appendix Fig. S5A), but not in VHL-deficient 786-O and RCC4 cells (Fig. 5B). As expected, restoration of VHL expression in 786-O and RCC4 cells increased glucose deprivation-induced cell apoptosis (Fig. 5B; Appendix Fig. S5B), and this increase was largely inhibited by Beclin1 P54A knock-in expression (Fig. 5B; Appendix Fig. S5B), indicating that Beclin1 P54 hydroxylation is required for VHL-promoted cell apoptosis under nutrient stress conditions. In addition, Beclin1 P54A-expressing and VHL-deficient 786-O and RCC4 cells exhibited no difference in tumor cell death (Fig. 5B), indicating that Beclin1 P54A suppresses glucose deprivation-induced cell apoptosis in a VHL-dependent manner. Notably, ectopic expression of non-degradable HIF2α to restore HIF2α expression levels in Flag-VHL expressed 786-O and RCC4 cells only partially alleviated Flag-VHL-induced apoptosis under glucose

deprivation conditions (Fig. 5C; Appendix Fig. S5C). These results indicated that both autophagy inhibition and HIF2α degradation mediated by VHL expression contribute to VHL-suppressed tumor cell survival.

BCL2, a pro-survival regulator, was known to interact with Beclin1 to inhibit autophagy (Pattingre et al, 2005). Co-immunoprecipitation analyses showed that both WT Beclin1 and Beclin1 P54A interacted with BCL2, and these bindings were all disrupted in glucose-starved tumor cells (Appendix Fig. S5D,E). In addition, we depleted ATG7, which is essential for autophagosome formation and autophagy occurrence and found that Beclin1 P54A expression-suppressed apoptosis was eliminated by ATG7 depletion in VHL-proficient SN12C cells (Appendix Fig. S5F,G) or VHL-ectopically expressed 786-O cells (Appendix Fig. S5H,I). These results suggested that Beclin1 P54 hydroxylation-increased cell death is independent of the interaction between Beclin1 and BCL2.

Beclin1 was shown to be required for tumor growth (Gong et al, 2013; Pan et al, 2022). Mouse studies showed that the depletion of Beclin1 in ccRCC cells (Fig. EV5A) blunted the tumor growth in mice (Fig. EV5B,C). We next examined the effect of Beclin1 P54 hydroxylation on tumor growth by subcutaneous injection of ccRCC cells. We showed that Beclin1 P54A knock-in expression in VHL-proficient SN12C cells increased tumor volumes (Fig. 5D), weight (Fig. EV5D), and expression of Ki67 (Fig. EV5E) and LC3B expression with a corresponding decrease in p62 expression (Fig. 5E) in tumor tissue. Consistent with the tumor suppressor role of VHL, VHL depletion increased tumor growth (Figs. 5D and EV5D). Notably, it eliminated the differential effects induced by Beclin1 P54A expression (Figs. 5D,E and EV5D,E). As expected, restored VHL expression in VHL-deficient 786-O cells reduced tumor volumes (Fig. 5F), weight (Fig. EV5F), and expression of Ki67 (Fig. EV5G) and LC3B-II, with a corresponding increase in p62 expression (Fig. 5G) in tumor tissue, and these changes were substantially abolished by Beclin1 P54A expression (Figs. 5F,G and EV5F,G). In contrast, the Beclin1 P54A-elicited effects were diminished in the VHL-deficient tumors (Figs. 5F,G and EV5F,G). Co-immunoprecipitation analyses showed that WT Beclin1 and Beclin1 P54A comparably interacted with BCL2 in tumor tissues (Fig. EV5H,I), further supporting that Beclin1 P54 hydroxylation regulates tumor growth in a BCL2-independent manner. These results indicated that Beclin1 P54 hydroxylation is required for the suppression of autophagy and tumor.

Given that both inhibition of autophagy and HIF2α expression contributed to VHL-elicited tumor-suppressing effects, we next treated mouse tumor derived from VHL-deficient 786-O cells with SAR405, a VPS34 inhibitor (Pasquier, 2015), and PT2385, a HIF2α inhibitor, which is currently used for advanced ccRCC patient treatment (Chen et al, 2016; Courtney et al, 2018; Wallace et al, 2016). Treatment with SAR405 or PT2385 alone reduced tumor volumes (Fig. 5H), weight (Fig. EV5J), and decreased Ki67 expression (Fig. EV5K) with increased apoptosis in tumor tissues (Fig. 5I). In addition, treatment with SAR405, but not PT2385, reduced LC3B-II expression with a corresponding increase in p62 expression in tumor tissue whereas treatment with PT2385, but not SAR405, reduced tumor tissue-expressed cyclin D1, which is encoded by CCND1, a HIF2α target gene (Choueiri and Kaelin, 2020) (Fig. EV5L), Notably, combined treatment with SAR405 and PT2385 resulted in an added effect on tumor growth inhibition

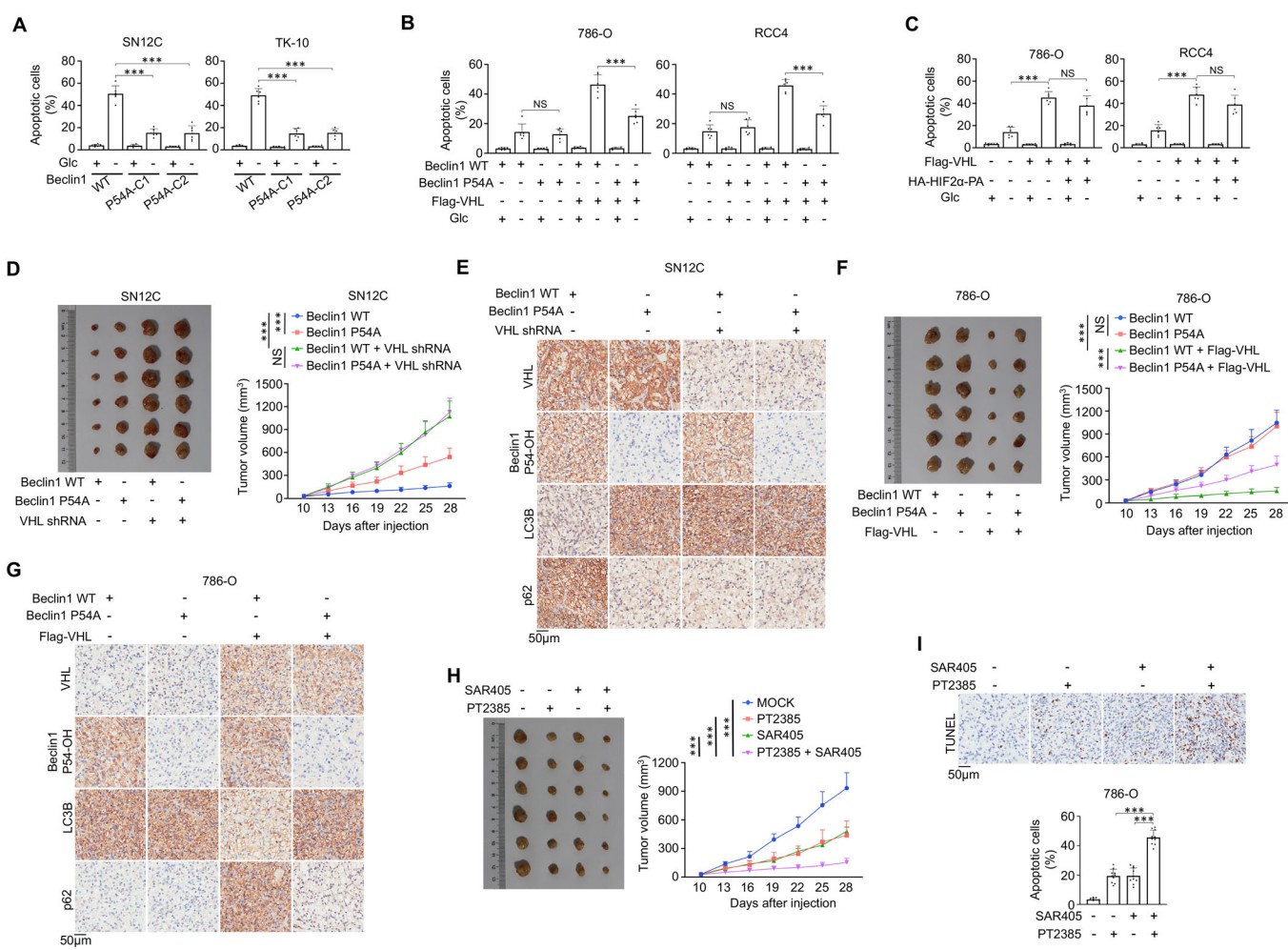

**Figure 5. PHD1-mediated Beclin1 P54 hydroxylation is required for VHL-inhibited tumor growth.**

(A) Parental SN12C and TK-10 cells and the indicated clones with Beclin1 P54A knock-in expression were treated with or without glucose deprivation for 8 h. Apoptotic cells were counted. (B, C) Parental 786-O and RCC4 cells and the indicated clones with Beclin1 P54A knock-in expression were stably transfected with the indicated plasmid and treated with or without glucose deprivation for 8 h. Apoptotic cells were counted. (D, E) Parental SN12C cells (1 × 10⁶) or the clones with Beclin1 P54A knock-in expression with or without stably transfected with VHL shRNA were subcutaneously injected into the flanks of athymic nude mice, respectively. The growth of xenografted tumors in the mice was measured (D). IHC analyses of tumor samples were performed as indicated (E). (F, G) Parental 786-O cells and the indicated clones with Beclin1 P54A knock-in expression were stably transfected with the indicated plasmid and subcutaneously injected into the flanks of athymic nude mice, respectively. The growth of xenografted tumors in the mice was measured (F). IHC analyses of tumor samples were performed as indicated (G). (H, I) 786-O cells were subcutaneously injected into athymic nude mice. When the tumor reached 50 mm³, the mice were assigned randomly into different treatment groups. PT2385 or SAR405 was intraperitoneally injected daily at a dose of 100 mg/kg until the endpoint at day 28. Tumor volumes were calculated (H). TUNEL analyses of the indicated tumor samples were performed (I, upper). Apoptotic cells were stained brown and quantified in $n = 10$ microscopic fields (I, upper). Data information: data represent the mean ± SD. The statistical significance was determined using Student's $t$ test. ***$P < 0.001$; NS no significance. C1 clone 1, C2 clone 2. Experiments were repeated at least twice with similar results. Source data are available online for this figure.

(Figs. 5H and EV5J,K) and elicited a much higher rate of tumor cell apoptosis than did SAR405 or PT2385 alone (Fig. 5I). These results revealed a potential for ccRCC treatment by inhibiting both autophagy and HIF2α activation elicited by VHL deficiency.

## Beclin1 P54-OH levels are inversely correlated with autophagy levels in human ccRCC and predict poor prognosis in patients

To further determine the clinical relevance of VHL-mediated autophagy inhibition, we performed IHC or immunoblotting analyses of human ccRCC specimens with WT-VHL expression and showed that Beclin1 P54-OH levels were inversely correlated with autophagy levels, as reflected by the increased LC3B and decreased p62 expression (Fig. 6A–C). In addition, ccRCC patients whose tumors contained low Beclin1 P54-OH (49 cases) had shorter survival duration than those whose tumors exhibited high levels of Beclin1 P54-OH (41 cases) (Fig. 6D). These results supported the role of VHL-mediated autophagy inhibition in the clinical behavior of human ccRCC and revealed a relationship between Beclin1 P54-OH levels and the clinical aggressiveness of ccRCC.

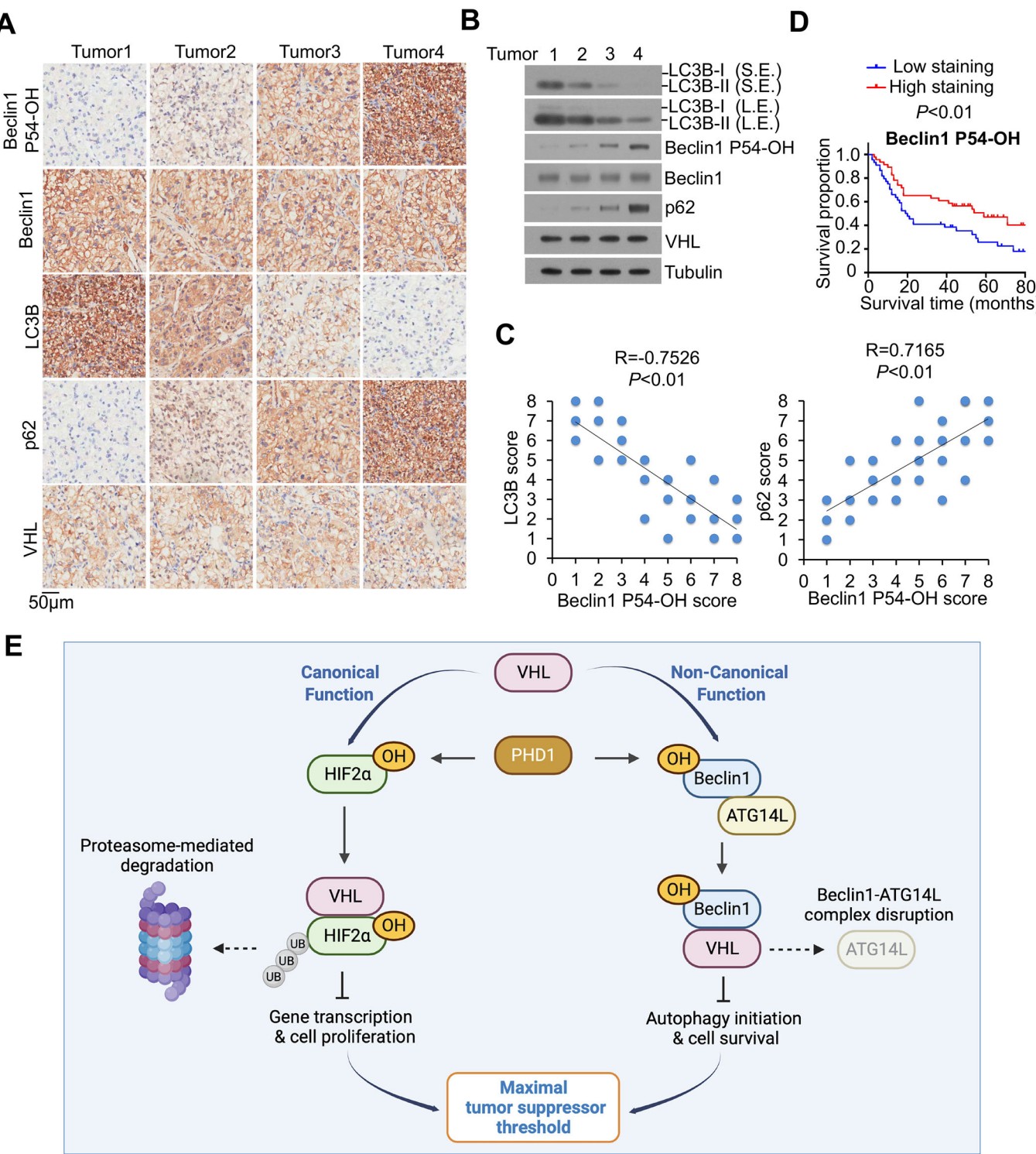

**Figure 6. Beclin1 P54-OH levels are inversely correlated with autophagy levels in human ccRCC and predict poor prognosis of patients.**

(A–C) Representative IHC and immunoblotting analyses of human ccRCC samples (VHL WT) were performed with the indicated antibodies (A, B). IHC staining of human ccRCC samples with the indicated antibodies was scored, and correlation analyses were performed (C). A Pearson correlation test was used (two-tailed) ($n = 40$). Note that the scores for some samples overlap. (D) Kaplan–Meier plots of the overall survival rates in 90 patients with ccRCC (VHL WT) grouped according to high (staining score, 4–8) and low (staining score, 0–3) expression of Beclin1 P54 hydroxylation normalized to Beclin1 protein. P values were calculated using a log-rank test (two-tailed). (E) The molecular mechanism by which VHL suppresses autophagy through PHD1-dependent Beclin1 hydroxylation. Data information: experiments were repeated twice with similar results. Source data are available online for this figure.

## Discussion

Tumor suppressor VHL deficiency or mutation frequently occurs in the majority (80–90%) of sporadic RCCs and promote tumor growth (Gossage et al, 2015). Rapid tumor growth inevitably results in nutrient and oxidative stresses that trigger autophagy to sustain the metabolic needs, survival, and proliferation of tumor cells (Li et al, 2017; Qian et al, 2017b; White, 2012). However, whether tumor cells differentially respond to these stresses and whether VHL exerts its tumor suppressor activity through direct regulation of autophagy remain unclear. We show here that VHL suppresses nutrient stress-induced autophagy. VHL deficiency in ccRCC specimens resulted in elevated levels of autophagy compared to WT VHL-expressing tumor tissues and was associated with poor prognosis of the patients. Mechanistic studies revealed that VHL directly bound to Beclin1, and this interaction was dependent on Beclin1 hydroxylation at P54, which was mediated by PHD1, but not PHD2 or PHD3. This interaction inhibited the association of ATG14L with Beclin1/VPS34 without altering the endocytic trafficking-related Beclin1/UVRAG complex, thereby inhibiting VPS34-dependent PI(3)P production and subsequent autophagy initiation, which was induced by deficiency of nutrients, such as glucose, amino acids, glutamine and serum. Beclin1 P54A knock-in expression disrupted the binding of VHL to Beclin1 and VHL-mediated autophagy inhibition, largely abolished the tumor-suppressing effect of VHL, promoted tumor growth in mice (Fig. 6E). These findings revealed a previously uncovered function of VHL in the inhibition of autophagy and unearthed a critical mechanism by which VHL suppresses tumor growth.

VHL mediated HIF degradation, and an increase in HIF2 in conjunction with HIF1 expression with increased HIF target gene expression correlates with more advanced ccRCCs (Kaelin, 2022). Intensive studies demonstrated that HIF2α, but not HIF1α, play a vital role in ccRCC progression (Kaelin, 2022); Consequently, HIF2α inhibitors, including PT2385, have been developed and used in clinical trials for ccRCC treatment (Chen et al, 2016; Courtney et al, 2018; Wallace et al, 2016). However, some VHL-mutant ccRCCs were less responsive to HIF2α depletion or HIF2α inhibitor treatment (Choueiri and Kaelin, 2020), suggesting that VHL has other cellular targets that are involved in ccRCC development. It was shown that VHL is involved in the regulation of other proteins, including zinc fingers and homeoboxes 2 (ZHX2), AKT, and TBK1 (Gossage et al, 2015; Guo et al, 2016; Hu et al, 2020; Zhang et al, 2018). Nevertheless, in vitro hydroxylation analyses revealed that recombinant PHDs do not efficiently hydroxylated these reported non-HIF substrates (Cockman et al, 2019). We demonstrated that Beclin1 P54 is hydroxylated in ccRCC cells by mass spectrometry analyses. Importantly, our results showed that PHD1 hydroxylated Beclin1 P54 both in vitro and in vivo, and this PHD1-regulated hydroxylation can be recognized by a specific Beclin1 P54 hydroxylation antibody. Functional studies showed that inhibition of PHD1-mediated Beclin1 P54 hydroxylation and autophagy by Beclin1 P54A knock-in expression or an autophagy inhibitor in combination with a HIF2α inhibitor treatment exhibited much-improved tumor growth inhibition compared to abrogation of autophagy or HIF2α alone. These results substantiated the findings that Beclin1 is a VHL target protein and that VHL exerts its tumor-suppressing function through inhibition of both HIF2α and autophagy.

The dehydroxylation of asparagine hydroxylation on intact proteins has been reported, suggesting that hydroxylation is a reversible process (Rodriguez et al, 2020). However, whether proline hydroxylation in proteins can undergo dehydroxylation remains unknown. In addition, it is unclear whether Beclin1 hydroxylation and the binding of VHL to hydroxylated Beclin1 can be mutually regulated by other protein-protein interactions and/or posttranslational modifications, such as phosphorylation, acetylation, and ubiquitylation under distinct physiological conditions. Furthermore, the role of Beclin1 modification in 786-O or RCC4 cells lacking VHL expression is not clear. Exploring whether Beclin1 hydroxylation affects autophagy-irrelevant but known or unknown Beclin1-dependent functions, such as vesicular trafficking, is of interest. Determining these aspects is crucial for understanding the dynamic regulation of autophagy and Beclin1 functions in tumor cells.

Both tumor-suppressing and promoting functions of autophagy have been reported. Accumulated evidence suggested that autophagy suppresses tumor initiation during early tumor development, but is required for supporting metabolic adaptation of established and metastasizing tumors by mitigating energy stresses and facilitating the clearance of damaged or dysfunctional organelles (Debnath et al, 2023). The effects of Beclin1-mediated autophagy on tumor growth appear to depend on tumor stages and specific activation of oncogene or inactivation of tumor suppressor, such as VHL, in a cellular context-dependent manner. Given that autophagy in tumors can be regulated simultaneously by both intrinsic and extrinsic factors, involving a complex set of inputs, further studies on autophagy in rapidly developing tumors in vivo are needed.

Compared to adjacent normal tissues, human ccRCC exhibited high levels of autophagy. In addition, low Beclin1 P54-OH levels were correlated with high levels of autophagy in WT-VHL-expressing ccRCC specimens and shorter survival duration in ccRCC patients. These findings unveil for the first time that VHL directly regulates autophagy by exclusion of ATG14L to Beclin1/VPS34 in a manner independent of its E3 ubiquitination ligase activity and HIF regulation and underscore the role of VHL deficiency-increased autophagy in tumor progression and clinical aggressiveness of the disease. Our preclinical proof-of-concept studies establish the combined inhibition of both autophagy and HIF2α as a novel strategy to treat ccRCC.

## Methods

### Materials

Rabbit antibodies against Myc tag (ab9106), VPS34 (ab227861), PHD3 (ab184714), VHL (for IHC) (ab140989), VPS15 (ab128903), p62 (ab207305), Cul2 (ab166917), LC3B (ab51520) and IgG heavy chain (HRP) (ab99702) antibodies were purchased from Abcam (Cambridge, UK). Normal mouse IgG (sc-2025), normal rabbit IgG (sc-2027), mouse anti-HA tag (sc-7392), mouse anti-Myc (SC-40), GST (sc-138), Ubiquitin (sc-271289), Biotin (sc-101339), and tubulin (sc-8035) antibodies were obtained from Santa Cruz Biotechnology (Dallas, TX). Rabbit anti-Ki67 antibody (AB9260) was obtained from Millipore (Burlington, MA). Antibodies against HIF2α (#59973), HIF1β (#5537), ACC1 pS79 (#11818), UVRAG

(#5320), ACC1 (#3676), VHL (for WB) (#68547), Beclin1 (for WB) (#3738), Beclin1 (for IP or WB) (#4122), BCL2 (#2243) (for WB), ATG14L (#96752), S6 (#2217), S6 pS240/244 (#2215), Lamin B1 (#13435), Cleaved PARP (#5625) and PHD2 (#3293) were purchased from Cell Signaling Technology (Danvers, MA). Secondary horseradish peroxidase (HRP)-coupled anti-mouse or rabbit antibodies from Jackson ImmunoResearch and anti-mouse or rabbit IgG for IP AlpSdAbs VHH (HRP) from AlpVHHs were used at 1:5000. Dimethyloxaloylglycine (DMOG), anti-His (SAB1305538), hydrogen peroxide ($H_2O_2$), CHX, MG132, Trolox, N-acetylcysteine (NAC), SAR405, PT2385, anti-Flag M2 agarose beads, EDTA-free protease inhibitor cocktail, chloroquine (CQ), mouse anti-Flag (F1804) and rabbit anti-Flag (F7425) were purchased from Sigma-Aldrich (St. Louis, MO). Beclin1 Antibody (PA5-140987) (for IF), PHD1 (PA5-96102), Cul2 (51-1800), VHL antibody (for IP or WB) (PA5-27322), glutathione agarose and DAPI were obtained from Thermo Fisher Scientific (Waltham, MA). BCL2 antibody (60178-1-Ig) (for IP) was obtained from proteintech. Beclin1 P54 hydroxylation blocking peptide (PLLTTA-QAK-P(OH)-GETQEEETN) and Beclin1 WT control peptide (PLLTTAQAKPGETQEEETN) were synthesized by Selleck-Chem. Ni-NTA agarose was obtained from Qiagen. The Beclin1 P54 hydroxylation-specific antibody was generated by Signalway Biotechnology (Pearland, TX). PI(3)P Mass ELISA was obtained from Echelon Biosciences.

## Cell lines and cell culture conditions

293T, 786-O, HeLa, HCT116, and MDA-MB-231 cells were obtained from ATCC. TK-10, SN12C, and RCC4 cells were obtained from the Cell Bank in the Chinese Academy of Sciences (Shanghai, China). All these cells were maintained in Dulbecco's modified Eagle's medium (DMEM) (GIBCO, #C11995500BT) supplemented with 10% bovine calf serum (HyClone, #SH30071). No cell lines used in this study were found in the database of commonly misidentified cell lines that is maintained by the International Cell Line Authentication Committee and the BioSample database maintained by the NCBI. Glutamine or glucose starvation was carried out by culturing cells in the culture medium without glutamine (GIBCO, #11960) or glucose (GIBCO, #11966) supplemented with dialyzed FBS, sodium pyruvate (GIBCO, #11360) and Pen Strep (GIBCO, #15140). Amino acid-free medium was made following the Invitrogen (#11965-092) high-glucose DMEM recipe with all vitamins (choline chloride, Sigma-Aldrich #C7017; calcium D-(+)-pantothenate, Santa Cruz Biotechnology #sc-202515; folic acid, Sigma-Aldrich #F8758; nicotinamide, Supelco#47865-U; pyridoxine hydrochloride, Sigma-Aldrich #P6280; riboflavin, Sigma-Aldrich #R9504; thiamine hydrochloride, Sigma-Aldrich # T1270; myo-Inositol, Sigma-Aldrich # I7508), inorganic salts (calcium chloride, Sigma-Aldrich #499609; Iron(III) nitrate nonahydrate, Sigma-Aldrich #F8508; magnesium sulfate, Sigma-Aldrich #M7506; potassium chloride, Sigma-Aldrich #P5405; sodium bicarbonate, Sigma-Aldrich #S5761; sodium chloride, Sigma-Aldrich #S9625; sodium phosphate monobasic monohydrate, Sigma-Aldrich #71507), D-glucose (Sigma-Aldrich #G7021) and phenol red (Sigma-Aldrich #P3532), but without all amino acids. Amino acid starvation contained 10% dialyzed FBS. Cell lines were authenticated by short tandem repeat profiling and were routinely tested for mycoplasma contamination. The cells

were plated at a density of $4 \times 10^5$ per 60-mm dish or $1 \times 10^5$ per well of a six-well plate 18 h before transfection. The transfection procedure was performed as previously described (Du et al, 2020).

## DNA construction and mutagenesis

PCR-amplified human VHL, ATG14L, Beclin1, Beclin1 Δ1-173, Beclin1 Δ174-265, Beclin1 Δ266-336, Beclin1 Δ337-450, VPS34, PHD1, PHD2, PHD3 and HIF2α were cloned into pcDNA3.1/hygro(+)-(Flag, myc, HA or twin Flag-streptavidin), pCDH-CMV-MCS-EF1-Puro/Neo-(Flag, HA or SFB), PGEX4T-1 (GST) or pColdI (His) vector. HIF2α-PA (P405A/P531A), VHL Y98H, Y111H, W117R, C162F, PHD1 D311A, Beclin1 P54A, short hairpin RNA (shRNA)-resistant VHL constructs containing A456T, G459A, A362T and G465A mutations, shRNA-resistant PHD1 constructs containing C1020A, G1023A and A1026C mutations, shRNA-resistant Beclin1 constructs containing C891T, G894A and A897G mutations were constructed using a Quik-Change site-directed mutagenesis kit (Stratagene, La Jolla, CA). The PLKO.1 shRNA target sequences were as follows: VHL shRNA1, 5'-TATCACACTGCCAGTGTATAC-3'; VHL shRNA2, 5'-CCATC TCTCAATGTTGACGGA-3'; EGLN2 (PHD1) shRNA1, 5'-GCTG CATCACCTGTATCTATT-3'; EGLN2 (PHD1) shRNA2, 5'-GC CAACATCGAGCCACTCTTT-3'; EGLN1 (PHD2) shRNA, 5'-GA CGACCTGATACGCCACTGT-3'; EGLN3 (PHD3) shRNA, 5'-CACCTGCATCTACTATCTGAAC-3'; BECN1 (Beclin1) shRNA, 5'-CCCGTGGAATGGAATGAGATT-3'; HIF2α shRNA1, 5'-CA GTACCCAGACGGATTTCAA-3'; HIF2α shRNA2, 5'-GCGCAA ATGTACCCAATGATA-3'; HIF1β shRNA1, 5'-ACTAGGTCCCA CAGCTAATTT-3'; HIF1β shRNA2, 5'-GAGAAGTCAGATGGTT-TATTT-3'; Cul2 shRNA1, 5'-CCCTTGGAGAAAGACTTTA TA-3', Cul2 shRNA2, 5'-GCCCTTATTCAAGAGGTGATT-3'; ATG7 shRNA, 5'-CCCAGCTATTGGAACACTGTA-3'. GFP-LC3B plasmids were generously provided by Prof. Xu Qian in Nanjing Medical University, China.

## Immunoprecipitation and immunoblotting analysis

The extraction of proteins from cultured cells was performed with a modified buffer consisting of 20 mM Tris-HCl, pH 7.5, 137 mM NaCl, 5 mM EDTA, 1% NP-40, 10% glycerol, 50 mM NaF, 1 mM $Na_3VO_4$ and protease inhibitor cocktail, and was followed by immunoprecipitation and immunoblotting using corresponding antibodies as described previously (Xu et al, 2015), (Harada et al, 2003; Rui et al, 2004). Briefly, cell extracts were clarified by centrifugation at 12,000 rpm, and the supernatants were subjected to immunoblotting or immunoprecipitation with the indicated antibodies. After overnight incubation with antibodies at 4 °C, protein A/G agarose beads were added and left for an additional 3 h. Immunocomplexes were then subjected to immunoblotting analysis by SDS-PAGE electrophoresis.

## CRISPR–Cas9-mediated genome editing

Genomic mutations were introduced into cells using the CRISPR–Cas9 system, as described previously. Single-guide RNAs (sgRNAs) were designed to target the genomic area adjacent to mutation sites in Beclin1 P54 using the CRISPR design tool (http://crispr.mit.edu/). The annealed guide RNA oligonucleotides

were inserted into a PX458 vector (Addgene) digested with the BbsI restriction enzyme (Xu et al, 2019). Cells were seeded at 60% confluence, followed by co-transfection of sgRNAs (0.5 µg) and single-stranded donor oligonucleotides (10 pmol) as a template to introduce mutations. Twenty-four hours after transfection, cells were trypsinized, diluted for single cells and seeded into 96-well plates. Genomic DNA was extracted from GFP-positive cells, followed by sequencing of the PCR products spanning the mutation sites. The sgRNA-targeting sequence for Beclin1 P54 was 5′-ACCACAGCCCAGGCGAAACC-3′; single-stranded donor oligo-nucleotide (ssODN) sequence for Beclin1 P54 was 5′-CAG ATGCCCTCCTGCTTTAATAAGACTGTTTCTGACCCAATATT TTCCTTGCCCTTAGCTCCATTACTTACCACAGCCCAaGCaA AggCAGGAGAGACCCAGGAGGAAGAGACTAACTCAGGAGA GGTAATAGAAGTGCCCTCTCCCCTATCCTCCTCATGTAGTA AA-3′. The lower-case letters in the ssODN sequences indicate the mutated nucleotides that will replace the endogenous nucleotides in the genomic DNA of parental cells using the CRISPR–Cas9 system. VHL-KO and Beclin1-KO cell lines were generated using the CRISPR–Cas9 system (Xu et al, 2019). The sgRNA-targeting sequence for VHL-KO is 5′-TACGGCCCTGAAGAAGACGG-3′. The sgRNA-targeting sequence for Beclin1-KO is 5′-GTCCAA-CAACAGCACCATGC-3′. Genotyping was performed by sequen-cing PCR products amplified from the following primers: Beclin1 forward: 5′-TAATGGGGGTGAGGAAGACCA-3′; Beclin1 reverse: 5′-TTCTTTCTGTCCAGGGCAATCA-3′. VHL-KO forward: 5′-CCTCCGTTACAACGGCCTAC-3′; VHL-KO reverse: 5′-TCGAAGTTGAGCCATACGGG-3′. Beclin1-KO forward: 5′-ACC CTCACGGCTCTTATTGG-3′; Beclin1-KO reverse: 5′-GGAAA GCTCTCAGAAGTCCAC-3′.

## GST pulldown

Equal amounts of His-tagged purified protein (200 ng/sample) were incubated with 100 ng of GST fusion proteins together with glutathione agarose beads in modified binding buffer (50 mM Tris-HCl, pH 7.5; 1% Triton X-100; 150 mM NaCl; 1 mM DTT; 0.5 mM EDTA; 100 µM PMSF; 100 µM leupeptin; 1 µM aprotinin; 100 µM sodium orthovanadate; 100 µM sodium pyrophosphate; and 1 mM sodium fluoride). The glutathione agarose beads were then washed four times with the binding buffer and subjected to immunoblot-ting analysis as previously described (Qian et al, 2019b).

## Purification of recombinant proteins

GST-VHL and its different mutations were expressed in bacteria and purified as described previously (Qian et al, 2019a). Flag/streptavidin double-tagged Beclin1, ATG14L or VSP34 expressed in 293T cells was enriched by anti-Flag agarose beads and eluted with 3XFlag peptide, respectively. After ultrafiltration and removal of the 3XFlag peptide from the eluted protein, a secondary streptavidin pull-down assay was performed.

## In vitro hydroxylation assays

In vitro hydroxylation assays were performed as described previously (Guo et al, 2016). In brief, peptides (10 µg) or purified recombinant proteins (1 µg) were mixed with 1 µg of purified recombinant PHD1 proteins in a 30 µL reaction buffer [50 mM

HEPES (pH 7.4), 1500 U/µL Catalase, 100 µM FeSO4, 1 µM ascorbic acid, 0.2 µM α-ketoglutarate (α-KG)] at 37 °C for 1 h, and then the reactions were subjected to immunoblotting or mass spectrometry analysis.

## Mass spectrometry analysis

Transfected Flag/Strep Beclin1 were immunoprecipitated in SN12C cells by anti-Flag antibody on protein A/G agarose beads, followed by elution with Flag peptide (Thermo Fisher Scientific, #A36805), and then were desalted with Zeba Spin desalting Columns (Thermo Fisher Scientific, #89882). Immunoprecipitated Beclin1 proteins were reduced with TCEP, subjected to alkylation, and digested with GluC at 37 °C overnight. The sample was acidified by trifluoroacetic acid and analyzed by LC-MS/MS on an Orbitrap-Elite mass spectrometer (Thermo Fisher Scientific, Waltham, MA), as described previously (Qian et al, 2017b).

## In vitro PI(3)P ELISA

The total amount of PI(3)P was examined in a quantitative and competitive ELISA format assay according to the manufacturer's instructions (Echelon Biosciences K-3000) as previously described (Xu et al, 2016). Briefly, different VPS34 complexes were enriched by various antibodies and then mixed with phosphatidylinositol substrates in 2× kinase reaction buffer (100 mM HEPES pH 7.5, 300 mM NaCl, and 2 mM CHAPS, 10 mM MnCl$_2$, 2 mM DTT, and 100 µM ATP) to react at 37 °C for 1 h. After the PI3K reactions were complete and quenched, reaction products were diluted and added to the PI(3)P-coated microplate for competitive binding to a PI(3)P detector protein. The amount of PI(3)P detector protein bound to the plate was determined through colorimetric detection.

## Autophagy assay

Cells grown on gelatinized coverslips were transfected with or without GFP-LC3 plasmids. Forty-eight hours later, the medium was changed to complete or glucose-deprived medium. The fluorescence of puncta from Green Fluorescent Protein (GFP) of exogenous GFP-LC3 or Alexa Fluor 488 dye-conjugated antibody recognizing endogenous LC3B was observed under a confocal microscope (ZEISS LSM900) with the same parameters. The numbers of puncta were counted manually in 30 cells by setting 488 channel pictures as 8-bit type and adjusting the threshold at the same parameters in Image J software. Autophagic flux was calculated by analyzing the difference of LC3B-II/Tubulin levels in the presence versus the absence of CQ.

## Subcellular fractionation

Nuclear and cytosolic fractions were isolated from the cells using a nuclear/cytosol fractionation kit (K266, BioVision) according to the manufacturer's instructions. Briefly, cells were collected by centrifugation at 600 × g for 5 min at 4 °C, and resuspended with cytosol extraction buffer containing DTT and protease inhibitors. Collect the supernatant (Cytoplasmic extract) fraction by centrifuge at 16,000 × g for 5 min at 4 °C. Then resuspend the pellet in nuclear extraction buffer and collect supernatant by centrifuge at 16,000 × g for 10 min after incubating for 40 min at 4 °C.

## Apoptosis analysis

The number of apoptotic cells was analyzed by using DAPI staining as described previously (Lee et al, 2018). Cells ($1 \times 10^5$) were seeded in a six-well plate and cultured overnight. The cells were then fixed by the direct addition of formaldehyde (final concentration of 12%) to the culture medium. After fixation, the cells were stained with DAPI (1 μg/mL) for 5 min, followed by washing with PBS. Cells with condensed and/or fragmented chromatin were determined to be undergoing apoptosis and counted (Qian et al, 2018). For a TUNEL assay, tumor tissues were sectioned at 5-μm thickness. Apoptotic cells were counted using a Dead-End colorimetric TUNEL system (Promega) according to the manufacturer's instructions.

## Dot immunoblotting assays

Biotin labeled peptides were spotted onto nitrocellulose membrane slowly to penetrate. The membrane was dried and blocked in PBST buffer with 5% non-fat milk for 30 min, and then subjected for immunoblot analysis using indicated primary antibodies and HRP-coupled secondary antibodies as previously reported (Guo et al, 2016; Zhang et al, 2018).

## IHC analysis and histological evaluation of human ccRCC specimens

We purchased the human ccRCC tissue array (HkidE180Su02) from Shanghai Outdo Biotech (http://www.superchip.com.cn/) under the approval by the Ethics Committee of Taizhou Hospital, Zhejiang, China. Sections of paraffin-embedded human ccRCC samples were stained with antibodies against LC3B, p62, VHL, or Beclin P54-OH and nonspecific IgG as a negative control. The staining of the tissue sections was quantitatively scored according to the percentage of positive cells and the staining intensity as described previously (Yang et al, 2011). The following proportion scores were assigned to the sections, 0 if 0% of the tumor cells exhibited positive staining, 1 for 0–1% of cells stained, 2 for 2–10% of cells stained, 3 for 11–30% of cells stained, 4 for 31–70% of cells stained and 5 for 71–100% of cells stained. In addition, the staining intensity was rated on a scale of 0–3: 0, no staining; 1, weak; 2, moderate; and 3, strong. The proportion and intensity scores were then combined to obtain a total score (range, 1–8) as described previously (Xu et al, 2020). The scores were compared with overall survival duration, defined as the time from the date of diagnosis to that of death or last known follow-up examination. All patients had received standard therapies after surgery.

## Animal studies

One million ccRCC cells were collected in 20 μL of DMEM with 33% Matrigel and subcutaneously injected into the livers of 6-week-old male athymic BALB/c nude mice. The injections were performed as described previously (Li et al, 2016). On the 4th day after injection, the mice were assigned randomly into different treatment groups (Six mice per group). PT2395 or SAR405 was intraperitoneally injected daily at a dose of 100 mg/kg. The tumor volume was measured every 3 days until the 28th day after injection. The tumor of each euthanized mouse was dissected and then fixed in 4% formaldehyde and embedded in paraffin. Tumor formation and phenotype were determined by histologic analysis of H&E-stained sections. The tumor volume was calculated using the formula: $V = 1/2a^2b$ (V, volume; a, shortest diameter; and b, longest diameter). The animals were treated in accordance with relevant institutional and national guidelines and regulations. The use of the animals was approved by the Institutional Review Board at The First Affiliated Hospital, Zhejiang University School of Medicine, Hangzhou and Qingdao Cancer Institute, China. No statistical method was used to predetermine the sample size. The investigators were not blinded to treatment allocation during experiments or to the outcome assessment.

## Statistical analyses and reproducibility

All statistical data are presented as the means ± standard deviation (SD). All experiments were repeated at least twice independently with similar results. The mean values obtained in the control and experimental groups were analyzed for significant differences. Pairwise comparisons were performed using a two-tailed $t$ test. $P$ values of less than 0.05 were considered significant. Unless stated otherwise, the experiments were not randomized, and the investigators were not blinded to treatment allocation during experiments or to the outcome assessment.

# Data availability

This study includes no data deposited in external repositories.

# Peer review information

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

## Acknowledgements

This study was supported by grants from the Ministry of Science and Technology of the People's Republic of China (2021YFA0805600, DX; 2020YFA0803300, ZL), the National Natural Science Foundation of China (82072630, 92157113, DX; 82188102, 82030074, ZL; 82372814, 82173114, ZW; 82002811, MY; 82103658, QZ), Shanghai Pujiang Program (No. 2022PJD040, QZ), Natural Science Foundation of Heilongjiang Province of China (LH2023H095, B.L.), The Project of Beijing Medical Award Foundation (YXJL-2021-0581-0484, BL), the Zhejiang Natural Science Foundation Key Project (LD22H160002, DX; LD21H160003, ZL) and Zhejiang Natural Science Foundation Discovery Project (LQ22H160023, ZW). ZL is the Kuancheng Wang Distinguished Chair.

## Author contributions

**Zheng Wang**: Data curation; Investigation; Methodology; Project administration. **Meisi Yan**: Data curation; Investigation; Methodology. **Leiguang Ye**: Formal analysis; Validation; Investigation; Methodology. **Qimin Zhou**: Conceptualization; Data curation; Investigation; Methodology. **Yuran Duan**: Conceptualization; Data curation. **Hongfei Jiang**: Data curation; Methodology. **Lei Wang**: Investigation. **Yuan Ouyang**: Conceptualization; Methodology. **Huahe Zhang**: Data curation. **Yuli Shen**: Validation. **Guimei Ji**: Data curation. **Xiaohan Chen**: Validation. **Qi Tian**: Validation. **Liwei Xiao**: Formal analysis. **Qingang Wu**: Software; Visualization. **Ying Meng**: Visualization; Methodology. **Guijun Liu**: Formal analysis. **Leina Ma**: Conceptualization. **Bo Lei**: Funding acquisition; Project administration; Writing—review and editing. **Zhimin Lu**: Supervision; Funding acquisition; Writing—original draft; Project administration; Writing—review and editing. **Daqian Xu**: Supervision; Funding acquisition; Writing—original draft; Project administration; Writing—review and editing.

## Disclosure and competing interests statement

ZL owns shares in Signalway Biotechnology (Pearland, TX), which supplied rabbit antibodies that recognize Beclin1 P54 hydroxylation. ZL's interest in this company had no bearing on its being chosen to supply these reagents. The remaining authors declare no competing interests.

# Expanded View Figures

**Figure EV1.  VHL inhibits nutrient deficiency-induced autophagy in ccRCC cells in its E3 ligase activity- and HIF2α/HIF1β-independent manner.**

(A) Flag-VHL transfected 786-O cells (left) or VHL-depleted SN12C cells (right) were treated with glucose deprivation for the indicated time. (B) 786-O cells stably transfected with HIF2α (left) or HIF1β (right) shRNA were treated with or without glucose deprivation for 2 h. (C) SN12C and TK-10 cells stably transfected with Cul2 shRNA were treated with or without glucose deprivation for 2 h. (D) SN12C and TK-10 cells stably transfected with VHL shRNA were harvested for immunoblotting analyses as indicated. (E) SN12C and TK-10 cells stably transfected with VHL shRNA were treated with CHX (100 μg/ml) for the indicated periods of time. The quantification of Beclin1 protein levels relative to tubulin levels is shown. (F, G) SN12C cells stably transfected with the indicated constructs were harvested for immunoprecipitation and immunoblotting analyses as indicated. (H) 786-O and RCC4 cells stably transfected with the indicated constructs were treated with or without glucose deprivation for 2 h. VPS34 complexes were immunoprecipitated by ATG14L antibody followed by PI(3)P detection by a quantitative ELISA. The PI(3)P level was normalized to the amount of ATG14L used in the assay. (I) 786-O cells transfected with the indicated plasmids were treated with or without glucose deprivation in the presence or absence of 20 μM chloroquine (CQ) for 2 h. (J) 786-O cells with or without Flag-VHL transfection were treated with amino acid (AA) deprivation for 1 h, serum deprivation for 9 h or glutamine (Gln) deprivation for 4 h, in the presence or absence of 20 μM chloroquine (CQ) for 2 h, respectively. (K) SN12C cells with or without VHL shRNA transfection were treated with amino acid (AA) deprivation for 1 h, serum deprivation for 9 h or glutamine (Gln) deprivation for 4 h, respectively. (L) SN12C and TK-10 cells were treated with or without glucose deprivation for 2 h. VPS34 complexes were immunoprecipitated by ATG14L or VHL antibodies followed by PI(3)P detection by a quantitative ELISA. The PI(3)P level was normalized to the amount of VPS34 used in the assay. Data information: Data represent the mean ± SD. The statistical significance was determined using two-tailed Student's *t* test. ***P < 0.001; NS no significance. (A–G, I–K) Immunoprecipitation or immunoblotting analyses were performed. (B, C, I, J) Autophagic flux were shown. All experiments were repeated three times with similar results.

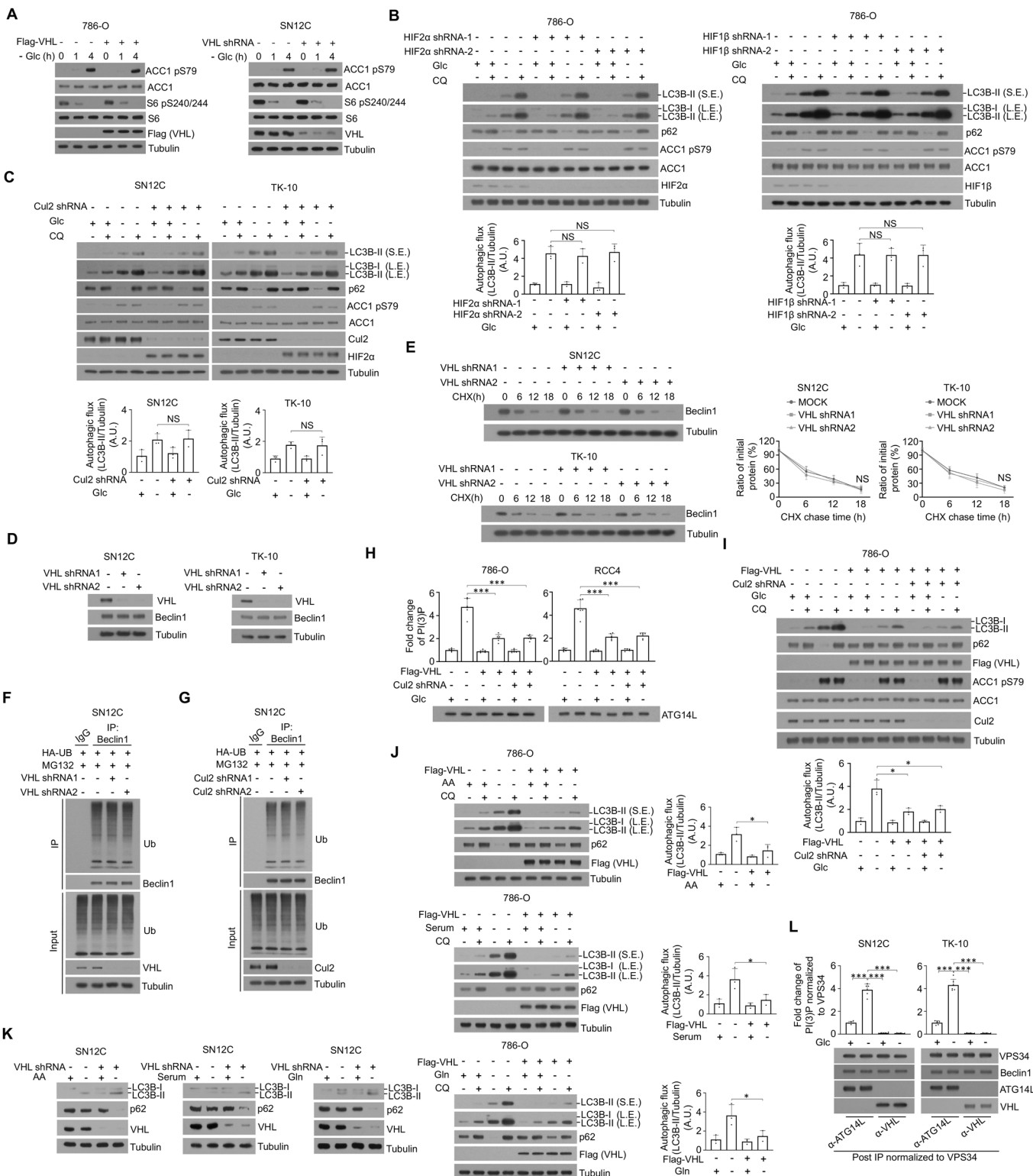

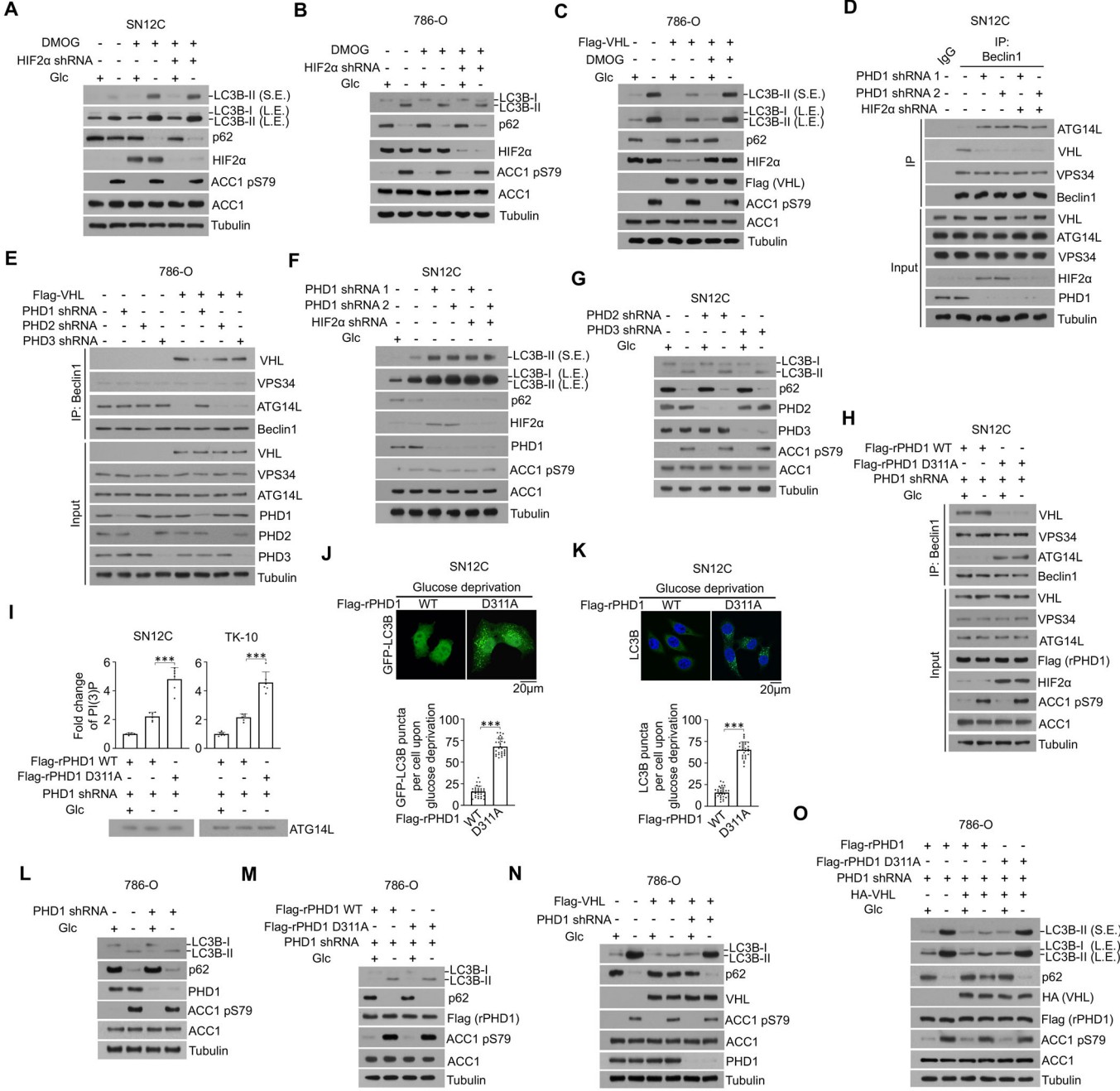

◀ **Figure EV2. PHD1 is required for the binding of VHL to Beclin1 and VHL-mediated autophagy inhibition.**

(A, B) SN12C (A) or 786-O (B) cells with and without HIF2α depletion were pretreated with or without DMOG (200 μM) for 1 h before glucose deprivation for 2 h. (C) Flag-VHL-transfected 786-O cells were pretreated with or without DMOG (200 μM) for 1 h before glucose deprivation for 2 h. (D, E) SN12C (D) or 786-O (E) cells transfected with the indicated plasmids were harvested. (F, G) SN12C cells transfected with the indicated shRNA were treated with or without glucose deprivation for 2 h. (H) SN12C cells with or without PHD1 depletion and reconstituted expression of the indicated shRNA-resistant Flag-PHD1 proteins were treated with or without glucose deprivation for 2 h. (I) PHD1 depleted SN12C or TK-10 cells with reconstituted expression of the indicated shRNA-resistant Flag-PHD1 proteins were treated with or without glucose deprivation for 2 h. VPS34 complexes were immunoprecipitated by ATG14L antibody followed by PI(3)P detection by the quantitative ELISA. The PI(3)P level was normalized to the amount of ATG14L used in the assay. (J, K) PHD1 depleted SN12C cells with reconstituted expression of the indicated shRNA-resistant Flag-rPHD1 proteins were treated with glucose deprivation for 2 h. Representative images of GFP-LC3B (J) or endogenous LC3B (K) puncta are shown (upper). The numbers of LC3B puncta from 30 cells were quantitated (lower). (L) 786-O cells with or without PHD1 depletion were treated with or without glucose deprivation for 2 h. (M) Endogenous PHD1 depleted 786-O cells with reconstituted expression of the indicated shRNA-resistant Flag-PHD1 proteins were treated with or without glucose deprivation for 2 h. (N) 786-O cells with or without PHD1 depletion were transfected with the indicated plasmids and treated with or without glucose deprivation for 2 h. (O) 786-O cells with reconstituted expression of the indicated shRNA-resistant Flag-rPHD1 proteins were transfected with the indicated plasmids and treated with or without glucose deprivation for 2 h. Data information: data represent the mean ± SD. The statistical significance was determined using two-tailed Student's *t* test. ***$P < 0.001$. (A–H, L–O) Immunoprecipitation and/or immunoblotting analyses were performed as indicated. All experiments were repeated at least twice with similar results.

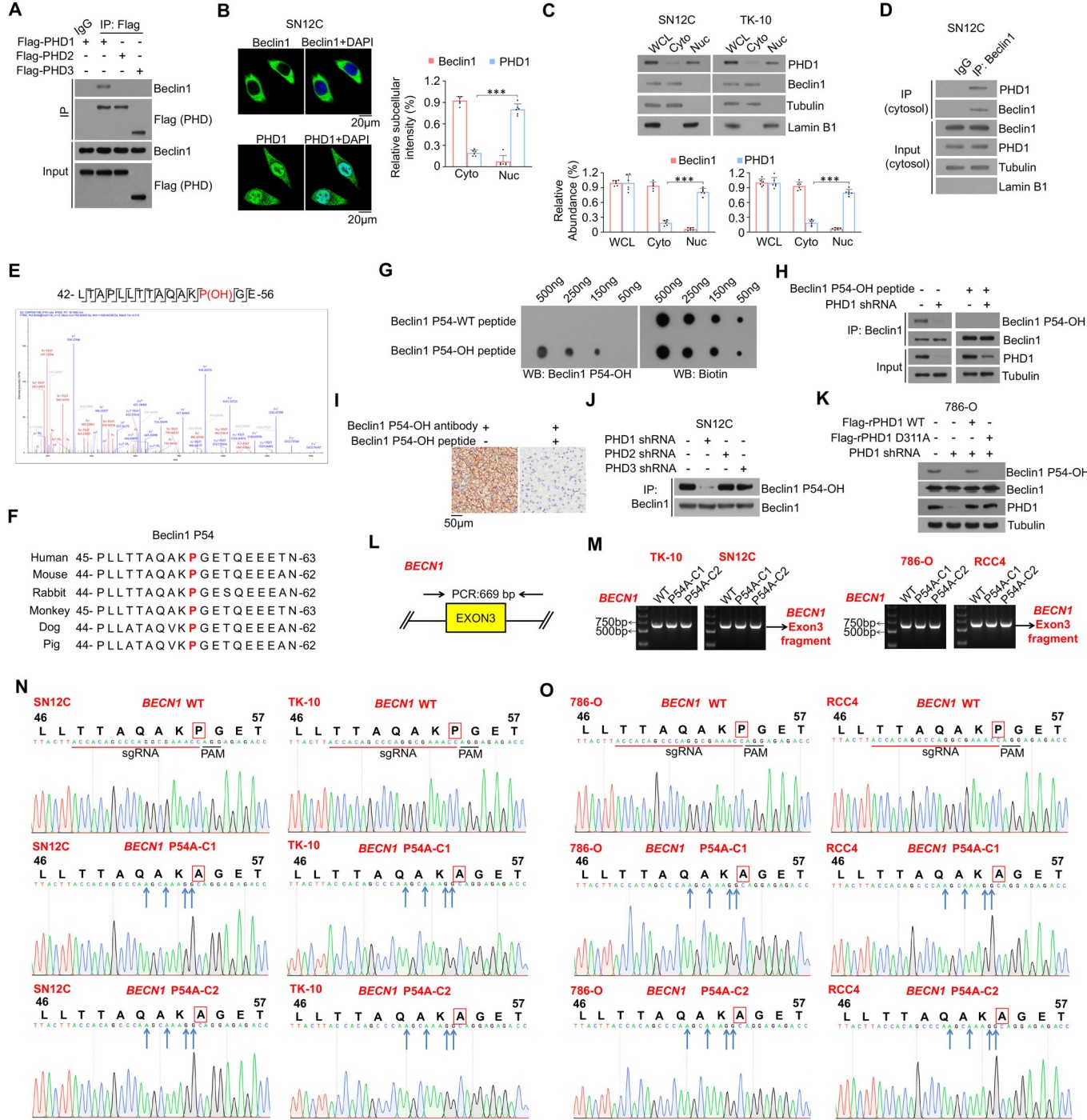

◀ **Figure EV3. PHD1 hydroxylates Beclin1 P54.**

(A) 293T cells transfected with the indicated plasmids were harvested for immunoprecipitation and immunoblotting analyses as indicated. (B) Immunofluorescence analyses were performed with the indicated antibodies (left). The relative subcellular distribution intensity of Beclin1 and PHD1 is shown (right). (C) Cytosolic and nuclear fractions of SN12C and TK-10 cells were prepared. The relative Beclin1 and PHD1 abundance in different fractions was quantified by densitometric analysis of the blots ($n = 6$). $n$ represents independent biological replicate. (D) The cytosolic fraction of SN12C cells were harvested for immunoprecipitation and immunoblotting analyses as indicated. (E) Flag/Strep Beclin1 immunoprecipitated from SN12C cells was subjected to liquid chromatography-tandem mass spectrometry/mass spectrometry (LC-MS/MS) analyses. Representative LC-MS/MS spectra showing hydroxyl-proline-containing fragments derived from Beclin1. Mass spectrometric analysis of a tryptic fragment at m/z 1391.02930 Da ($+ 0.06$ mmu/$+0.08$ ppm), which was matched with the $+2$ charged peptide 42-LTAPLLTTAQAKPGE-56, suggested that Beclin1 P54 was hydroxylated. The XCorr score was 4.47. (F) Alignment of protein sequences spanning Beclin1 Pro54 from different species. (G) Dot immunoblotting analyses were performed with the indicated synthetic peptides diluted with different concentrations and detected with immunoblotting analyses as indicated. (H) SN12C cells transfected with the indicated plasmids were harvested for immunoprecipitation and immunoblotting analyses in the presence or absence of a blocking peptide for Beclin1 P54 hydroxylation. (I) IHC analyses of human ccRCC samples were performed with the indicated antibodies in the presence or absence of a blocking peptide for Beclin1 P54 hydroxylation. (J) SN12C cells stably transfected with the indicated shRNA were harvested for immunoprecipitation and immunoblotting analyses as indicated. (K) 786-O cells reconstituted with the PHD1 WT or D311A mutant were harvested for immunoprecipitation and immunoblotting analyses as indicated. (L–O) Genomic DNA was extracted from two individual clones of different parental ccRCC cells with knock-in expression of Beclin1 P54A. PCR products were amplified from the indicated DNA fragment (L) and separated on an agarose gel (M). Sequencing of different parental ccRCC cells and two individual clones with knock-in expression of Beclin1 P54A (N, O). The red line indicates the sgRNA-targeting sequence. The black line indicates the protospacer adjacent motif (PAM). Blue arrows indicate mutated nucleotides. A mutated amino acid and its wild-type counterpart are indicated by the solid red box. Data information: Data represent the mean ± SD. The statistical significance was determined using two-tailed Student's $t$ test. ***$P < 0.001$. WCL whole-cell lysate, Cyto cytosol, Nuc nucleus. Experiments were repeated at least twice with similar results.

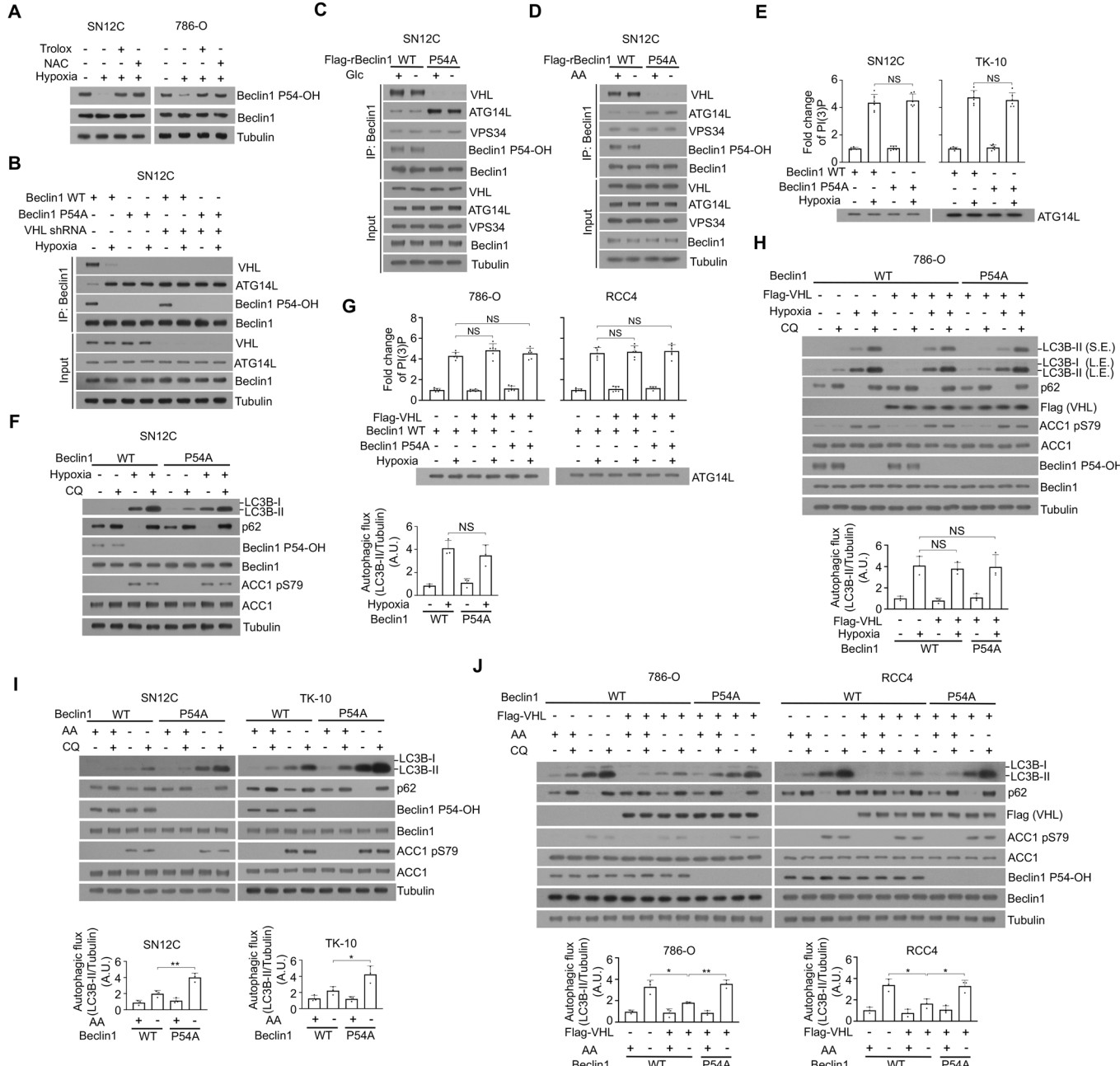

◀ **Figure EV4. VHL suppresses autophagy in a manner depended on PHD1-mediated Beclin1 P54 hydroxylation and plays a distinct role in nutrient stress- and hypoxia-induced autophagy.**

(A) SN12C and 786-O cells were pretreated with or without N-acetyl-L-cysteine (NAC) (5 mM) or Trolox (100 mM) for 1 h before hypoxia stimulation for 8 h. The whole-cell lysates were harvested for immunoblotting analyses as indicated. (B) Parental SN12C cells and the indicated clones with knock-in expression of Beclin1 P54A were stably transfected with or without VHL shRNA and treated with hypoxia for 8 h. (C, D) Endogenous Beclin1 depleted SN12C cells with reconstituted expression of the indicated shRNA-resistant Flag-Beclin1 proteins were treated with or without glucose deprivation for 2 h (C) or amino acid (AA) deprivation for 1 h (D), respectively. (E) Parental SN12C and TK-10 cells and the indicated clones with knock-in expression of Beclin1 P54A were treated with hypoxia for 8 h. VPS34 complexes were immunoprecipitated by ATG14L antibody followed by PI(3)P detection by quantitative ELISA. The PI(3)P level was normalized to the amount of ATG14L used in the assay. (F) Parental SN12C cells and the indicated clones with knock-in expression of Beclin1 P54A were treated with hypoxia in the presence or absence of 20 μM chloroquine (CQ) for 8 h. (G) Parental 786-O and RCC4 cells and the indicated clones with knock-in expression of Beclin1 P54A with or without Flag-VHL transfection were treated with or without hypoxia for 8 h. VPS34 complexes were immunoprecipitated by ATG14L antibody followed by PI(3)P detection by quantitative ELISA. The PI(3)P level was normalized to the amount of ATG14L used in the assay. (H) Parental 786-O cells and the indicated clones with knock-in expression of Beclin1 P54A were transfected with or without Flag-VHL and treated with or without hypoxia in the presence or absence of 20 μM chloroquine (CQ) for 8 h. (I) Parental SN12C and TK-10 cells and the indicated clone with knock-in expression of Beclin1 P54A were treated with or without amino acid (AA) deprivation in the presence or absence of 20 μM chloroquine (CQ) for 1 h. (J) Parental 786-O and RCC4 cells and the indicated clones with knock-in expression of Beclin1 P54A transfected with or without Flag-VHL were treated with or without amino acid (AA) deprivation in the presence or absence of 20 μM chloroquine (CQ) for 1 h. Data information: data represent the mean ± SD. The statistical significance was determined using two-tailed Student's *t* test. NS no significance. (A–D, F, H–J) Immunoprecipitation and/or immunoblotting analyses were performed as indicated. (F, H–J) Autophagic flux were shown. All experiments were repeated at least twice with similar results.

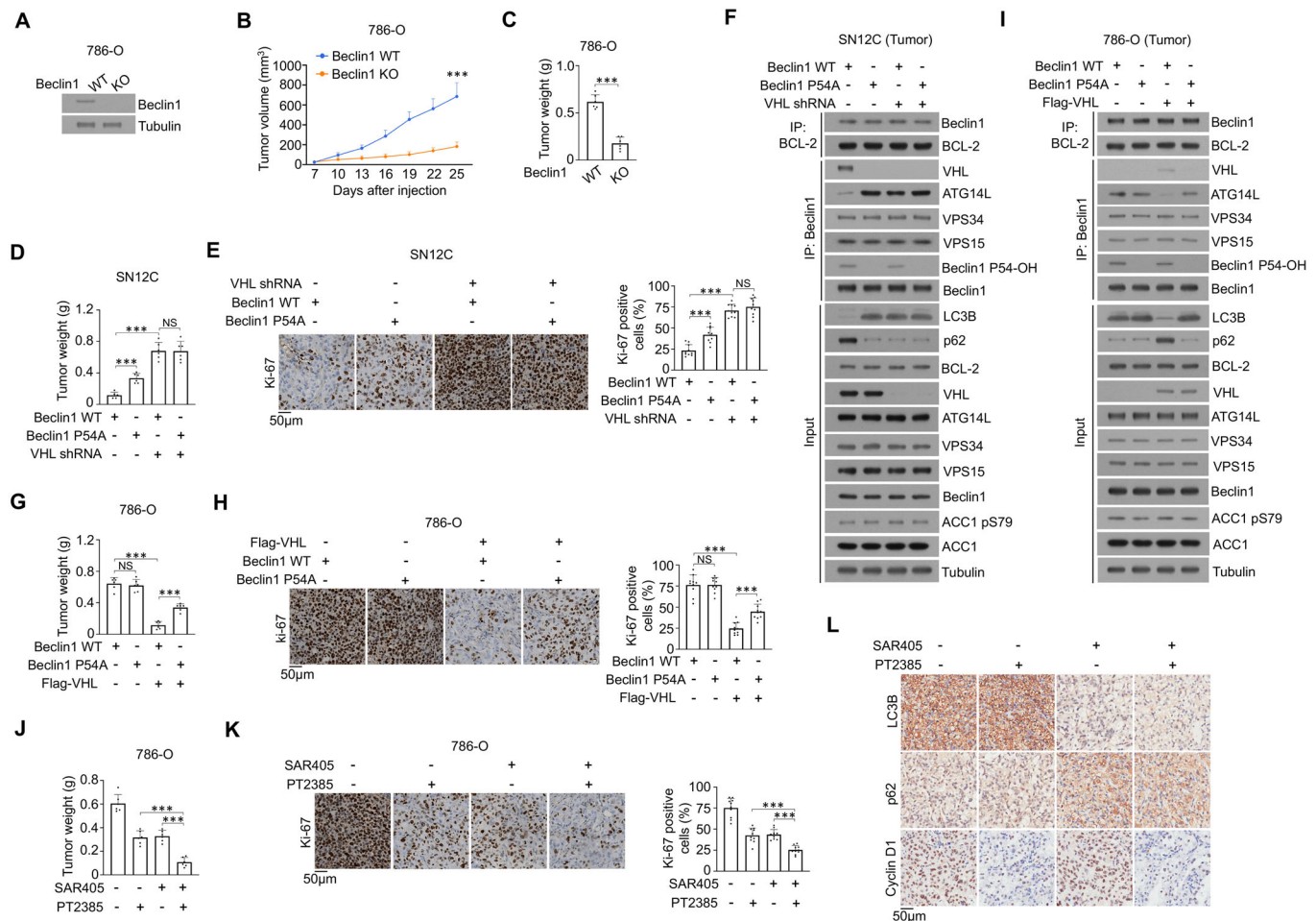

**Figure EV5.  PHD1-mediated Beclin1 P54 hydroxylation is required for VHL-inhibited tumor growth.**

(A–C) The whole-cell lysates of WT and Beclin1 knockout 786-O cells were harvested for immunoblotting analyses as indicated (A). WT and Beclin1 knockout 786-O cells ($1 \times 10^6$) were subcutaneously injected into the left or right flanks of 6-week-old male athymic nude mice ($n = 6$). n represents the number of independent animals in each group. The resulting tumors were resected 25 days after injection. Tumor volume and weight were analyzed. ***$P < 0.001$ by Student's two-tailed $t$ test (B, C).

(D–F) Parental SN12C cells ($1 \times 10^6$) or the clones with Beclin1 P54A knock-in expression stably transfected with or without VHL shRNA were subcutaneously injected into the left or right flanks of athymic nude mice, respectively. The weight of xenografted tumors in the mice was measured (D). IHC analyses of tumor samples were performed as indicated (E, left). Ki67-positive cells were quantified in 10 microscopic fields (E, right). Immunoprecipitation and immunoblotting analyses of the indicated tumors were performed with the indicated antibodies (F). Data represent the mean ± SD. ***$P < 0.001$; NS no significance by Student's two-tailed $t$ test. (G–I) Parental 786-O cells ($1 \times 10^6$) and the indicated clones with Beclin1 P54A knock-in expression stably transfected with or without Flag- VHL were subcutaneously injected into the flanks of athymic nude mice, respectively. The weight of xenografted tumors in the mice was measured (G). IHC analyses of tumor samples were performed as indicated (H, left). Ki67-positive cells were quantified in 10 microscopic fields (H, right). Immunoprecipitation and immunoblotting analyses of the indicated tumors were performed with the indicated antibodies (I). Data represent the mean ± SD. ***$P < 0.001$; NS no significance by Student's two-tailed $t$ test. (J–L) 786-O cells were subcutaneously injected into athymic nude mice. When the tumor reached 50 mm³, the mice were assigned randomly into different treatment groups. PT2385 or SAR405 was intraperitoneally injected daily at a dose of 100 mg/kg until the endpoint at day 28. The weight of xenografted tumors in the mice was measured (J). IHC analyses of tumor samples were performed as indicated (K, left and L). Ki67-positive cells were quantified in 10 microscopic fields (K, right). Data represent the mean ± SD. ***$P < 0.001$ by Student's two-tailed $t$ test.

