## [Peer Review File · The EMBO Journal]

VHL suppresses autophagy and tumor growth through PHD1-dependent Beclin1 hydroxylation

Zheng Wang, Meisi Yan, Leiguang Ye, Qimin Zhou, Yuran Duan, Hongfei Jiang, Lei Wang, Yuan Ouyang, Huahe Zhang, Yuli Shen, Guimei Ji, Xiaohan Chen, Qi Tian, Liwei Xiao, Qingang Wu, Ying Meng, Guijun Liu, Leina Ma, Bo Lei, Zhimin Lu, Daqian Xu

Corresponding authors: Bo Lei (leibo@hrbmu.edu.cn), Zhimin Lu (zhiminlu@zju.edu.cn) and Daqian Xu (xudaqian@zju.edu.cn)

Review Timeline:

Submission Date:	9th Mar 23
Editorial Decision:	19th Apr 23
Revision Received:	17th Sep 23
Editorial Decision:	16th Oct 23
Revision Received:	21st Nov 23
Editorial Decision:	7th Dec 23
Revision Received:	27th Dec 23
Editorial Decision:	18th Jan 24
Revision Received:	23rd Jan 24
Accepted:	24th Jan 24

Editor: William Teale

Transaction Report:

Dear Dr Lu and Dr Xu,

Thank you again for the submission of your manuscript entitled "VHL suppresses autophagy and tumor growth through PHD1-dependent Beclin1 hydroxylation" (EMBOJ-2023-113983) and for your patience during the review process. We have now received the reports from the referees, which I copy below.

As you can see from their comments, all referees were impressed by the exciting connection your data make between VHL and autophagy in the context of renal carcinoma cells. However, they all also highlight areas in which the manuscript should be improved. These comments will require your attention before your manuscript can be published in The EMBO Journal.

Based on the overall interest expressed in the reports, I would like to invite you to address the comments of all referees in a revised version of the manuscript. I should add that it is The EMBO Journal policy to allow only a single major round of revision and that it is therefore important to resolve the main concerns at this stage. I believe the concerns of the referees are reasonable and addressable, but please contact me if you have any questions, need further input on the referee comments or if you anticipate any problems in addressing any of their points. I am always prepared to go through the referee reports with authors via Zoom. Please, follow the instructions below when preparing your manuscript for resubmission.

I would also like to point out that as a matter of policy, competing manuscripts published during this period will not be taken into consideration in our assessment of the novelty presented by your study ("scooping" protection). We have extended this 'scooping protection policy' beyond the usual 3 month revision timeline to cover the period required for a full revision to address the essential experimental issues. Please contact me if you see a paper with related content published elsewhere to discuss the appropriate course of action.

Again, please contact me at any time during revision if you need any help or have further questions.

Thank you very much again for the opportunity to consider your work for publication. I look forward to your revision.

Best regards,

William

William Teale, Ph.D.
Editor
The EMBO Journal

When submitting your revised manuscript, please carefully review the instructions below and include the following items:

- 1) a .docx formatted version of the manuscript text (including legends for main figures, EV figures and tables). Please make sure that the changes are highlighted to be clearly visible.
- 2) individual production quality figure files as .eps, .tif, .jpg (one file per figure).
- 3) a .docx formatted letter INCLUDING the reviewers' reports and your detailed point-by-point response to their comments. As part of the EMBO Press transparent editorial process, the point-by-point response is part of the Review Process File (RPF), which will be published alongside your paper.
- 4) a complete author checklist, which you can download from our author guidelines ([https://wol-prod-cdn.literatumonline.com/pb-assets/embo-site/Author Checklist%20-%20EMBO%20J-1561436015657.xlsx](https://wol-prod-cdn.literatumonline.com/pb-assets/embo-site/Author%20Checklist%20-%20EMBO%20J-1561436015657.xlsx)). Please insert information in the checklist that is also reflected in the manuscript. The completed author checklist will also be part of the RPF.
- 5) Please note that all corresponding authors are required to supply an ORCID ID for their name upon submission of a revised manuscript.
- 6) We require a 'Data Availability' section after the Materials and Methods. Before submitting your revision, primary datasets produced in this study need to be deposited in an appropriate public database, and the accession numbers and database listed under 'Data Availability'. Please remember to provide a reviewer password if the datasets are not yet public (see

<https://www.embopress.org/page/journal/14602075/authorguide#datadeposition>). If no data deposition in external databases is needed for this paper, please then state in this section: This study includes no data deposited in external repositories. Note that the Data Availability Section is restricted to new primary data that are part of this study.

Note - All links should resolve to a page where the data can be accessed.

8) For data quantification: please specify the name of the statistical test used to generate error bars and P values, the number (n) of independent experiments (specify technical or biological replicates) underlying each data point and the test used to calculate p-values in each figure legend. The figure legends should contain a basic description of n, P and the test applied. Graphs must include a description of the bars and the error bars (s.d., s.e.m.).

9) We would also encourage you to include the source data for figure panels that show essential data. Numerical data can be provided as individual .xls or .csv files (including a tab describing the data). For 'blots' or microscopy, uncropped images should be submitted (using a zip archive or a single pdf per main figure if multiple images need to be supplied for one panel). Additional information on source data and instruction on how to label the files are available at .

10) We replaced Supplementary Information with Expanded View (EV) Figures and Tables that are collapsible/expandable online (see examples in <https://www.embopress.org/doi/10.15252/embj.201695874>). A maximum of 5 EV Figures can be typeset. EV Figures should be cited as 'Figure EV1, Figure EV2" etc. in the text and their respective legends should be included in the main text after the legends of regular figures.

12) Our journal encourages inclusion of *data citations in the reference list* to directly cite datasets that were re-used and obtained from public databases. Data citations in the article text are distinct from normal bibliographical citations and should directly link to the database records from which the data can be accessed. In the main text, data citations are formatted as follows: "Data ref: Smith et al, 2001" or "Data ref: NCBI Sequence Read Archive PRJNA342805, 2017". In the Reference list, data citations must be labeled with "[DATASET]". A data reference must provide the database name, accession number/identifiers and a resolvable link to the landing page from which the data can be accessed at the end of the reference. Further instructions are available at .

Further instructions for preparing your revised manuscript:

We realize that it is difficult to revise to a specific deadline. In the interest of protecting the conceptual advance provided by the work, we recommend a revision within 3 months (18th Jul 2023). Please discuss the revision progress ahead of this time with the editor if you require more time to complete the revisions.

Referee #1:

In this study, Wang et al. demonstrated a new role and mechanism of von Hippel-Lindau (VHL) as a tumor suppressor in regulating autophagy activity in sporadic clear-cell renal cell carcinoma cells. VHL is a component of the CUL2-RING E3 ligase and is known to bind and ubiquitinate HIF1/2. The authors found that VHL interacts with the autophagy protein Beclin1, which competitively inhibits Beclin1-ATG14 binding and the autophagy activity. Similar to VHL-HIF binding, VHL-Beclin1 binding is mediated by PHD1 and Beclin1 hydroxylation, at P54. VHL-Beclin1 binding is important to suppress autophagy and tumor growth in xenograft models. Overall, it is a well-designed study demonstrating a novel HIF-independent mechanism by which VHL regulates autophagy and suppresses tumor growth, and the significance is high. There are several specific questions that need to be addressed to further strengthen the conclusions.

Specific comments:

1. The study relies mostly on overexpressed proteins. A key question to address is whether binding of VHL to Beclin1 leads to changes in endogenous Beclin1 protein level (via ubiquitination as to HIF). Ubiquitination experiments are needed to rule out that Beclin1 is a substrate of the VHL-CUL2-RING E3, using endogenous proteins.
2. Fig. 1, the condition of Glc+ (basal condition) with and without CQ should be added in Fig. 1G and Fig. EV1F, as based on Fig. 1F, the autophagy-inhibiting effect of VHL is more apparent under basal conditions (Glc+).
3. Fig. 6, the Beclin1 P54-OH level and score in patient tumor samples should be normalized to Beclin1 protein expression levels, because high staining of Beclin1 P54-OH may be a reflection of high Beclin1 expression.
4. An important premise of the study is that autophagy is tumor-promoting. To directly support it, Beclin1 KO xenografts should be included in Fig. 5 to test whether they form smaller tumors as a control.
5. CQ should be added to many panels to demonstrate the autophagy flux, such as Fig. 4H and EV11-K.
6. A discussion should be included to reconcile the facts that Beclin1-mediated autophagy is considered tumor-promoting while Beclin1 also functions as a tumor suppressor.
7. Some conclusions in the texts are not accurate and need proofreading. For example, pg 11 "... Beclin1 P54A knock-in expression alleviated the autophagy-suppressing effects mediated by ectopically expressed VHL in 786-O or RCC4 cells (Fig 4J-M), indicating that Beclin1 P54 hydroxylation inhibit glucose deprivation induced autophagy in a VHL-dependent manner"; it should be "VHL inhibits ... autophagy in a P54-dependent manner".

Referee #2:

The manuscript presents an interesting finding that VHL acts as a negative regulator of autophagy in renal carcinoma cells. The

authors found that PHD1, whose activity is suppressed by hypoxia or oxidative stress, hydroxylates a proline residue in Beclin 1, a key regulator of autophagy. As a consequence, the modification enables VHL to recognize and bind to the modified Beclin 1. This binding disrupts the interaction between Beclin 1 and Atg14L, an essential interaction for autophagy, resulting in autophagy suppression. The authors show that this role of VHL is important for its ability to suppress tumor growth.

Overall, the presented results are interesting and supportive of the proposed mechanism. If the mechanism is fully elucidated, it may define VHL as a novel negative regulator of autophagy. The link between PHD1/VHL and autophagy has a potential to advance our understanding of how cancer cells survive and grow under hypoxia conditions. The identification of the hydroxylation site of Beclin 1 has been solidly demonstrated. Despite the recognized merits, there are multiple issues that need to be addressed for further consideration.

1. The study has only investigated the regulation in specific renal cancer cell lines. Therefore, it is unclear whether the regulation is a general mechanism that broadly regulates the canonical autophagy pathway in cells or if it is specific to certain cell types.
2. It is unclear why the authors tested glucose starvation instead of amino acid starvation. There is controversy surrounding whether glucose starvation can robustly induce autophagy. Even if it can induce autophagy in some cells, it is generally considered to be less effective than amino acid starvation. The presented data on autophagy assays after glucose starvation is unconvincing. The starvation effects on autophagy flux in wild type cells are not drastic, and the effects of overexpression/knockout/knockdown are not strong throughout the presented data.
3. Throughout the presented figures, the autophagy assays need to be improved. The data presented in Figure 1F and G do not sufficiently support the authors' claim that VHL WT, but not the mutant, suppresses autophagy flux. It appears that the autophagy flux based on LC3 blot was not significantly impaired in WT overexpressing cells (Figure 1F). Some reason, the basal expression level of LC3 looks much lower in VHL overexpressing cells. The LC3 blot for flux is even more ambiguous in Figure 1G. Figure 3J shows that autophagy flux (based on LC3 blots) was indeed reduced in PHD1 shRNA cells when comparing CQ plus and minus lanes. LC3 blots in Figure 3K are not acceptable, and the experiment needs to include flux analysis. In Figure 4L, the presentation of LC3 flux is not clear. Some key experiments, such as Figure 4M and EV1-L, lack flux analysis. The experiments, especially those presented in Figure 4M, require a more comprehensive analysis of autophagy.
4. The observed increase in LC3 puncta or PI3P production could be a result of autophagosome fusion blockage. It is highly recommended to analyze endogenous LC3 puncta in addition to the overexpressed construct, which is less reliable as an assay method. Moreover, the use of FYVE construct to analyze autophagy is questionable, as FYVE puncta may not necessarily reflect autophagosomes. To improve the clarity of the data, cell image data should include a broader view that encompasses multiple cells as representative images.
5. Figure 2 may suggest that VHL forms a distinct Beclin1-Vps34 complex free of Atg14L. It is unclear whether this is a novel complex present in the canonical pathway upon PHD1 activation. The nature of this complex needs to be further clarified, and whether it is a canonical part of the autophagy pathway present in mammalian cells needs to be clarified using one or two cell lines broadly used in autophagy research, such as HeLa, MEFs, and HCT116 cells. Additionally, to confirm the specific interaction of Figure 2A, it is recommended to perform the co-immunoprecipitation experiments using VHL KO cells for a negative control.
6. The PI3P *in vitro* assays throughout the presented figures may rely on the level of isolated Vps34, as well as the other components of the complex and VHL. It would be possible that the Vps34 complex containing VHL but not Atg14L could still be active, and the reason of the reduction of the activity in those assays could be simply because the amount of isolated Vps34 from Atg14L IP was lower.
7. Although the authors have investigated the mechanism using overexpression and knockout/knockdown models in some cancer cells, it is unclear whether the mechanism is physiologically relevant. To determine this, it is necessary to investigate whether glucose or amino acid starvation (or any physiologically relevant condition) can regulate the hydroxylation or VHL expression/interaction. A more relevant experiment for this issue might be to explore whether hypoxia or oxidative stress can regulate Vps34 activity and autophagy via the mechanism. Can hypoxia or oxidative stress enhance Vps34 activity and autophagy in a manner dependent upon the modification of Beclin 1?
8. The study has mainly investigated the disruption of the Atg14L-Beclin1 interaction to explain the reduction of autophagy by VHL. However, if the E3 ligase function is not required, as suggested by the data in Figure 1, it is not entirely clear how VHL simply acts to block Atg14L binding without utilizing its E3 ligase function. The authors need to provide further evidence to support the conclusion that VHL regulates autophagy without utilizing its E3 ligase function. This aspect has not been fully explored throughout the study, except in Figure 1. As pointed out above regarding the autophagy assays, the data in Figure 1 is not strong enough to draw such a conclusion about the role of E3 ligase activity.
9. Some key experiments are missing from the study to fully clarify the mechanism. Could PHD1-specific modification of Beclin 1 regulate the Vps34 activity and autophagy if VHL is depleted in SN12C cells or if VHL is expressed in 786-O cells? Is the Beclin

1 P54A mutant interaction with Atg14L still regulated by hypoxia and oxidative stress?

10. The authors will need to clarify whether the observed effects of modification and subsequent binding of VHL to Beclin 1 are specific to autophagy regulation or if they also affect other functions of Beclin 1, such as its involvement in apoptosis via interaction with Bcl-2. It is important to clarify whether the observed apoptotic changes and tumor growth are due to the role of Beclin 1 in apoptosis via Bcl-2 rather than via autophagy.

11. The tumor growth data in Figure 5 show a striking difference between WT and mutant cells. The presentation of the graph and the actual tumor sizes do not appear to match. To further support the analysis, the authors could test VHL knockdown in SN-12C cells along with WT and mutant Beclin 1 to validate the effects on tumor growth. Also, Figure 5B, C, F, G need P54A cells without VHL to exclude the possibility that the modification can affect apoptosis and tumor growth independently of VHL.

Minor:

12. It would be helpful to revise the presentation format of the western blots. Placing the labels on the side instead of at the bottom of each blot could improve the clarity and readability of the data.

13. To enhance the transparency of the data, it is recommended to present individual data points as dots in the bar graphs.

Referee #3:

In this manuscript the authors describe a novel role of VHL to inhibit nutrient deprivation-induced autophagy in tumor cells. First, they observe that VHL expression correlates with low autophagy activity. Re-expression of VHL can inhibit glucose (or other nutrients) deprivation-induced autophagy in VHL-deficient cells, which is independent on the function of VHL as an E3 ligase component and independent on HIF2a signaling. Then they find that VHL directly binds with Beclin1 and inhibits Beclin1-ATG14L interaction in a dose dependent manner. They show that VHL-Beclin1 interaction depends on hydroxylation on Beclin1 Pro54, which is catalyzed by PHD1 prolyl hydroxylase. Furthermore, they show that tumor cells bearing Beclin1 P54A mutant display stronger tumorigenesis ability and partially resistant to tumor suppression role of VHL.

Overall, the findings of this manuscript are interesting. The conclusions are supported by strong and clear biochemical and cell-based evidence from multiple cell lines (VHL WT or VHL null). There are a few limitations of the manuscript: VHL expression needs to be shown in several panels. The role of autophagy in ccRCC should be described clearly. The following are several suggestions that would improve the strength of the findings:

1. Fig 1A and 1B, although some patients harbor WT VHL, ccRCC may use other ways to downregulate VHL protein level (for example, silencing VHL by epigenetic pathway). So, VHL protein should be stained and blotted to show its protein level. This also applies to Fig 6A.
2. Fig EV1A, according to previous report (PMID: 24332042), p62 protein level is frequently increased in kidney cancer due to chromosome 5q copy number gains, especially in high grade tumor. However, the authors found that p62 protein level decreased in most tumors. This needs to be explained and discussed. Are most of the tumors examined low grade in this study?
3. Fig 2, which part of Beclin1 does ATG14L bind with? Do VHL and ATG14L bind to the same part of Beclin1?
4. PHD1 mainly localized in nucleus. However, Beclin1 is localized primarily in cytoplasmic structures like ER, Mt. Does Beclin1 localize in nucleus as well?
5. Fig 5A and 5B, although generally autophagy is critical for tumor cell survival, the function of autophagy in ccRCC, the connection of autophagy and cell death in ccRCC should be addressed.
6. Fig 5D, 5F and 5H. Since the authors showed that VHL inhibits nutrient stress-induced autophagy, is there any evidence suggested ccRCC tumor in mice undergoes nutrient stress?
7. Typo "VHL overexpression in VHL-deficient cells (Fig EV5H) or PHD1 depletion (Fig EV5I) or PHD1 P54A knock-in expression (Fig EV5J) in VHL-proficient ccRCC cells did not alter hypoxia-induced LC3B-II expression..." PHD1 P54A should be Beclin1 P54A.

Reviewer #1:

In this study, Wang et al. demonstrated a new role and mechanism of von Hippel-Lindau (VHL) as a tumor suppressor in regulating autophagy activity in sporadic clear-cell renal cell carcinoma cells. VHL is a component of the CUL2-RING E3 ligase and is known to bind and ubiquitinate HIF1/2. The authors found that VHL interacts with the autophagy protein Beclin1, which competitively inhibits Beclin1-ATG14 binding and the autophagy activity. Similar to VHL-HIF binding, VHL-Beclin1 binding is mediated by PHD1 and Beclin1 hydroxylation, at P54. VHL-Beclin1 binding is important to suppress autophagy and tumor growth in xenograft models. Overall, it is a well-designed study demonstrating a novel HIF-independent mechanism by which VHL regulates autophagy and suppresses tumor growth, and the significance is high. There are several specific questions that need to be addressed to further strengthen the conclusions.

Answer: We greatly appreciate the reviewer's acknowledgement of the potential significance of this report and the insightful comments, which are essential for improvement of this manuscript.

Specific comments:

1. The study relies mostly on overexpressed proteins. A key question to address is whether binding of VHL to Beclin1 leads to changes in endogenous Beclin1 protein level (via ubiquitination as to HIF). Ubiquitination experiments are needed to rule out that Beclin1 is a substrate of the VHL-CUL2-RING E3, using endogenous proteins.

Answer: We depleted VHL in ccRCC cells (Fig EV2D) and showed that VHL depletion did not alter the expression levels (Fig EV2D) and half-life (Fig EV2E) of Beclin1. Furthermore, neither VHL nor Cul2 depletion affected ubiquitylation (Fig EV2F and G) of Beclin1. These data, therefore, indicated that Beclin1 is not a substrate of the VHL-Cul2-RING E3 ubiquitin ligase.

Figure for referee with unpublished data and its description has been removed upon request by the authors.

2. Fig. 1, the condition of Glc+ (basal condition) with and without CQ should be added in Fig. 1G and Fig. EV1F, as based on Fig. 1F, the autophagy-inhibiting effect of VHL is more apparent under basal conditions (Glc+).

Answer: As suggested, the condition of Glc+ (basal condition) with and without CQ has been added in Fig. 1G and Fig. EV1G in the revised version. Furthermore, we repeated the indicated experiments in Fig. 1F and obtained more robust results.

Figure for referee with unpublished data and its description has been removed upon request by the authors.

3. Fig. 6, the Beclin1 P54-OH level and score in patient tumor samples should be normalized to Beclin1 protein expression levels, because high staining of Beclin1 P54-OH may be a reflection of high Beclin1 expression.

Answer: We have conducted the suggested normalization and revised the Figure 6.

Figure for referee with unpublished data and its description has been removed upon request by the authors.

4. An important premise of the study is that autophagy is tumor-promoting. To directly support it, Beclin1 KO xenografts should be included in Fig. 5 to test whether they form smaller tumors as a control.

Answer: As suggested, we depleted Beclin1 in ccRCC cells and showed that Beclin1 depletion blunted the tumor growth in mice (Fig EV10A-C), further supporting the tumor-promoting role of autophagy in the context of ccRCC.

Figure for referee with unpublished data and its description has been removed upon request by the authors.

5. CQ should be added to many panels to demonstrate the autophagy flux, such as Fig. 4H and EV1I-K.

Answer: We have added CQ in the suggested panels in Fig. 4M and EV2B, C, J in the current version.

Figure for referee with unpublished data and its description has been removed upon request by the authors.

6. A discussion should be included to reconcile the facts that Beclin1-mediated autophagy is considered tumor-promoting while Beclin1 also functions as a tumor suppressor.

Answer: We have included the discussion regarding tumor-promoting and tumor-suppressing functions of Beclin1-mediated autophagy (PMID: 36864290) in the revised manuscript.

7. Some conclusions in the texts are not accurate and need proofreading. For example, pg 11 "... Beclin1 P54A knock-in expression alleviated the autophagy-suppressing effects mediated by ectopically expressed VHL in 786-O or RCC4 cells (Fig 4J-M), indicating that Beclin1 P54 hydroxylation inhibit glucose deprivation induced autophagy in a VHL-dependent manner"; it should be "VHL inhibits ... autophagy in a P54-dependent manner".

Answer: We greatly appreciate the reviewer's carefully review of the manuscript. As suggested, we have revised the statement as "VHL inhibits glucose deprivation-induced autophagy in a Beclin1 P54 hydroxylation-dependent manner".

Referee #2:

The manuscript presents an interesting finding that VHL acts as a negative regulator of autophagy in renal carcinoma cells. The authors found that PHD1, whose activity is suppressed by hypoxia or oxidative stress, hydroxylates a proline residue in Beclin 1, a key regulator of autophagy. As a consequence, the modification enables VHL to recognize and bind to the modified Beclin 1. This binding disrupts the interaction between Beclin1 and Atg14L, an essential interaction for autophagy, resulting in autophagy suppression. The authors show that this role of VHL is important for its ability to suppress tumor growth.

Overall, the presented results are interesting and supportive of the proposed mechanism. If the mechanism is fully elucidated, it may define VHL as a novel negative regulator of autophagy. The link between PHD1/VHL and autophagy has a potential to advance our understanding of how cancer cells survive and grow under hypoxia conditions. The identification of the hydroxylation site of Beclin1 has been solidly demonstrated. Despite the recognized merits, there are multiple issues that need to be addressed for further consideration.

Answer: We greatly appreciate the reviewer's acknowledgement of the potential significance of this report and the insightful comments, which are essential for improvement of this manuscript.

1. The study has only investigated the regulation in specific renal cancer cell lines. Therefore, it is unclear whether the regulation is a general mechanism that broadly regulates the canonical autophagy pathway in cells or if it is specific to certain cell types.

Answer: We repeated the key experiments using HeLa human cervical cancer cells, HCT116 human colon cancer cells and MDA-MB-231 human breast cancer cells. We showed that depletion of PHD1 (Fig EV8A) or reconstituted expression of Beclin1 P54A (Fig EV8B and C) disrupted the binding of VHL to Beclin1 with correspondingly increased interaction between ATG14L and Beclin1 in these tumor cells. In addition, compared to expression of WT Beclin1, Beclin1 P54A expression enhanced glucose deprivation-induced autophagy (Fig EV8D). Consistently, overexpression of Flag-VHL suppressed glucose deprivation-induced autophagy in these cells expressing WT Beclin1, but not Beclin1 P54A (Fig EV8E). Thus, PHD1-dependent Beclin1 P54 hydroxylation and VHL-suppressed autophagy is not tumor type-specific.

Figure for referee with unpublished data and its description has been removed upon request by the authors.

2. It is unclear why the authors tested glucose starvation instead of amino acid starvation. There is controversy surrounding whether glucose starvation can robustly induce autophagy. Even if it can induce autophagy in some cells, it is generally considered to be less effective than amino acid starvation. The presented data on autophagy assays after glucose starvation is unconvincing. The

starvation effects on autophagy flux in wild type cells are not drastic, and the effects of overexpression/knockout/knockdown are not strong throughout the presented data.

Answer: We repeated the key experiments under amino acid starvation conditions. We showed that reconstituted expression of Beclin1 P54A could increase amino acid starvation-induced autophagy compared to its WT counterpart in VHL-proficient ccRCC cells (Fig EV7I). Additionally, expression of Flag-VHL in the VHL-deficient tumor cells suppressed amino acid starvation-induced autophagy in the tumor cells expressing WT Beclin1, but not Beclin1 P54A (Fig EV7J). These results suggested that Beclin1 P54 hydroxylation plays a critical role in regulation of amino acid- and glucose starvation-induced autophagy.

Figure for referee with unpublished data and its description has been removed upon request by the authors.

3. Throughout the presented figures, the autophagy assays need to be improved. The data presented in Figure 1F and G do not sufficiently support the authors' claim that VHL WT, but not the mutant, suppresses autophagy flux. It appears that the autophagy flux based on LC3 blot was not significantly impaired in WT overexpressing cells (Figure 1F). Some reason, the basal expression level of LC3 looks much lower in VHL overexpressing cells. The LC3 blot for flux is even more ambiguous in Figure 1G. Figure 3J shows that autophagy flux (based on LC3 blots) was indeed reduced in PHD1 shRNA cells when comparing CQ plus and minus lanes. LC3 blots in Figure 3K are not acceptable, and the experiment needs to include flux analysis. In Figure 4L, the presentation

of LC3 flux is not clear. Some key experiments, such as Figure 4M and EV1-L, lack flux analysis. The experiments, especially those presented in Figure 4M, require a more comprehensive analysis of autophagy.

Answer: Based on the reviewer's suggestion, we repeated the indicated experiments with improved autophagy assays and obtained more robust results.

The revised Figure 1F showed that ectopical expression of both Flag-WT VHL and VHL C162F (E3 ligase activity-dead mutant) equivalently inhibited glucose deprivation-induced autophagy in VHL-deficient ccRCC cells. The revised Figure 1G showed that the enhanced autophagy in the VHL-depleted SN-12C cells was abrogated by reconstituted expression of both WT VHL and VHL C162F E3 ligase activity-dead mutant. These results indicated that VHL inhibits autophagy initiation independent of its E3 ligase activity.

The revised Figure 3K (i.e., Figure 3J in the previous version) with flux analysis shows that depletion of PHD1 in VHL-proficient SN-12C or TK-10 cells enhanced autophagy initiation.

The revised Figure 3L (i.e., Figure 3K in the previous version) and Figure 4I presents a clearer LC3 flux.

The revised Figure 4M and EV2J (i.e., Figure EV1L in the previous version) shows the results with flux analysis.

In addition, more comprehensive flux analysis of autophagy showed that Beclin1 P54A knock-in expression or reconstitution further enhanced glucose deprivation-induced VPS34 activity (Fig EV6E, first four lanes) and autophagy initiation (Fig EV6F, first four lanes) in VHL-proficient SN-12C or TK-10 cells, but not in VHL-deficient 786-O or RCC-4 cells (Fig 4J and Reviewer Only Figure 1A, first four lanes). Of note, we showed that Beclin1 P54A expression lost its regulation of the VPS34 activity and autophagy induction in VHL-depleted SN-12C or TK-10 cells (Fig EV6E and F, last four lanes). On the contrary, Beclin1 P54A expression gained its regulation of the VPS34 activity and autophagy induction in VHL-expressed 786-O or RCC-4 cells (Fig 4J and Reviewer Only Figure 1A, last four lanes). These data, therefore, indicated that Beclin1 P54 hydroxylation inhibit glucose deprivation-induced autophagy in a VHL-dependent manner.

Figure for referee with unpublished data and its description has been removed upon request by the authors.

4. The observed increase in LC3 puncta or PI3P production could be a result of autophagosome fusion blockage. It is highly recommended to analyze endogenous LC3 puncta in addition to the overexpressed construct, which is less reliable as an assay method. Moreover, the use of FYVE

construct to analyze autophagy is questionable, as FYVE puncta may not necessarily reflect autophagosomes. To improve the clarity of the data, cell image data should include a broader view that encompasses multiple cells as representative images.

Answer: As suggested, endogenous LC3 puncta was examined. To avoid the confusion, we removed the FYVE puncta results from the manuscript. In addition, endogenous LC3 puncta in a broader view that encompasses multiple cells are shown as representative images (Figs 1E, 3D, 3J, 4H, 4L, EV4K and EV6C).

Figure for referee with unpublished data and its description has been removed upon request by the authors.

(Fig 4L) Parental 786-O cells and the indicated clones with knock-in expression of Beclin1 P54A were transfected with HA-VHL and treated with glucose deprivation for 2 h. Representative images of endogenous LC3B puncta are shown. The numbers of LC3B puncta were quantitated.

(Fig EV4K) PHD1 depleted SN-12C cells with reconstituted expression of the indicated shRNA-resistant Flag-rPHD1 proteins were treated with glucose deprivation for 2 h. Representative images of endogenous LC3B puncta are shown. The numbers of LC3B puncta were quantitated.

(Fig EV6C) Parental 786-O cells and the indicated clones with knock-in expression of Beclin1 P54A were treated with glucose deprivation for 2 h. Representative images of endogenous LC3B puncta are shown. The numbers of LC3B puncta were quantitated. Data represent the mean \pm SD. *** $P < 0.001$ (two-tailed t-test). NS, no significance. All experiments were repeated three times independently with similar results.

5. Figure 2 may suggest that VHL forms a distinct Beclin1-Vps34 complex free of Atg14L. It is unclear whether this is a novel complex present in the canonical pathway upon PHD1 activation. The nature of this complex needs to be further clarified, and whether it is a canonical part of the autophagy pathway present in mammalian cells needs to be clarified using one or two cell lines broadly used in autophagy research, such as HeLa, MEFs, and HCT116 cells. Additionally, to confirm the specific interaction of Figure 2A, it is recommended to perform the co-immunoprecipitation experiments using VHL KO cells for a negative control.

Answer: We performed immunoprecipitation Beclin1 from WT PHD1- and PHD1 D311A (PHD1 inactive mutant)-expressing SN-12C cells and found that Beclin1 interacted with canonical autophagy-regulatory components, including Beclin1, VPS34, VPS15, and UVRAG in both cell lines. In addition, the association of Beclin1 with VHL was only detected in WT PHD1-expressing tumor cells (Fig 3G). These results suggested that VHL forms a novel complex (which does not contain ATG14L, as shown in Figure 2A) present in the canonical pathway upon PHD1 activation.

In addition, we repeated the key experiments using HCT116 human colon cancer cells, MDA-MB-231 human breast cancer cells, and HeLa human cervical cancer cells and obtained consistent results (Fig EV8A-E).

To confirm the specific protein interaction in Figure 2A, we used the suggested VHL KO cells as a negative control.

Figure for referee with unpublished data and its description has been removed upon request by the authors.

6. The PI3P in vitro assays throughout the presented figures may rely on the level of isolated Vps34, as well as the other components of the complex and VHL. It would be possible that the Vps34 complex containing VHL but not Atg14L could still be active, and the reason of the reduction of

the activity in those assays could be simply because the amount of isolated Vps34 from Atg14L IP was lower.

Answer: This point is well taken. We normalized the amount of immunoprecipitated Vps34 from each condition and showed that the activity of VHL-associated VPS34 complex was abolished compared to that of ATG14L-associated VPS34 complex although VPS34 levels were comparable across all the samples (Fig EV2L), further supporting the inhibitory effect of VHL on VPS34-dependent PI(3)P production and subsequent autophagy initiation.

Figure for referee with unpublished data and its description has been removed upon request by the authors.

7. Although the authors have investigated the mechanism using overexpression and knockout/knockdown models in some cancer cells, it is unclear whether the mechanism is physiologically relevant. To determine this, it is necessary to investigate whether glucose or amino acid starvation (or any physiologically relevant condition) can regulate the hydroxylation or VHL expression/interaction. A more relevant experiment for this issue might be to explore whether hypoxia or oxidative stress can regulate Vps34 activity and autophagy via the mechanism. Can hypoxia or oxidative stress enhance Vps34 activity and autophagy in a manner dependent upon the modification of Beclin 1?

Answer: PHD activity is active under normoxic but inactive under hypoxic conditions. We showed that glucose or amino acid starvation does not alter Beclin1 P54 hydroxylation or the interaction between Beclin1 and VHL (Fig EV7C and D) under normoxic conditions. However, under hypoxic conditions, Beclin1 P54 hydroxylation was abolished due to inactivation of PHD (Fig EV7A), and the cells expressing WT Beclin1 and Beclin1 P54A exhibited no difference in VPS34 activity (Fig EV7E) and autophagy induction (Fig EV7F) in VHL-proficient SN-12C cells. In addition, overexpression of VHL exhibited non-differential effect on VPS34 activity (Fig EV7G) and autophagy induction (Fig EV7H) in the 786-O cells expressing WT Beclin1 and Beclin1 P54A under hypoxic conditions. These results suggested that Beclin1 P54 hydroxylation is regulated under physiological conditions and is inhibited by hypoxia, leading to diminished autophagy inhibition by VHL under low oxygen conditions.

Figure for referee with unpublished data and its description has been removed upon request by the authors.

8. The study has mainly investigated the disruption of the Atg14L-Beclin1 interaction to explain the reduction of autophagy by VHL. However, if the E3 ligase function is not required, as suggested by the data in Figure 1, it is not entirely clear how VHL simply acts to block Atg14L binding without utilizing its E3 ligase function. The authors need to provide further evidence to support the conclusion that VHL regulates autophagy without utilizing its E3 ligase function. This aspect has not been fully explored throughout the study, except in Figure 1. As pointed out above regarding the autophagy assays, the data in Figure 1 is not strong enough to draw such a conclusion about the role of E3 ligase activity.

Answer: We showed that WT VHL and E3 ligase-inactive VHL C162F exhibited similar effect on autophagy regulation (Fig 1F and G). In addition, depletion of Cul2, a component of VHL E3 ligase complex, did not affect VHL overexpression-suppressed binding of ATG14L to Beclin1 (Fig 2J), VPS34 activity (Fig EV2H), and autophagy (Fig EV2I) in VHL-deficient 786-O cells. Consistently, glucose starvation-induced autophagy was not altered upon Cul2 depletion in VHL-proficient SN-12C and TK-10 cells (Fig EV2C). These data, therefore, suggested that VHL regulates autophagy without utilizing its E3 ligase function.

Figure for referee with unpublished data and its description has been removed upon request by the authors.

9. Some key experiments are missing from the study to fully clarify the mechanism. Could PHD1-specific modification of Beclin1 regulate the Vps34 activity and autophagy if VHL is depleted in SN-12C cells or if VHL is expressed in 786-O cells? Is the Beclin1 P54A mutant interaction with Atg14L still regulated by hypoxia and oxidative stress?

Answer: This point is well taken. As suggested, we performed more comprehensive flux analysis of autophagy showed that Beclin1 P54A knock-in expression or reconstitution further enhanced glucose deprivation-induced VPS34 activity (Fig EV6E, first four lanes) and autophagy initiation (Fig EV6F, first four lanes) in VHL-proficient SN-12C or TK-10 cells, but not in VHL-deficient 786-O or RCC-4 cells (Fig 4J and Reviewer Only Figure 1A, first four lanes). Of note, we showed that Beclin1 P54A expression lost its regulation of the VPS34 activity and autophagy induction in VHL-depleted SN-12C or TK-10 cells (Fig EV6E and F, last four lanes). On the contrary, Beclin1 P54A expression gained its regulation of the VPS34 activity and autophagy induction in VHL-expressed 786-O or RCC-4 cells (Fig 4J and Reviewer Only Figure 1A, last four lanes). These data, therefore, indicated that Beclin1 P54 hydroxylation inhibit glucose deprivation-induced autophagy in a VHL-dependent manner.

Under normoxic conditions, we found that Beclin1 P54A exhibited the dramatically enhanced interaction with ATG14L and reduced interaction with VHL compared to its WT counterpart in VHL-proficient SN-12C cells (Fig EV7B, compare lane 1 and lane 3) or VHL-ectopically expressed 786-O cells (Reviewer Only Figure 1B, compare lane 5 and lane 7).

In contrast, both WT Beclin1 and Beclin1 P54A comparably bound to ATG14L under hypoxic conditions, which diminished Beclin1 P54 hydroxylation and abolished the binding of VHL to WT Beclin1 in VHL-proficient SN-12C cells (Fig EV7B, compare lane 2 and lane 4) or VHL-ectopically expressed 786-O cells (Reviewer Only Figure 1B, compare lane 6 and lane 8).

Figure for referee with unpublished data and its description has been removed upon request by the authors.

10. The authors will need to clarify whether the observed effects of modification and subsequent binding of VHL to Beclin1 are specific to autophagy regulation or if they also affect other functions of Beclin 1, such as its involvement in apoptosis via interaction with Bcl-2. It is important to clarify whether the observed apoptotic changes and tumor growth are due to the role of Beclin1 in apoptosis via Bcl-2 rather than via autophagy.

Answer: We showed that both WT Beclin1 and Beclin1 P54A interacted with BCL2, and these bindings were all disrupted in glucose-starved tumor cells (Fig EV9D and E). Consistently, both WT Beclin1 and Beclin1 P54A comparably interacted with BCL2 in tumor tissues (Fig EV10F and I). These results suggested that Beclin1 P54 hydroxylation does not affect the interaction between Beclin1 and BCL2 as well as the pro-survival function of BCL2.

In addition, the suppressive role of Beclin1 P54A on cell death upon glucose deprivation was eliminated by ATG7 depletion in VHL proficient SN-12C cells (Fig EV9F and G) or VHL-transfected 786-O cells (Fig EV9H and I), suggesting that Beclin1 P54 hydroxylation-increased cell death is independent on the interaction between Beclin1 and BCL2.

Figure for referee with unpublished data and its description has been removed upon request by the authors.

11. The tumor growth data in Figure 5 show a striking difference between WT and mutant cells. The presentation of the graph and the actual tumor sizes do not appear to match. To further support the analysis, the authors could test VHL knockdown in SN-12C cells along with WT and mutant Beclin1 to validate the effects on tumor growth. Also, Figure 5B, C, F, G need P54A cells without VHL to exclude the possibility that the modification can affect apoptosis and tumor growth independently of VHL.

Answer: We repeated the experiments with appropriated controls. The graphs match the actual tumor sizes (Fig 5D and F).

We showed that Beclin1 P54A knock-in expression in VHL-proficient SN-12C cells increased tumor growth (Fig 5D and EV10D). Consistent with the tumor suppressor role of VHL, VHL depletion increased tumor growth (Fig 5D and EV10D). Notably, the tumor-promoting effect induced by Beclin1 P54A expression was eliminated upon VHL depletion (Fig 5D and EV10D).

Furthermore, the following experimental settings were designed to exclude the possibility that the modification can affect tumor growth and apoptosis independently of VHL.

As expected, restored VHL expression in VHL-deficient 786-O cells reduced tumor growth (Fig 5F and EV10G), and LC3B-II, with a corresponding increase in p62 expression (Fig 5G) in tumor tissue, and these changes were substantially abolished by Beclin1 P54A expression (Figs 5F, G, and EV10G). In contrast, the Beclin1 P54A-elicited effects were diminished in the VHL-deficient tumors (Fig 5F, G, and EV10G).

As shown in revised Figure 5B, restoration of VHL expression in 786-O and RCC4 cells increased glucose deprivation-induced cell apoptosis (Fig 5B), and this increase was largely inhibited by Beclin1 P54A knock-in expression, indicating that Beclin1 P54 hydroxylation is required for VHL-promoted cell apoptosis under nutrient stress conditions. In addition, Beclin1 P54A-expressing and VHL-deficient 786-O and RCC-4 cells exhibited no difference in tumor cell

death (Fig 5B), indicating that Beclin1 P54A suppresses glucose deprivation-induced cell apoptosis in a VHL-dependent manner.

Figure for referee with unpublished data and its description has been removed upon request by the authors.

Minor:

12. It would be helpful to revise the presentation format of the western blots. Placing the labels on the side instead of at the bottom of each blot could improve the clarity and readability of the data.

Answer: As reviewer suggested, we have revised the presentation format of the western blots.

13. To enhance the transparency of the data, it is recommended to present individual data points as dots in the bar graphs.

Answer: We have revised the presentation format of the data according to the suggestion.

Referee #3:

In this manuscript the authors describe a novel role of VHL to inhibit nutrient deprivation-induced autophagy in tumor cells. First, they observe that VHL expression correlates with low autophagy activity. Re-expression of VHL can inhibit glucose (or other nutrients) deprivation-induced autophagy in VHL-deficient cells, which is independent on the function of VHL as an E3 ligase component and independent on HIF2a signaling. Then they find that VHL directly binds with Beclin1 and inhibits Beclin1-ATG14L interaction in a dose dependent manner. They show that VHL-Beclin1 interaction depends on hydroxylation on Beclin1 Pro54, which is catalyzed by PHD1 prolyl hydroxylase. Furthermore, they show that tumor cells bearing Beclin1 P54A mutant display stronger tumorigenesis ability and partially resistant to tumor suppression role of VHL.

Overall, the findings of this manuscript are interesting. The conclusions are supported by strong and clear biochemical and cell-based evidence from multiple cell lines (VHL WT or VHL null). There are a few limitations of the manuscript: VHL expression needs to be shown in several panels. The role of autophagy in ccRCC should be described clearly. The following are several suggestions that would improve the strength of the findings:

Answer: We greatly appreciate the reviewer's acknowledgement of the potential significance of this report and the insightful comments, which are essential for improvement of this manuscript. We have included the requested VHL expression in immunoblotting analyses and discussed the role of autophagy in ccRCC.

1. Fig 1A and 1B, although some patients harbor WT VHL, ccRCC may use other ways to downregulate VHL protein level (for example, silencing VHL by epigenetic pathway). So, VHL protein should be stained and blotted to show its protein level. This also applies to Fig 6A.

Answer: We have included the VHL expression levels in Fig 1A, 1B, and 6A.

Figure for referee with unpublished data and its description has been removed upon request by the authors.

2. Fig EV1A, according to previous report (PMID: 24332042), p62 protein level is frequently increased in kidney cancer due to chromosome 5q copy number gains, especially in high grade tumor. However, the authors found that p62 protein level decreased in most tumors. This needs to be explained and discussed. Are most of the tumors examined low grade in this study?

Answer: The previous report (PMID: 24332042) examined the expression of p62 levels in ccRCC cell lines and in an immortalized normal renal epithelial cell line under nonenergy stress conditions and revealed increased expression of p62 in tumor cells. This report did not examine the expression of p62 in human ccRCC tissues, which have dynamic tumor microenvironmental inputs, including the stresses that cause autophagy, which are missed in cell-cultured conditions.

To further address the reviewer's question, we performed IHC analyses of p62 in human ccRCC specimens and their adjacent normal tissues. We showed that both p62 expression levels (Fig EV1A) and mRNA levels of *SQSTM1* (encoding p62) (Fig EV1C) are higher in human ccRCC specimens containing WT VHL than in their adjacent normal tissues. Notably, p62 protein expression, but not mRNA levels of *SQSTM1*, was decreased in human ccRCC specimens containing deficient VHL expression compared to those expressing WT VHL (Fig EV1A and C).

p62 protein expression is dynamically controlled by multi-layers' regulations including transcriptional and posttranslational regulations. Given that p62 protein is essential for autophagy by recruiting ubiquitylated proteins into autophagosomes and is degraded together with the ubiquitylated proteins in lysosome, our results suggested that transcriptionally elevated p62 protein expression can have a quick turn over in tumor cells, which have highly activated autophagy. Thus, our results, which showed decreased p62 expression and increased LC3B in tumor specimens containing VHL deficiency, suggested that ccRCC often have high levels of autophagy due to decreased VHL-mediated inhibition.

Figure for referee with unpublished data and its description has been removed upon request by the authors.

3. Fig 2, which part of Beclin1 does ATG14L bind with? Do VHL and ATG14L bind to the same part of Beclin1?

Answer: ATG14L is known to bind to Beclin1 CCD domain (PMID: 21311563). We constructed various truncation mutants of Beclin1 and showed that Beclin1 with CCD domain deletion lost the binding of Beclin1 to both ATG14L and VHL (Fig 2D). These results suggested that VHL and ATG14L competitively bind to the same region of Beclin1.

Figure for referee with unpublished data and its description has been removed upon request by the authors.

4. PHD1 mainly localized in nucleus. However, Beclin1 is localized primarily in cytoplasmic structures like ER, Mt. Does Beclin1 localize in nucleus as well?

Answer: Immunofluorescence staining (Fig EV5B) and cell fractionation analyses (Fig EV5C) showed that PHD1 was also localized in the cytosol of ccRCC cells, where Beclin1 localizes. As shown in these figures, Beclin1 almost did not localized in the nucleus (Fig EV5B and C). As expected, coimmunoprecipitation analyses showed that Beclin1 interacted with PHD in the cytosol of the tumor cells (Fig EV5D).

Figure for referee with unpublished data and its description has been removed upon request by the authors.

5. Fig 5A and 5B, although generally autophagy is critical for tumor cell survival, the function of autophagy in ccRCC, the connection of autophagy and cell death in ccRCC should be addressed.

Answer: This point is well taken. We depleted ATG7, which is essential for autophagosome formation and autophagy occurrence in ccRCC cells (Fig EV9F-I) and found that ATG7 depletion eliminated the effect of Beclin1 P54A expression on autophagy regulation (Fig EV9F and H).

In addition, Beclin1 P54A expression-suppressed cell death upon glucose deprivation was eliminated by ATG7 depletion in VHL proficient SN-12C cells (Fig EV9F and G) or VHL-transfected 786-O cells (Fig EV9H and I).

Figure for referee with unpublished data and its description has been removed upon request by the authors.

levels are higher in mouse tumor than in their adjacent normal tissues (Reviewer Only Figure 2), suggesting that ccRCC tumor in mice undergoes nutrient stress.

Figure for referee with unpublished data and its description has been removed upon request by the authors.

7. Typo "VHL overexpression in VHL-deficient cells (Fig EV5H) or PHD1 depletion (Fig EV5I) or PHD1 P54A knock-in expression (Fig EV5J) in VHL-proficient ccRCC cells did not alter hypoxia-induced LC3B-II expression..." PHD1 P54A should be Beclin1 P54A.

Answer: We wholeheartedly appreciate the reviewer's meticulous and precise assessment of our manuscript. The indicated typo was corrected.

Dear Zhimin,

We have now received re-review reports from all referees which I paste below. As you will see, you have addressed the concerns of referees 1 and 3 satisfactorily. Referee 2 has, however, replied with a list of requested clarifications. These centre on the consequences of proline hydroxylation in this physiological context. I believe these points can be addressed adequately by expanding the discussion on the irreversibility of this modification. Please detail the exact experimental procedures followed as requested by referee 2. There are also some remaining editorial points which need to be addressed. In this regard would you please:

- include funding information on our online submission system for the following grants: the National Natural Science Foundation of China (82188102, 82030074, Z.L.; 82072630, 92157113, D.X.; 82372814, 82173114, Z.W.; 82002811, M.Y.; 82103658, Q.Z.), Shanghai Pujiang Program (No.2022PJD040, Q.Z.), Natural Science Foundation of Heilongjiang Province of China (LH2023H095, B.L.), The Project of Beijing Medical Award Foundation (YXJL-2021-0581-0484, B.L.), the Zhejiang Natural Science Foundation Key Project (LD22H160002, D.X.; LD21H160003, Z.L.), and Zhejiang Natural Science Foundation Discovery Project (LQ22H160023, Z.W.),
- alter the reference format to 10 authors + et al.,
- rename the Conflict of Interest section the "DISCLOSURE AND COMPETING INTERESTS STATEMENT",
- remove the author credit section from the manuscript,
- there are no Source Data (SD) files attached to the manuscript. Please supply files according to the uploaded SD checklist,
- a separate 'Data Information' section is required in the legends of all figures except for figure EV10,
- indicate the statistical test used for data analysis in the legends of figures EV2e; EV6b-c; EV7e and g,
- in the legends of figures EV5c and EV10b and c, although 'n' is provided, please describe the nature of entity for 'n',
- there are ten EV figures, please rearrange your manuscript so there are only five. The EV figure legends should not be in a separate file, but included in the main ms file, after the main figure legends. I have copied EV figure legends to the main ms file, and
- correct the order of the sections as indicated in our Guide to Authors.

We include a synopsis of the paper (see <http://emboj.embopress.org/>). Please provide me with a two-sentence general summary statement and 3-5 bullet points that capture the key findings of the paper.

EMBO Press is an editorially independent publishing platform for the development of EMBO scientific publications.

Best wishes,

William

William Teale, PhD
Editor
The EMBO Journal
w.teale@embojournal.org

- a point-by-point response to the referees' comments, with a detailed description of the changes made (as a word file).
 - a word file of the manuscript text.
 - individual production quality figure files (one file per figure)
 - a complete author checklist, which you can download from our author guidelines (<https://www.embopress.org/page/journal/14602075/authorguide>).
 - Expanded View files (replacing Supplementary Information)
- Please see out instructions to authors

We realize that it is difficult to revise to a specific deadline. In the interest of protecting the conceptual advance provided by the work, we recommend a revision within 3 months (14th Jan 2024). Please discuss the revision progress ahead of this time with the editor if you require more time to complete the revisions.

Referee #1:

The authors have satisfactorily addressed all the questions.

Referee #2:

The revision has addressed some issues but requires further clarification. The physiological relevance of the modification still remains ambiguous. While the data show the modification occurs in normal conditions (normoxia, PHD1 active state), it results in an inactive complex that cannot mediate autophagy. Since proline hydroxylation is irreversible, this means that the VHL-Beclin1 complex is a dead-end product. It is unclear why cells make this complex that is not subject to any regulation. The only way for cells to form the active functional Beclin1 complex under hypoxia or when PHD1 is inactivated is through de novo synthesis of Beclin1. Thus, the proposed mechanism does not entirely make sense and needs further reconciliation. The authors need to explain how actually the modification acts in physiological contexts, going beyond the comparison of WT and mutant cells. The following numbering corresponds to the previous comments.

1. In the new data, control cells maintain VHL-Beclin1 interaction in normal conditions, suggesting that the inactive VHL-Beclin1 complex is predominant in normal physiology. The purpose to make this unregulated, irreversible complex is unclear. Additionally, it is necessary to specify whether the LC3B band corresponds to LC3B I or LC3B II. Quantifying the flux is also necessary. Justification or prior references for such a short term 2-hour glucose starvation condition for flux assay are needed. Figure EV8E should include flux analysis (the figure legend says it analyzed flux but the actual data does not reflect it) and assessment of VHL expression. The choice of Beclin1 antibody over the flag antibody in some experiments needs to be explained.

2. It is challenging to believe that increasing Atg14-Beclin1 interaction alone is enough to enhance autophagy flux, which requires multiple coordinated events. The 1-hour treatment for flux analysis is unusual; a reference or rationale for this condition is needed. The role of Beclin1 modification in 786-O or RCC4 cells without VHL expression is obscure.

3. Figure 3L needs a longer exposure for LC3B blot. Figures EV6E and 4J should include Beclin1 and Vps34 amounts. The effect of glucose starvation on Vps34 activity lacks substantial evidence in literature, apart from one paper (PMID: 23332761). Including a positive control (e.g. amino acid starvation) would help clarify the significance of the observed change. Otherwise, this Vps34 assay might not be critical if the activity simply reflects the amount of Vps34 isolated from the immunoprecipitation.

4. The LC3B staining pattern needs clarification. Detailed information on how LC3B dots were identified and counted is required, as some groups of cells are very bright, while others are overall too weak but still form puncta. The drastic variation in overall LC3B intensity among the stained cells between groups raises concerns. Overall, the image quality is poor and may not be acceptable for top-tier journals. The method section lacks necessary details. To confirm that the LC3B dot accumulation is not due to the blockage of autophagosome maturation, representative experiments using chloroquine or bafilomycin A1 are needed.

5. Fig 3G lacks Atg14L analysis, which should be included. If the VHL-Atg14L interaction is the predominant form in cells under normal conditions, its role in canonical autophagy regulation is doubtful. Is this modification merely increasing the inactive complex by recruiting VHL? Since proline hydroxylation is irreversible, cells need de novo synthesis of Beclin1 to make the functional complex. Thus, it is obscure what is the purpose of making this inactive complex in normal conditions (normoxia, PHD1 active state). Also, whether the label "VHL WT" in Fig 2A indicates VHL overexpression needs to be clarified.

6. Fig EV2L presents clear results. However, it is essential to include the amounts of Vps34 and relevant molecules in the samples used for the assay in the specific experiments shown in the original figures. Showing only the amount of Atg14L is insufficient for interpreting the results. Please check the comment #3 above.

7. The physiological relevance of the findings (what the modification actually does) still needs more clarification. The irreversible nature of the modification raises questions about the observed reduction in interaction during hypoxia. Does the result suggest that the modified Beclin1 is degraded while the unmodified version of Beclin1 is being de novo synthesized during 8h of hypoxia, a condition that typically suppresses overall protein synthesis? Additionally, the reported increase in Vps34 activity under glucose starvation is not entirely supported by previous literature, and the assay conditions should be scrutinized, with the inclusion of necessary proteins and a positive control for validation. Please check the comment #3 above related to this issue.

8. The role of VHL in the proposed mechanism is still not clearly understandable. If VHL serves as a negative regulator, suppressing autophagy under normal conditions (normoxia, PHD1 active state), what are the implications of this role for cellular responses in normal wild type cells to regulate autophagy? Does VHL simply maintain the inactive complex without being subject to any regulation?

9. The irreversibility of the modification raises questions about its regulatory role, given that it results in a dead-end product according to the proposed mechanism. The purpose of forming such an inactive, dead-end product in cells remains not entirely clarified.

Other points:

The manuscript has numerous issues related to the method section and figure legends. It requires more comprehensive details for each experiment and figure, including incubation times, buffer compositions, and other key details. For instance, more information is needed regarding how the amino acid starvation medium was prepared, the microscopy analysis of LC3, the in vitro PI(3)P assay, and the quantification of autophagy assays. The term "autophagy induction" is used improperly in some instances. The presented assays are insufficient for evaluating autophagy induction.

Referee #3:

The revised manuscript addressed all of previous critiques and now is acceptable for publication.

Referee #2:

The revision has addressed some issues but requires further clarification. The physiological relevance of the modification still remains ambiguous. While the data show the modification occurs in normal conditions (normoxia, PHD1 active state), it results in an inactive complex that cannot mediate autophagy. Since proline hydroxylation is irreversible, this means that the VHL-Beclin1 complex is a dead-end product. It is unclear why cells make this complex that is not subject to any regulation. The only way for cells to form the active functional Beclin1 complex under hypoxia or when PHD1 is inactivated is through de novo synthesis of Beclin1. Thus, the proposed mechanism does not entirely make sense and needs further reconciliation. The authors need to explain how actually the modification acts in physiological contexts, going beyond the comparison of WT and mutant cells. The following numbering corresponds to the previous comments.

Answer: Dehydroxylation does occur in cells or biological and physiological processes in animals (PMID: 35325017, 35154445, 35868525); therefore, proline hydroxylation is not irreversible. We have expanded the discussion on the irreversibility of hydroxylation in the revised manuscript (highlighted in the manuscript). We will extend our studies to determine the dynamic regulation of PHD1-mediated Beclin1 P54 hydroxylation in the context of tumor progression, which is beyond the scope of the current report.

1. In the new data, control cells maintain VHL-Beclin1 interaction in normal conditions, suggesting that the inactive VHL-Beclin1 complex is predominant in normal physiology. The purpose to make this unregulated, irreversible complex is unclear. Additionally, it is necessary to specify whether the LC3B band corresponds to LC3B I or LC3B II. Quantifying the flux is also necessary. Justification or prior references for such a short term 2-hour glucose starvation condition for flux assay are needed. Figure EV8E should include flux analysis (the figure legend says it analyzed flux but the actual data does not reflect it) and assessment of VHL expression. The choice of Beclin1 antibody over the flag antibody in some experiments needs to be explained.

Answer: As described in the aforementioned discussion, dehydroxylation and hydroxylation can be dynamically regulated depending on the signaling input.

The alteration of the LC3B band indicates autophagy levels.

We did not include include flux analysis in Figure EV8E and have revised the figure legend by correcting the mistakes.

The choice of the Beclin1 antibody over the flag antibody in some experiments is made to assess the endogenous Beclin1 expression.

2. It is challenging to believe that increasing Atg14-Beclin1 interaction alone is enough to enhance autophagy flux, which requires multiple coordinated events. The 1-hour treatment for flux analysis is unusual; a reference or rationale for this condition is needed. The role of Beclin1 modification in 786-O or RCC4 cells without VHL expression is obscure.

Answer: References are provided (PMID: 24013218; 24980960).

3. Figure 3L needs a longer exposure for LC3B blot. Figures EV6E and 4J should include Beclin1 and Vps34 amounts. The effect of glucose starvation on Vps34 activity lacks substantial evidence in literature, apart from one paper (PMID: 23332761). Including a positive control (e.g. amino acid starvation) would help clarify the significance of the observed change. Otherwise, this Vps34 assay might not be critical if the activity simply reflects the amount of Vps34 isolated from the immunoprecipitation.

Answer: A longer exposure for LC3B blot is provide for Figure 3L.

4. The LC3B staining pattern needs clarification. Detailed information on how LC3B dots were identified and counted is required, as some groups of cells are very bright, while others are overall too weak but still form puncta. The drastic variation in overall LC3B intensity among the stained cells between groups raises concerns. Overall, the image quality is poor and may not be acceptable for top-tier journals. The method section lacks necessary details. To confirm that the LC3B dot accumulation is not due to the blockage of autophagosome maturation, representative experiments using chloroquine or bafilomycin A1 are needed.

Answer: Detailed information on how LC3B dots were counted is provided. The method section is enriched with necessary details. The fluorescence brightness variation between groups were introduced by fluorescence from green-fluorescent dye of endogenous LC3B or fluorescence from Green Fluorescent Protein (GFP) of exogenous GFP-LC3B.

5. Fig 3G lacks Atg14L analysis, which should be included. If the VHL-Atg14L interaction is the predominant form in cells under normal conditions, its role in canonical autophagy regulation is doubtful. Is this modification merely increasing the inactive complex by recruiting VHL? Since proline hydroxylation is irreversible, cells need de novo synthesis of Beclin1 to make the functional complex. Thus, it is obscure what is the purpose of making this inactive complex in normal conditions (normoxia, PHD1 active state). Also, whether the label "VHL WT" in Fig 2A indicates VHL overexpression needs to be clarified.

Answer: ATG14L is included in Fig 3G. As described in the aforementioned discussion, dehydroxylation and hydroxylation can be dynamically regulated depending on the signaling input. The label "VHL WT" in Fig 2A indicates that the cells expressed endogenous VHL, which were compared to the cells with VHL knockout. We have relabeled the figure.

6. Fig EV2L presents clear results. However, it is essential to include the amounts of Vps34 and relevant molecules in the samples used for the assay in the specific experiments shown in the original figures. Showing only the amount of Atg14L is insufficient for interpreting the results. Please check the comment #3 above.

Answer: ATG14L expression is used as a control. Additional controls seem unnecessary.

7. The physiological relevance of the findings (what the modification actually does) still needs more clarification. The irreversible nature of the modification raises questions about the observed reduction in interaction during hypoxia. Does the result suggest that the modified Beclin1 is

degraded while the unmodified version of Beclin1 is being de novo synthesized during 8h of hypoxia, a condition that typically suppresses overall protein synthesis? Additionally, the reported increase in Vps34 activity under glucose starvation is not entirely supported by previous literature, and the assay conditions should be scrutinized, with the inclusion of necessary proteins and a positive control for validation. Please check the comment #3 above related to this issue.

Answer: As described in the aforementioned discussion, dehydroxylation and hydroxylation can be dynamically regulated depending on the signaling input.

8. The role of VHL in the proposed mechanism is still not clearly understandable. If VHL serves as a negative regulator, suppressing autophagy under normal conditions (normoxia, PHD1 active state), what are the implications of this role for cellular responses in normal wild type cells to regulate autophagy? Does VHL simply maintain the inactive complex without being subject to any regulation?

Answer: As described in the aforementioned discussion, dehydroxylation and hydroxylation can be dynamically regulated depending on the signaling input.

9. The irreversibility of the modification raises questions about its regulatory role, given that it results in a dead-end product according to the proposed mechanism. The purpose of forming such an inactive, dead-end product in cells remains not entirely clarified.

Answer: As described in the aforementioned discussion, dehydroxylation and hydroxylation can be dynamically regulated depending on the signaling input.

Other points:

The manuscript has numerous issues related to the method section and figure legends. It requires more comprehensive details for each experiment and figure, including incubation times, buffer compositions, and other key details. For instance, more information is needed regarding how the amino acid starvation medium was prepared, the microscopy analysis of LC3, the in vitro PI(3)P assay, and the quantification of autophagy assays. The term "autophagy induction" is used improperly in some instances. The presented assays are insufficient for evaluating autophagy induction.

Answer: We have revised the manuscript accordingly.

Referee 2 has, however, replied with a list of requested clarifications. These centre on the consequences of proline hydroxylation in this physiological context. I believe these points can be addressed adequately by expanding the discussion on the irreversibility of this modification.

Answer: We have expanded the discussion on the irreversibility of hydroxylation.

Please detail the exact experimental procedures followed as requested by referee 2.

Answer: We have detailed the experimental procedures requested by referee 2.

Dear Zhimin,

I have now received a re-review report from Referee #2, who was generally unsatisfied with the changes you have made to the text. Please read the report (copied below) carefully and address all of the concerns in a revised version of the manuscript. As we discussed over Zoom, I do not consider extra experiments to be necessary (beyond the data clarifications requested). I understand your argument to be that hydroxylation status of proline residues can be regulated via signalling inputs via regulating rates of hydroxylation and protein turnover, but please take care to address the referee's valid requests directly in focussed additions to the discussion and method sections.

Thank you for the opportunity to consider your work for publication. I look forward to your revision.
Best wishes,

William

William Teale, PhD
Editor
The EMBO Journal
w.teale@embojournal.org

We realize that it is difficult to revise to a specific deadline. In the interest of protecting the conceptual advance provided by the work, we recommend a revision within 3 months (6th Mar 2024). Please discuss the revision progress ahead of this time with the editor if you require more time to complete the revisions.

Referee #2:

The revision has only minimally addressed the remaining issues. There is a misconception regarding dehydroxylation by the

authors. The publications referenced are not pertinent to proline hydroxylation or even protein hydroxylation. The authors should acknowledge the absence of published work on dehydroxylation events in hydroxylated prolines or any demodifying enzymes reported to counteract PHDs or P4HAs. Hence, the arguments presented by the authors lack justification. The following numbering corresponds to the previous comments.

1. The authors did not address these issues: (1) The claim "dehydroxylation and hydroxylation of proline can be dynamically regulated based on signaling input" is inaccurate. There is no supporting evidence in the literature. The authors have listed irrelevant publications; (2) State clearly whether the band observed in blots corresponds to LC3B I or LC3B II; (3) Provide quantitation for the flux data; (4) Justify the use of such a short-term 2-hour glucose starvation condition for the flux assay by providing references supporting the condition.
2. These issues were not addressed: (1) The assertion that enhancing Atg14-Beclin1 interaction alone can increase autophagy flux seems oversimplified, considering the complex nature of autophagy. A more reasonable explanation is necessary in the Discussion section; (2) The role of Beclin1 modification in 786-O or RCC4 cells lacking VHL expression is obscure. This needs a reasonable explanation in the Discussion section.
3. This issue remains unaddressed: Figures EV6E and 4J should include measurements of Beclin1 and Vps34 amounts.
4. Although the authors claim that detailed information is provided, the Method section lacks relevant details. LC3B puncta images in Figures 1E, 3D, 4H and 4L are still problematic. It is not entirely clear if the observed dots represent autophagosomes. Further clarification on how those dots were identified as autophagosomes or the criteria for counting are necessary. Even, the Method section does not specify the microscope used.
5. The authors have included Atg14L blots in Figure 3G. The Atg14L-Beclin 1 interaction is barely detected in PHD1 WT cells according to the new data. Does this suggest impaired autophagy in the WT cells (or normal cells)? The authors still need to provide a plausible explanation for the physiological role of the modification in regulating autophagy. This does not need any new experiment but a reasonable explanation about how the modification acts in the autophagy pathway. The authors' assertion regarding the reversibility of the modification is incorrect. There is no documented evidence supporting the reversibility of proline hydroxylation or the existence of demodifying enzymes acting against PHDs or P4HAs.
6. This issue has been raised in the original version, but the authors have not yet addressed it. It is important to validate those amounts to ensure the proper execution of the experiment, especially considering that immunoprecipitates used can exhibit variability across samples.
7. This issue remains unaddressed. The authors have not provided sufficient explanations or responses to this previous comment: 'The physiological relevance of the findings (what the modification actually does) still needs more clarification. The irreversible nature of the modification raises questions about the observed reduction in interaction during hypoxia. Does the result suggest that the modified Beclin1 is degraded while the unmodified version of Beclin1 is being de novo synthesized during 8h of hypoxia, a condition that typically suppresses overall protein synthesis? Additionally, the reported increase in Vps34 activity under glucose starvation is not entirely supported by previous literature, and the assay conditions should be scrutinized, with the inclusion of necessary proteins and a positive control for validation.'
8. The authors have not yet provided a reasonable explanation regarding this previous comment: 'The role of VHL in the proposed mechanism is still not clearly understandable. If VHL serves as a negative regulator, suppressing autophagy under normal conditions (normoxia, PHD1 active state), what are the implications of this role for cellular responses in normal wild type cells to regulate autophagy? Does VHL simply maintain the inactive complex without being subject to any regulation?' No new experiments are needed. Instead of repeating information already in the Results section, the authors should use the Discussion section to provide explanations on this issue or actually how the modification acts in the autophagy pathway.
9. The authors should offer a reasonable explanation in the Discussion section addressing this previous comment: 'The irreversibility of the modification raises questions about its regulatory role, given that it results in a dead-end product according to the proposed mechanism. The purpose of forming such an inactive, dead-end product in cells remains not entirely clarified.'
10. Some improvements are noted, yet clarity issues persist. The authors should provide explicit information in the Methods section. Some examples are noted below.
 - It is not entirely clear how the amino acid-free medium was prepared. Did the authors use Invitrogen 11965-092 or prepare the medium themselves? If self-prepared, the method should detail the elements used and catalog numbers of the elements. Also, state if the medium included dialyzed FBS or not.
 - Many catalog numbers for items, especially in the cell lines and cell culture conditions section, are missing.
 - The authors should offer more specific details about the imaging experiments. For example, they need to specify the microscope used, including the vendor name and system name.

- The section describing the autophagy assay requires clarification regarding the criteria used to classify LC3B dots as autophagosomes for counting.
- Provide more information in the immunoprecipitation and immunoblotting analysis section by detailing the composition of the modified buffer used and briefly describing the procedure, rather than simply referencing prior literature.
- Further elaboration is needed regarding the mass spectrometry analysis. Explanation on the procedure for immunoprecipitation, the isolation and processing of immunoprecipitated proteins for digestion, and related steps is needed.
- The Subcellular fractionation section needs more information.
- The Dot immunoblotting assays section needs more information.

Referee #3:

The revised manuscript addressed all of previous critiques and is acceptable.

Referee #2:

The revision has only minimally addressed the remaining issues. There is a misconception regarding dehydroxylation by the authors. The publications referenced are not pertinent to proline hydroxylation or even protein hydroxylation. The authors should acknowledge the absence of published work on dehydroxylation events in hydroxylated prolines or any demodifying enzymes reported to counteract PHDs or P4HAs. Hence, the arguments presented by the authors lack justification. The following numbering corresponds to the previous comments.

Answer: We sincerely appreciate the reviewer's careful examination of the manuscript and insightful comments, which have significantly enhanced the quality of the work.

1. The authors did not address these issues: (1) The claim "dehydroxylation and hydroxylation of proline can be dynamically regulated based on signaling input" is inaccurate. There is no supporting evidence in the literature. The authors have listed irrelevant publications; (2) State clearly whether the band observed in blots corresponds to LC3B I or LC3B II; (3) Provide quantitation for the flux data; (4) Justify the use of such a short-term 2-hour glucose starvation condition for the flux assay by providing references supporting the condition.

Answer: (1) We concur with the reviewer that the study of proline dehydroxylation in proteins remains underexplored. Nevertheless, evidence exists for the reversibility of protein hydroxylation as a post-translational modification (PMID: 32759169). Rodriguez et al. demonstrated the dehydroxylation of asparagine hydroxylation on intact proteins, suggesting that 'asparagine hydroxylation is a flexible and dynamic post-translational modification akin to modifications involved in regulating signaling networks, such as phosphorylation, methylation, and ubiquitylation' (quoted from the published article). In addition, autophagy can be influenced by protein-protein interactions and/or other posttranslational modifications, such as phosphorylation, acetylation, and ubiquitylation. These interactions and/or modifications may impact Beclin1 hydroxylation and/or the binding of VHL to hydroxylated Beclin1 under various physiological conditions including energy stress. Thus, it is unlikely that VHL permanently maintains the inactive complex without being subject to any regulations. We have included the relevant discussion in the main text (Page 19-20).

(2) We have labeled LC3B-I or LC3B-II in all figures.

(3) Quantitation for the flux data was provided.

(4) References supporting a short-term 1 or 2-hour glucose starvation condition for the flux assay have been provided (PMID: 34107300; 24014036; 29507183; 28661473; 20176105; 33682133; 24141421).

2. These issues were not addressed: (1) The assertion that enhancing Atg14-Beclin1 interaction alone can increase autophagy flux seems oversimplified, considering the complex nature of autophagy. A more reasonable explanation is necessary in the Discussion section; (2) The role of Beclin1 modification in 786-O or RCC4 cells lacking VHL expression is obscure. This needs a reasonable explanation in the Discussion section.

Answer: (1) In our manuscript, we didn't claim that enhancing Atg14-Beclin1 interaction alone is sufficient to increase autophagy flux. Our results showed that the interaction is required but not sufficient to induce autophagy, which still needs stimulation (such as glucose starvation). (2) The role of Beclin1 modification in 786-O or RCC4 cells lacking VHL expression is not clear. We cannot exclude the possibility that this Beclin1 modification may affect autophagy-irrelevant, but known or unknown Beclin1-dependent functions, such as vesical trafficking. We have included the relevant discussion into the manuscript (Page 19-20).

3. This issue remains unaddressed: Figures EV6E and 4J should include measurements of Beclin1 and Vps34 amounts.

Answer: We have included the quantification of Beclin1 and Vps34 amounts.

(1) As anticipated, Atg14L immunoprecipitates contained reduced levels of Beclin1 and Vps34 in HA-VHL overexpressed 786-O or RCC4 cells expressing WT Beclin1 compared to those expressing Beclin1 P54A (Appendix Figure S3E). In addition, Beclin1 P54A knock-in expression eliminated the ectopically expressed VHL-mediated suppression of ATG14L-linked VPS34 activity in 786-O or RCC4 cells (Appendix Figure S3E).

(2) Consistently, Atg14L immunoprecipitates contained increased levels of Beclin1 and VPS34 in VHL-proficient SN12C and TK-10 cells expressing Beclin1 P54A compared to those expressing WT Beclin1 (Appendix Figure S3F). Of note, Beclin1 P54A expression lost its regulation on Beclin1-ATG14L interaction, as well as ATG14L-linked VPS34 activity in VHL-depleted SN12C and TK-10 cells (Appendix Figure S3F).

These results suggest that the activity of ATG14L-lined Vps34 regulated by PHD1-dependent Beclin1 P54 hydroxylation and VHL correlates with the amount of Vps34 associated with Atg14L. We agree with the reviewer that these results may not be crucial and have consequently moved them to supplementary data (Appendix Figures S3E and S3F).

4. Although the authors claim that detailed information is provided, the Method section lacks relevant details. LC3B puncta images in Figures 1E, 3D, 4H and 4L are still problematic. It is not entirely clear if the observed dots represent autophagosomes. Further clarification on how those dots were identified as autophagosomes or the criteria for counting are necessary. Even, the Method section does not specify the microscope used.

Answer: In Figures 1E, 3D, 4H, and 4L, we investigated the distribution and expression of endogenous LC3B. Since lysosomal inhibitors were not employed, we replaced the term 'autophagosome' with 'autophagosomes/autolysosomes' in the manuscript. The criteria for counting dots and the details of the microscope used were provided in the Methods section.

5. The authors have included Atg14L blots in Figure 3G. The Atg14L-Beclin 1 interaction is barely detected in PHD1 WT cells according to the new data. Does this suggest impaired autophagy in the WT cells (or normal cells)? The authors still need to provide a plausible explanation for the physiological role of the modification in regulating autophagy. This does not need any new experiment but a reasonable explanation about how the modification acts in the autophagy pathway. The authors' assertion regarding the reversibility of the modification is incorrect. There is no documented evidence supporting the reversibility of proline hydroxylation or the existence of demodifying enzymes acting against PHDs or P4HAs.

Answer: A moderate interaction between Atg14L and Beclin1 in PHD1 WT VHL-proficient SN12C cells in Figure 3G is due to normal cell culture conditions used for this experiment. Under energy-sufficient conditions, autophagy level is reasonably low. Consistently, expression of catalytically inactive PHD1 D311A mutant inhibited the interaction between VHL and Beclin1 with correspondingly increased binding of ATG14L to Beclin1 in VHL-proficient SN12C cells (Figure 3G).

As aforementioned in the answer to Question 1, we have provided a discussion regarding the potential reversibility of hydroxylation and the regulation of the protein complex by potential unidentified protein-protein interactions and other types of modifications induced by different physiological conditions.

6. This issue has been raised in the original version, but the authors have not yet addressed it. It is important to validate those amounts to ensure the proper execution of the experiment, especially considering that immunoprecipitates used can exhibit variability across samples.

Answer: We have included the amounts of VPS34, Beclin1, and VHL into the revised Figure EV1L.

7. This issue remains unaddressed. The authors have not provided sufficient explanations or responses to this previous comment: 'The physiological relevance of the findings (what the modification actually does) still needs more clarification. The irreversible nature of the modification raises questions about the observed reduction in interaction during hypoxia. Does the result suggest that the modified Beclin1 is degraded while the unmodified version of Beclin1 is being de novo synthesized during 8h of hypoxia, a condition that typically suppresses overall protein synthesis? Additionally, the reported increase in Vps34 activity under glucose starvation is not entirely supported by previous literature, and the assay conditions should be scrutinized, with the inclusion of necessary proteins and a positive control for validation.'

Answer: As aforementioned in the answer to Question 1, we have provided a discussion regarding the potential reversibility of hydroxylation and the regulation of the protein complex by potential unidentified protein-protein interactions and other types of modifications induced by different physiological conditions.

In addition, we have provided all necessary controls for Vps34 activity under glucose starvation, as described in the answers to Question 3 and 6, as well as input controls for the assay.

8. The authors have not yet provided a reasonable explanation regarding this previous comment: 'The role of VHL in the proposed mechanism is still not clearly understandable. If VHL serves as a negative regulator, suppressing autophagy under normal conditions (normoxia, PHD1 active state), what are the implications of this role for cellular responses in normal wild type cells to regulate autophagy? Does VHL simply maintain the inactive complex without being subject to any regulation?' No new experiments are needed. Instead of repeating information already in the Results section, the authors should use the Discussion section to provide explanations on this issue or actually how the modification acts in the autophagy pathway.

Answer: We demonstrated differential regulation of autophagy in normal wild-type cells and VHL-deficient cells. However, our findings do not imply that this regulation is the sole mechanism in autophagy regulation. As aforementioned, autophagy can be influenced by protein-protein interactions and/or other posttranslational modifications, such as phosphorylation, acetylation, and ubiquitylation. These interactions and/or modifications may impact Beclin1 hydroxylation and/or the binding of VHL to Beclin1 under various physiological conditions including energy stress. Thus, it is unlikely that VHL permanently maintains the inactive complex without being subject to any regulations. We have included the relevant discussion in the main text (Page 19-20).

9. The authors should offer a reasonable explanation in the Discussion section addressing this previous comment: 'The irreversibility of the modification raises questions about its regulatory role, given that it results in a dead-end product according to the proposed mechanism. The purpose of forming such an inactive, dead-end product in cells remains not entirely clarified.'

Answer: As aforementioned in the answer to Question 1, we have provided a discussion regarding the potential reversibility of hydroxylation and the regulation of the protein complex by potential unidentified protein-protein interactions and other types of modifications induced under different physiological conditions.

10. Some improvements are noted, yet clarity issues persist. The authors should provide explicit information in the Methods section. Some examples are noted below.

- It is not entirely clear how the amino acid-free medium was prepared. Did the authors use Invitrogen 11965-092 or prepare the medium themselves? If self-prepared, the method should detail the elements used and catalog numbers of the elements. Also, state if the medium included dialyzed FBS or not.

Answer: The amino acid-free medium was prepared in house following the Invitrogen (#11965-092) recipe (but without all amino acids), and all components were listed in the revised method. Amino acid starvation medium contained 10% dialyzed FBS. We have included this information in the revised method.

- Many catalog numbers for items, especially in the cell lines and cell culture conditions section, are missing.

Answer: The catalog numbers were provided in the revised method.

- The authors should offer more specific details about the imaging experiments. For example, they need to specify the microscope used, including the vendor name and system name.

Answer: The microscope information is included in the revised method.

- The section describing the autophagy assay requires clarification regarding the criteria used to classify LC3B dots as autophagosomes for counting.

Answer: Dots were counted by setting 8-bit pictures at the same threshold (with low and high threshold set at 105 and 255, respectively) in Image J software. The criteria for counting dots and the details of the microscope used were provided in the Methods section.

- Provide more information in the immunoprecipitation and immunoblotting analysis section by detailing the composition of the modified buffer used and briefly describing the procedure, rather than simply referencing prior literature.

Answer: Composition of the modified buffer and procedure for immunoprecipitation and immunoblotting analysis were described briefly in the revised method.

- Further elaboration is needed regarding the mass spectrometry analysis. Explanation on the procedure for immunoprecipitation, the isolation and processing of immunoprecipitated proteins for digestion, and related steps is needed.

Answer: Elaboration for the mass spectrometry analysis was added in the revised method.

- The Subcellular fractionation section needs more information.

Answer: Subcellular fractionation procedures were described briefly in the revised method.

- The Dot immunoblotting assays section needs more information.

Answer: Dot immunoblotting assays were described with more detail in the revised method.

Dear James,

Referees 2 is now largely satisfied with the revisions you have made (please see the report below). However, some issues around referencing and your experimental design need some further minor discussion and clarification. Please indicate in a point-by-point response how these concerns were addressed.

Thank you for the opportunity to consider your work for publication. I look forward to your revision.

Yours sincerely,

William

William Teale, PhD
Editor
The EMBO Journal
w.teale@embojournal.org

We realize that it is difficult to revise to a specific deadline. In the interest of protecting the conceptual advance provided by the work, we recommend a revision within 3 months (17th Apr 2024). Please discuss the revision progress ahead of this time with the editor if you require more time to complete the revisions.

Referee #2:

1. (1) Acceptable; (2) Well addressed; (3) Well addressed; (4) Still needs clarification.

The references indicated by the authors do not contain data supporting the experimental condition used in the manuscript.

- PMID 34107300: Lacks autophagy flux data for endogenous LC3B.
- PMID 24014036: Absence of autophagy flux data using CQ or BafA1. Also, the paper indicates that glucose starvation suppresses autophagy.
- PMID 29507183: A good assay example, yet, like the previous paper, it shows that glucose starvation suppresses autophagy flux.
- PMID 28661473: No autophagy flux data found. Which figures do the authors refer to?
- PMID 20176105: No autophagy flux data found. Which figures do the authors refer to?
- PMID 33682133: No autophagy flux data found. Which figures do the authors refer to?
- PMID 24141421: No autophagy flux data found. Which figures do the authors refer to?

As noted in the comments on the original manuscript, the authors should have used amino acid starvation instead of glucose starvation for this study. The impact of glucose starvation on autophagy is currently under scrutiny and in dispute. The authors have not provided supporting literature for the chosen experimental condition, and it remains unclear why the data are inconsistent with previous reports. To address this concern, the authors should provide a rationale for choosing glucose starvation over amino acid starvation, the typically robust condition for inducing autophagy, in the face of the ongoing controversy and inconsistency in the field. As part of this, it is important to specify that the assays were conducted almost exclusively in the specific cancer cells, not in widely used cell lines, like HeLa, thus limiting the relevance of the findings to those specific cancer cell types. This concern should be addressed and discussed throughout the whole manuscript.

2. (1) Acceptable. (2) Please include a discussion specifically about the situation in 786-O and RCC4 cells, which lack VHL, regarding how the modification might function independently of VHL.
3. (1) Although the authors mention the quantification of Beclin 1 and Vps34 amount, the data are not found in Appendix Figure S3E. It is presumed to be Appendix Figure S4E, but even this figure is the same as the previous version without the quantification. (2) Appendix Figure S3F (or S4F?) appears unrelated to this issue.
4. The updated information on autophagy assay section in Method has been improved. However, this section or the relevant figure legend should include more details, such as the number of cells analyzed and provide more in-depth procedural information.
5. Acceptable.
6. Well addressed.
7. A positive control for the assay, validating the experimental condition, is not provided. Addressing this issue may be challenging at this stage, and this issue could be considered for exclusion.
8. Please provide additional discussion on the specific situation in those cancer cells lacking VHL regarding the role of the modification.
9. Acceptable.
10. Improved.

Referee #2:

1. (1) Acceptable; (2) Well addressed; (3) Well addressed; (4) Still needs clarification.

The references indicated by the authors do not contain data supporting the experimental condition used in the manuscript.

- PMID 34107300: Lacks autophagy flux data for endogenous LC3B.

Answer: Figure 3C, 3D and S1L in PMID 34107300 show that endogenous LC3 puncta increased about 5-10 folds upon glucose-free treatment for 1h, and Figure S3E shows that lysosome numbers were increased about 2 folds at the same conditions. In addition, it is well documented that glucose deficiency increases biogenesis of autophagosome and lysosome for enhanced autophagy (PMID 28552616). These results suggest that the endogenous autophagy flux is enhanced under short-term glucose-free conditions. Notably, as shown in Figure 6A in PMID 34107300, enhanced autophagy flux was indeed observed upon glucose-free treatment for 1h by using mRFP-GFP-LC3. Taken together, these published results, which are consistent with our findings, strongly suggested that short-term glucose starvation induces autophagy flux.

- PMID 24014036: Absence of autophagy flux data using CQ or BafA1. Also, the paper indicates that glucose starvation suppresses autophagy.

Answer: In Figure 4A, 4C and 5C of PMID 24014036, the authors detected no changes in autophagy flux upon 3-6 h glucose deprivation in Hela cells (which lacks LKB1) and Rh4 cells (which did not exhibit detectable p-AMPK or p-ACC). In addition, the authors' conclusion that glucose starvation suppresses autophagy is basing on addition of protease inhibitors E64d and pepstatin A (abbreviated as EP) upon prolonged glucose deprivation. Thus, the authors' conclusion may be not accurate because of utilization of cells that do not have detectable AMPK activation and application of EP that does not connect to lysosome function directly.

- PMID 29507183: A good assay example, yet, like the previous paper, it shows that glucose starvation suppresses autophagy flux.

- PMID 28661473: No autophagy flux data found. Which figures do the authors refer to?

- PMID 20176105: No autophagy flux data found. Which figures do the authors refer to?

- PMID 33682133: No autophagy flux data found. Which figures do the authors refer to?

- PMID 24141421: No autophagy flux data found. Which figures do the authors refer to?

Answer: As aforementioned discussion (about PMID 34107300), our results and published results suggest that short-term glucose starvation induces autophagy flux.

As noted in the comments on the original manuscript, the authors should have used amino acid starvation instead of glucose starvation for this study. The impact of glucose starvation on autophagy is currently under scrutiny and in dispute. The authors have not provided supporting literature for the chosen experimental condition, and it remains unclear why the data are inconsistent with previous reports. To address this concern, the authors should provide a rationale for choosing glucose starvation over amino acid starvation, the typically robust condition for inducing autophagy, in the face of the ongoing controversy and inconsistency in the field. As part of this, it is important to specify that the assays were conducted almost exclusively in the specific cancer cells, not in widely used cell lines, like HeLa, thus limiting the relevance of the findings to those specific cancer cell types. This concern should be addressed and discussed throughout the whole manuscript.

Answer: Literatures sufficiently document that glucose deficiency is a common metabolic stress in tumor microenvironment during cancer progression (PMID: 21376230, 38020920). In addition, it has been well published that autophagy is critical for cancer cell survival upon glucose deficiency stress (PMID: 24862259, 37594406, 28552616). Our results generated with established experimental conditions are consistent with the findings in PMID 34107300, 28661473, 20176105, 33682133 and 24141421, supporting that short time glucose deprivation promotes autophagy (although not many literatures have autophagy flux results).

We agree with the reviewer that amino acid starvation is a typically robust condition for inducing autophagy. As suggested, we also used amino acid starvation to induce autophagy and repeated the key experiments. Similar to the effects on glucose deprivation-induced autophagy, reconstituted expression of Beclin1 P54A increased amino acid starvation-induced autophagy compared to its WT counterpart in VHL-proficient ccRCC cells (Fig EV4I). In addition, expression of Flag-VHL in the VHL-deficient ccRCC cells suppressed amino acid starvation-induced autophagy in the tumor cells expressing WT Beclin1, but not Beclin1 P54A (Fig EV4J). These results suggest that Beclin1 P54 hydroxylation also plays a critical role in regulation of amino acid-induced autophagy.

We agree with the reviewer that the assays should be performed in more cell lines to confirm our findings. As suggested, we also used HCT116 human colon cancer cells, MDA-MB-231 human breast cancer cells, and HeLa human cervical cancer cells to repeat the key experiments. We showed that depletion of PHD1 (Appendix Fig S4A) or reconstituted expression of Beclin1 P54A (Appendix Fig S4B and C) disrupted the binding of VHL to Beclin1 with correspondingly increased interaction between ATG14L and Beclin1 in these tumor cells. In addition, compared to expression of WT Beclin1, Beclin1 P54A expression enhanced glucose deprivation-induced autophagy (Appendix Fig S4D). Consistently, overexpression of Flag-VHL suppressed autophagy in these cells expressing WT Beclin1, but not Beclin1 P54A upon glucose

deprivation (Appendix Fig S4E). These data suggest that PHD1-dependent Beclin1 hydroxylation and VHL-suppressed autophagy is not cell line-specific. Nevertheless, as the reviewer indicated, autophagy in tumors can be regulated simultaneously by both intrinsic and extrinsic factors, involving a complex set of inputs. Further studies on autophagy in rapidly developing tumors in vivo are needed. We have included relevant discussion in the manuscript (Page 20).

2. (1) Acceptable. (2) Please include a discussion specifically about the situation in 786-O and RCC4 cells, which lack VHL, regarding how the modification might function independently of VHL.

Answer: We have included the discussion (on Page 20).

3. (1) Although the authors mention the quantification of Beclin 1 and Vps34 amount, the data are not found in Appendix Figure S3E. It is presumed to be Appendix Figure S4E, but even this figure is the same as the previous version without the quantification. (2) Appendix Figure S3F (or S4F?) appears unrelated to this issue.

Answer: We included the Beclin1 and VPS34 expression and quantification in Appendix Figure S3E (previous Figure 4J) and Appendix Figure S3F (previous Figure EV6E).

(1) As anticipated, Atg14L immunoprecipitates contained reduced levels of Beclin1 and Vps34 in HA-VHL overexpressed 786-O or RCC4 cells expressing WT Beclin1 compared to those expressing Beclin1 P54A (Appendix Figure S3E). In addition, Beclin1 P54A knock-in expression eliminated the ectopically expressed VHL-mediated suppression of ATG14L-linked VPS34 activity in 786-O or RCC4 cells (Appendix Figure S3E).

(2) Consistently, Atg14L immunoprecipitates contained increased levels of Beclin1 and VPS34 in VHL-proficient SN12C and TK-10 cells expressing Beclin1 P54A compared to those expressing WT Beclin1 (Appendix Figure S3F). Of note, Beclin1 P54A expression lost its regulation on Beclin1-ATG14L interaction, as well as ATG14L-linked VPS34 activity in VHL-depleted SN12C and TK-10 cells (Appendix Figure S3F).

These results suggest that the activity of ATG14L-lined Vps34 regulated by PHD1-dependent Beclin1 P54 hydroxylation and VHL correlates with the amount of Vps34 associated with Atg14L. We agree with the reviewer that these results may not be crucial and have consequently moved them to supplementary data (Appendix Figures S3E and S3F).

4. The updated information on autophagy assay section in Method has been improved. However, this section or the relevant figure legend should include more details, such as the number of cells analyzed and provide more in-depth procedural information.

Answer: We have included the number of cells analyzed in figure legends and Method.

5. Acceptable.

6. Well addressed.

7. A positive control for the assay, validating the experimental condition, is not provided. Addressing this issue may be challenging at this stage, and this issue could be considered for exclusion.

8. Please provide additional discussion on the specific situation in those cancer cells lacking VHL regarding the role of the modification.

Answer: We have included the discussion (on Page 20)

9. Acceptable.

10. Improved.

Dear James,

I am pleased to inform you that your manuscript has been accepted for publication in the EMBO Journal.

Congratulations to you and your team! This is a really impressive piece of work.

Best wishes,

William

William Teale, PhD
Editor
The EMBO Journal
w.teale@embojournal.org
